# A fully autonomous robotic ultrasound system for thyroid scanning

Kang Su [1,9], Jingwei Liu [1,9], Xiaoqi Ren[2,3,9], Yingxiang Huo [2,3,9], Guanglong Du [1,9] ✉, Wei Zhao[4], Xueqian Wang [5] ✉, Bin Liang[6] ✉, Di Li[7] & Peter Xiaoping Liu [8] ✉

The current thyroid ultrasound relies heavily on the experience and skills of the sonographer and the expertise of the radiologist, and the process is physically and cognitively exhausting. In this paper, we report a fully autonomous robotic ultrasound system, which is able to scan thyroid regions without human assistance and identify malignant nod- ules. In this system, human skeleton point recognition, reinforcement learning, and force feedback are used to deal with the difficulties in locating thyroid targets. The orientation of the ultrasound probe is adjusted dynamically via Bayesian optimization. Experimental results on human participants demonstrated that this system can perform high-quality ultrasound scans, close to manual scans obtained by clinicians. Additionally, it has the potential to detect thyroid nodules and provide data on nodule characteristics for American College of Radiology Thyroid Imaging Reporting and Data System (ACR TI-RADS) calculation.

Ultrasound (US) diagnosis is widely used in examining organs, such as the liver, gallbladder, pancreas, spleen, kidney, and thyroid[1-6]. How- ever, the qualities of US diagnosis rely heavily on the experience and skills of the sonographer and radiologist[7-9]. The acquisition of US images usually exhibits variability among clinicians, and even the same examiner may potentially produce very different results from different scans[10]. Furthermore, the standard practice of having patients to assume a supine position and maintain still- ness during examinations can present challenges, as these requirements may not always be met[11]. Consequently, operator-dependency and patient-specific factors introduce inconsistency and unreliability into US diagnosis results[12].

From pure human control to complete autonomy, the level of autonomy of medical robots can be classified into different categories[13,14]. According to the framework presented in ref. 13, the level of autonomy includes level 0, which is defined as no autonomy, e.g., tele-operated systems or prosthetic devices; level 1, robotic

assistance, the robot guides the human during a task while the human maintains a continues control; level 2, task autonomy, the robot pro- vides discrete rather than continuous control over a specific task; level 3, conditional autonomy, the system is capable of generating different task strategies but relies on the human's selection or approvement; level 4, high autonomy, the system is capable of making medical decisions but only when supervised by a qualified physician; level 5, full autonomy, the robot can perform the entire procedure without any human involvement. Under the concept of the autonomy level of medical robots (Level 0–5), the autonomy level of ultrasonic inspec- tion robots (Level 0–3) has been defined at present. Level 0 is defined as "manual probe manipulation", the proposed tele-echography systems[15-19]. The US robotic system in Level 1 utilizes visual servo technology to allow the robot to automatically track desired image features[20-23] and compensate for unnecessary patient movement dur- ing remote operation. Level 2 is described as performing autonomous

[1]School of Computer Science and Engineering, South China University of Technology, Guangzhou 510006, China. [2]School of Future Technology, South China University of Technology, Guangzhou 511442, China. [3]Peng Cheng Laboratory, Shenzhen 518000, China. [4]Division of Vascular and Interventional Radiology, Nanfang Hospital Southern Medical University, Guangzhou 510515, China. [5]Tsinghua Shenzhen International Graduate School, Tsinghua University, Shenzhen 518055, China. [6]Department of Automation, Tsinghua University, 100854 Beijing, China. [7]School of Mechanical and Automotive Engineering, South China University of Technology, Guangzhou 510641, China. [8]Department of Systems and Computer Engineering, Carleton University, Ottawa, ON K1S 5B6, Canada. [9]These authors contributed equally: Kang Su, Jingwei Liu, Xiaoqi Ren, Yingxiang Huo, Guanglong Du. ✉e-mail: csgldu@scut.edu.cn; wang.xq@sz.tsinghua.edu.cn; bliang@tsinghua.edu.cn; xpliu@sce.carleton.ca

US acquisition along a manually planned path[24,25]. The US robotic system in Level 3 can autonomously plan and perform US acquisition without any instruction from a human operator but requires the supervision of an operator[26–29].

With the improvement of the autonomy of ultrasonic inspection of robots, substantial advancements in the field have been achieved, providing promising solutions to improve the accuracy and efficiency of US procedures. The prerequisite for robotic US acquisitions is to plan the scanning path to ensure finding a desired imaging plane or covering a selected region of interest. In general, existing systems typically rely on global information of the target tissue acquired from preoperative medical images or surface information obtained from external sensors[27,30,31]. Given the inherent variability in individual anatomy and the dynamic of human motion, executing a scanning task based on a predetermined trajectory presents significant challenges. To address these difficulties, Jiang et al.[32] integrated the feedback of segmented images into the control process. Zhan et al.[33] have proposed a visual servoing framework for motion compensation. However, these methods usually assume that the target features exist in the US image, and once the features are lost, the control methods may fail. In an effort to apply online image-guided methods to define the scanning trajectory and derive clinically relevant information out of the 3D reconstructed image, Zielke et al.[34] implemented in-plane navigation specifically designed for robotic sonography in thyroid volumetry. However, in actual clinical practice, sonographers often use a combination of multiple views, such as transverse and longitudinal scans, for the identification and diagnosis of both benign and malignant pathologies[35]. Furthermore, the increased autonomy of the US robotic system may lead to a higher risk of injury to patients due to machine failure, so the clinical effectiveness of the system needs to be further studied. Although many implementations have been proposed[36–40], the overall success scanned rate is low due to the differences between human bodies. To this end, the fully autonomous robotic diagnosis system adapted to clinical practice is still challenging, as it calls for more perception, planning, and control on the part of the robot while taking into account patient safety.

In order to eliminate the above obstacles, we developed a fully autonomous robotic ultrasound system (FARUS), as shown in Fig. 1. To the best of our knowledge, this is the first in-human study of fully autonomous robotic US scanning for thyroid. In conventional US examinations, the process involves a division of responsibilities between sonographer and radiologist. However, the presented FARUS integrates the both roles into a single autonomous unit. Here, we achieved a human-like fusion of both in-plane and out-of-plane scanning, allowing for comprehensive scanning of the thyroid region, and providing a detailed evaluation of the anatomy. The FARUS overcomes the challenges associated with the localization of thyroid targets through a reinforcement learning strategy. It enables to optimize the orientation of the probe based on Bayesian optimization. It also uses deep learning techniques for real-time segmentation of the thyroid gland and potential nodules. As a result, this system provides a convenient autonomous tool integrating nodule detection, lesion localization and automatic classification.

The second contribution of this work lies in the practical realization and clinical application of the FARUS, which achieves high-quality US images in comparison with those manually collected by sonographers, and realizes accurate and real-time detection of thyroid nodules. We investigated the validity of our approach by conducting a comparative evaluation of FARUS-driven diagnostic results for thyroid nodules against the established hospital benchmark. We have conducted extensive evaluations and studied the system's performance and reliability. Our work addresses the gap between existing research and clinical application by demonstrating the deployment of this system in a real-world clinical setting.

## Results

### System design for autonomous ultrasound imaging

The robotic scanning procedure comprises four phases, mirroring the clinical workflow: thyroid searching (TS), in-plane scanning (IPS), out-of-plane scanning (OPS), and multi-view scanning (MVS), as depicted in Fig. 1a. An overview of our autonomous robotic system for thyroid scanning and real-time analysis is presented in Fig. 1e. The system consists of a six-degree-of-freedom UR3 manipulator that carries a linear US probe, a US probe fixture and a six-axis force/torque sensor. The high-frequency 2D linear US probe enables the optimal depth penetration within the superficial location of the thyroid tissue. The six-axis force/torque sensor can detect three orthogonal forces and torques between the human neck and probe. The Kinect camera tracks 3D view skeleton joints of the human body, while its 2D view provides visual feedback for the operator supervising the robotic system. It is remarkable that the entire scanning process, including thyroid searching, force control, image quality optimization, and suspected nodule detection was completed autonomously.

The following describes the thyroid scanning workflow of the proposed FARUS, as in Fig. 1b–d. First, the participant was instructed to turn his/her head after applying US gel to the neck skin. The scanning range was specified as a rectangular area with a length of 6.47 cm and a width of 5.48 cm. The contact force within the range of 2.0 N and 4.0 N was maintained to ensure sufficient pressure and prevent pressure-induced shape distortion of the thyroid anatomy. The thyroid search procedure began when the probe reached the estimated position given by skeleton joint locations. Due to individual anatomical variations, the thyroid gland may not be immediately visible in the US image obtained at the probe's estimated position. In such case, we used reinforcement learning to adjust the probe's movement until the thyroid gland is accurately located. The search procedure was finished when the gland region, segmented by our gland segmentation model, exceeds a predetermined threshold. Subsequently, the probe orientation was optimized through Bayesian optimization[41], see Fig. 1c. In the IPS procedure, the probe scans upward until the upper thyroid lobe end is invisible, then scans downward. Nodule locations are recorded during detection by our segmentation network. Out-of-plane scanning is employed for previously recorded nodules while avoiding clavicle and jaw collisions. The scanning halts if FARUS detects significant participant movement. Finally, we use the ACR TI-RADS scoring method to classify nodules as either benign or malignant, based on their distinct characteristics, see Fig. 1f.

### Deep learning for gland and nodule segmentation

For a fully autonomous robotic thyroid scan, real-time location information of thyroid lobes and nodules is crucial. Given the stable characteristics of healthy thyroid lobes and the diverse nature of nodules, we used two separate networks for the thyroid lobe and nodule segmentation tasks, respectively, as in Fig. 2a. Each model incorporates a pre-trained encoder for feature extraction and employs the UNet[42] architecture for the decoder, generating masks from extracted features. To train the nodule segmentation network with prior knowledge from the thyroid lobe mask, thyroid lobe pseudo-labels are generated for nodule images. Spatial and feature constraints are then applied to enhance nodule segmentation based on these pseudo-labels. The spatial constraint ensures proximity to the thyroid lobe, while the feature constraint emphasizes regions with similar gray values. To address overfitting with limited samples, a two-step approach is employed: pre-training on a nodule source dataset followed by fine-tuning on our smaller dataset.

To understand the segmentation preferences of the gland segmentation model across different slices, we used Grad-CAM[43] to visualize the fourth level output of the encoder. In Fig. 2b, the highlighted regions are more closely aligned with the areas of interest during central

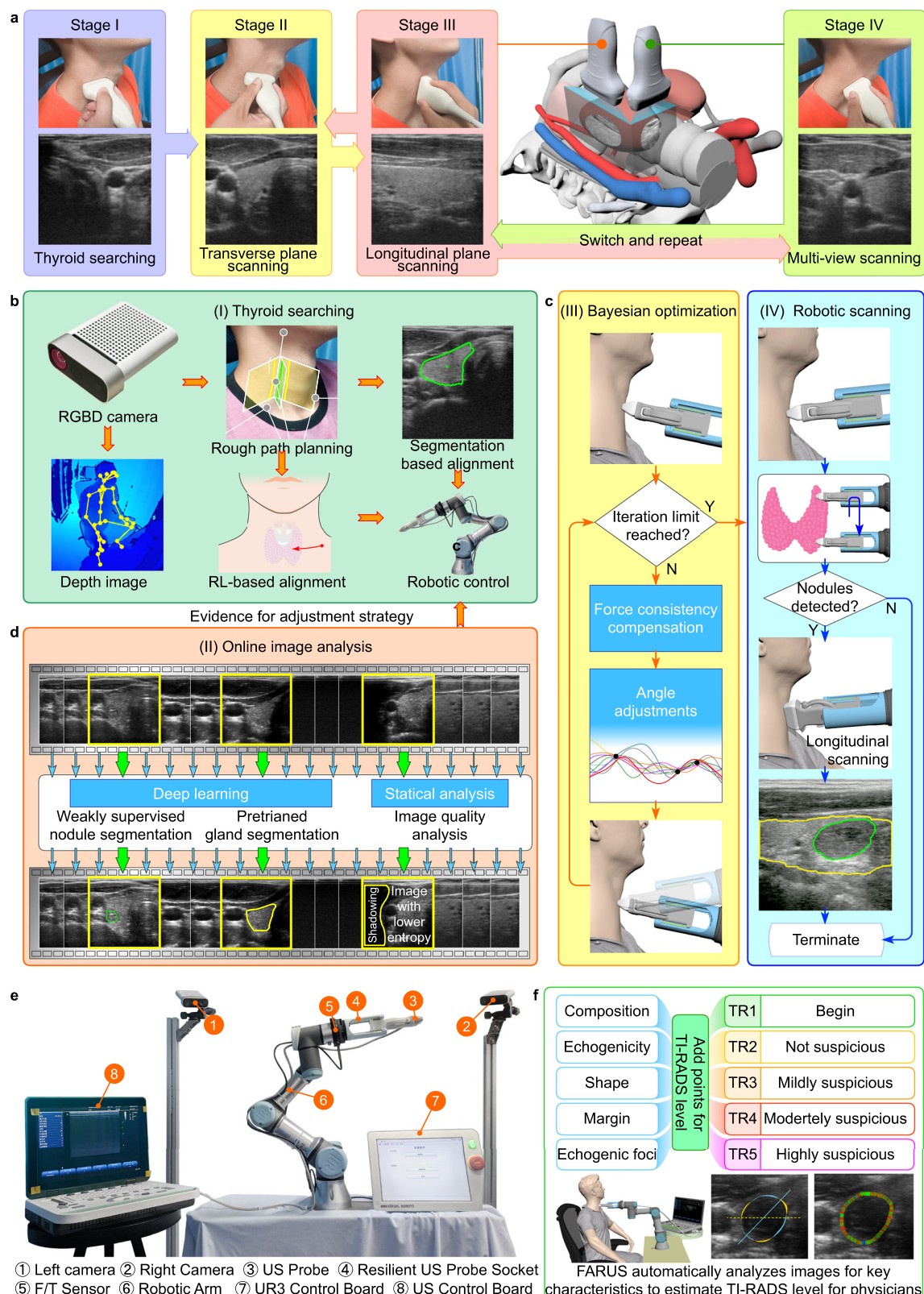

**Fig. 1 | Description of the thyroid scan procedure and the proposed FARUS.**
**a** The four-stage thyroid scan procedure used by clinical doctors. Stage I: the doctor placed a US probe below the thyroid cartilage and found the thyroid lobe in the US image; Stage II: the doctor performed IPS from the breastbone to the hyoid bone, and backward; Stage III: the doctor performed OPS to screen for thyroid disease; Stage IV: the doctor checked multi-view of the thyroid. **b**–**d** Control architecture of the full autonomous control strategy for robotic thyroid ultrasound imaging. Initially, we plan the preliminary scanning path through the human skeleton and subsequently complete the thyroid search process using reinforcement learning and thyroid segmentation. Gland and nodule identification are performed using a pretrained gland segmentation model and a weakly supervised nodule segmentation model, respectively. Throughout the scanning process, we used Bayesian optimization to adjust the scanning angle. Additionally, we combined IPS and OPS to perform multi-angle scanning for suspected nodule areas. **e** The overview of the experimental setup. **f** FARUS enables estimate TI-RADS level with key characteristics of nodules.

thyroid scanning. However, when scanning the upper pole of the thyroid, the attention map reveals other targets, such as muscles. To address cases where the thyroid target is small or even invisible, we introduced reinforcement learning to the thyroid search process, see "Thyroid search and probe orientation optimization" section. Figure 2c shows the segmentation results of our proposed VariaNet on three types of thyroid nodules. Here, VariaNet-B, VariaNet, and VariaNet+ denote models trained with different types of loss functions: no additional loss, feature loss only, and a combination of feature and spatial loss, respectively. VariaNet-T represents the refined model, trained with a dual loss function and source domain data. The integration of distance loss was critical to constrain hypoechoic nodules, which often have similar gray value distributions to other tissues, and thereby minimize the false positive rate. At the same time, the implementation of feature loss improved the segmentation performance on isoechoic nodules, especially in cases where the gray values of the nodules closely match those observed in the thyroid region. In addition, the application of transfer learning augmented with prior knowledge proved to be effective in strengthening the robustness of the network, as evidenced by the segmentation of hyperechoic nodules. In Fig. 2d, a comparative evaluation between our proposed VariaNet and other semantic segmentation models shows that the fusion of weakly supervised learning improves the segmentation capabilities of the nodule network. The proposed VariaNet exceeds the baseline model VariaNet-B by 0.97% IoU score on the SCUTN1k testset. We present ROC curves to intuitively illustrate the performance of the proposed method, proving that VariaNet outperforms other existing methods due to the tailor-made loss function, see Fig. 2e.

## Thyroid search and probe orientation optimization

An important problem in robotic thyroid scanning is the localization of the thyroid on the body surface. We present a coarse-to-fine approach to thyroid localization. In the coarse estimation step, the neck region is identified by human skeletal key points. Notably, the thyroid lobe may not be visible based on the location predicted by the skeletal data, because the anatomy of the neck varies widely in different populations. Therefore, we added the fine-tuning step to further localize the thyroid lobe. To enable robotic scanning in such a case, we used reinforcement learning to determine the location of the thyroid.

Figure 3 shows the training process of Deep Q-Network (DQN) learning with panoramic environment. The process starts with sequences of data that are collected and labeled manually. We labeled each sequence of thyroid images a goal or an ideal position for model to learn as shown in Fig. 3a. After that, each of sequence of image will be aligned and attached to be a panorama as shown in Fig. 3b. Then a bunch sequence of image will be randomized and generated into panoramas as shown in Fig. 3c, d. There is a sliding window will slide according to the action given by the agent. To mimic the real environment when sometimes the probe is not fully attached, we combine simulated view with the random noise as shadow mask, to generate imperfect thyroid images that mimic the appearance when the probe is not fully attached, as shown in Fig. 3e. Figure 3f presents the results of the training evaluation for reinforcement learning. During the exploration stage, which comprises the initial 30,000 steps. Figure 3g illustrates a progressive increase in the mean reward, indicating the gradual improvement of the RL model's performance on the given task as it continues to learn over time. Figure 3h depicts the process of thyroid scanning facilitated by the DQN Learning model in the context of FARUS. The DQN learning model enables the robotic arm to execute movements based on the input received from US images. As illustrated in Fig. 3h, the model accurately predicts the required movements to guide the robotic arm effectively.

In the first and fourth images, the probe is first attached to the patient's neck. The model then predicts the appropriate directions for the robotic arm to move in each frame. The blue bar represents the model's predictions for movement to reach the ideal position. The green bar represents the model's prediction for maintaining a stationary position once the ideal position is reached. Conversely, the yellow bar represents the model's predictions when the patient is already in motion. The DQN learning model instructs the robotic arm to move to the right based on the first thyroid image until the arm reaches the position with the ideal thyroid image, as shown in the second thyroid image. Notably, our proposed FARUS can effectively guide the robotic arm even when the patient is moving. For example, at 6.3 s, when the patient's neck moves to the left, the model adjusts the position of the robotic arm accordingly, resulting in the leftward movement shown in the third thyroid image. In the fourth thyroid image, the robotic arm is attached to the mid-neck region, which is not the correct location to detect the presence of the thyroid gland in the neck. The model guides the robotic arm to move to the left until it reaches the ideal position. In addition, even in the absence of thyroid presence in the fifth image, the model has learned to predict the ideal position and guides the robotic arm to stop at that position. The DQN learning model in FARUS demonstrates its ability to accurately guide the robotic arm during thyroid scanning, even in the presence of patient movement or the absence of the thyroid gland.

Experienced sonographers usually fine-tune the probe angle after locating the thyroid to obtain high-quality US images. To imitate such an expert behavior, an autonomous robotic system should assess the quality of US images and feed them back to adjust the probe orientation. However, taking into account the limited resolution of the Kinect and potential participant movements during the scan, the pre-estimated normal vector cannot be directly applied as the normal vector for subsequent scans. To tackle this problem, Bayesian optimization algorithm with image entropy as the loss function was used to obtain a better probe orientation with very few adjustments. Although some statistical methods such as grayscale, confidence map and root mean square error (RMSE) have been proposed, there is still no gold standard for evaluating US image quality[44]. In this work, we used the image entropy to evaluate US image quality because it is highly effective for image processing and can be used to assess texture in images based on a statistical measure of randomness[45].

In many instances, Bayesian optimization (BO) outperforms expert as well as other state-of-the-art global optimization algorithms. Bayesian optimization constructs a surrogate model for the objective function and quantifies the uncertainty through Bayesian inference. This surrogate model determines where the next candidate will be, as in Fig. 4a. The significant differences in image entropy values were observed after the BO phase for 89 participants (Fig. 4b). During the BO phase, the position of the US probe remains constant; therefore, probe orientation and contact force are the two major factors in the entropy value of the US image. We consider a limited budget of $N = 5$ iterations to speed up the BO phase. The entropy of the US image varies from 7.172 to 7.173 in 5 iterations, and the max entropy corresponding to the optimal orientation was reached at the second iteration (Fig. 4c, d). When the number of iterations reached 5, a higher drop in performance was observed due to its exploration nature. During this experiment, the probe was initially set at a relatively optimal angle, resulting in a less noticeable entropy increase before and after Bayesian optimization. However, between 21.3 and 24.8 s, the US probe underwent a 10-degree angle adjustment, revealing that the entropy value of the US image responded more sensitively to angle changes compared to image confidence. This sensitivity could be attributed to the level of contact between the skin and the probe. As illustrated in Fig. 4e, there is a positive relationship between the image entropy and confidence map[46] that evaluates the contact condition at each pixel. As the entropy value consumes less computation, it allows for a real-time control of image quality. To further explore the influence of contact force on image quality, we conducted an investigation

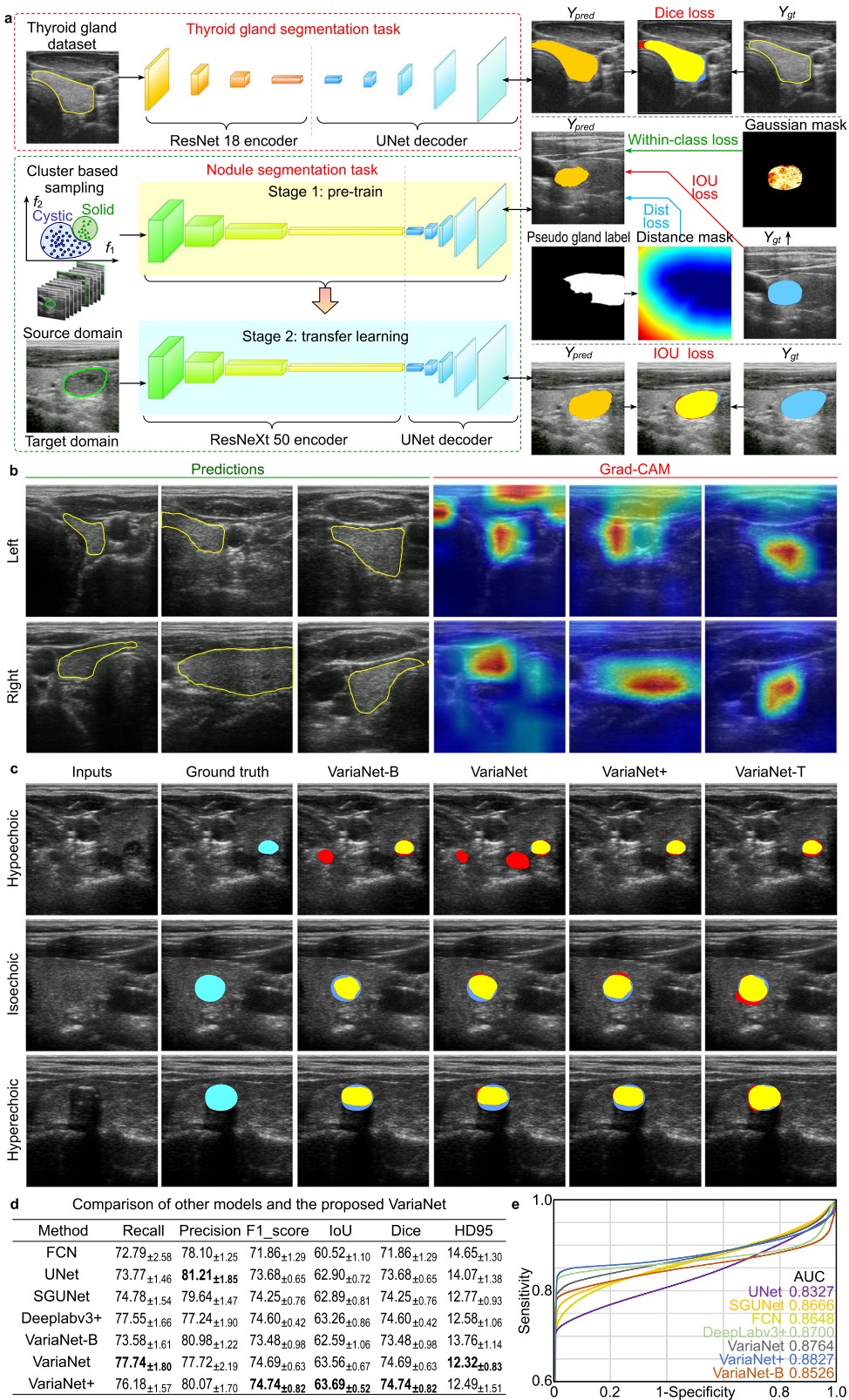

**Fig. 2 | Thyroid gland and nodule segmentation with deep learning networks.**
**a** Thyroid and nodule segmentation network based on pre-training and weakly supervision. The feature and spatial losses are used to provide prior knowledge to the network considering the diversity of nodule samples and the space constrain between thyroid lobes. **b** Thyroid lobe segmentation based on pre-training. **c** Our

proposed VariaNet and its variants predict results for different types of nodules.
**d** Comparisons with the existing segmentation models on the SCUTN1K testset (best result in bold). **e** Receiver Operating Characteristic (ROC) curve of different algorithms on the SCUTN1K testset. Source data are provided as a Source Data file.

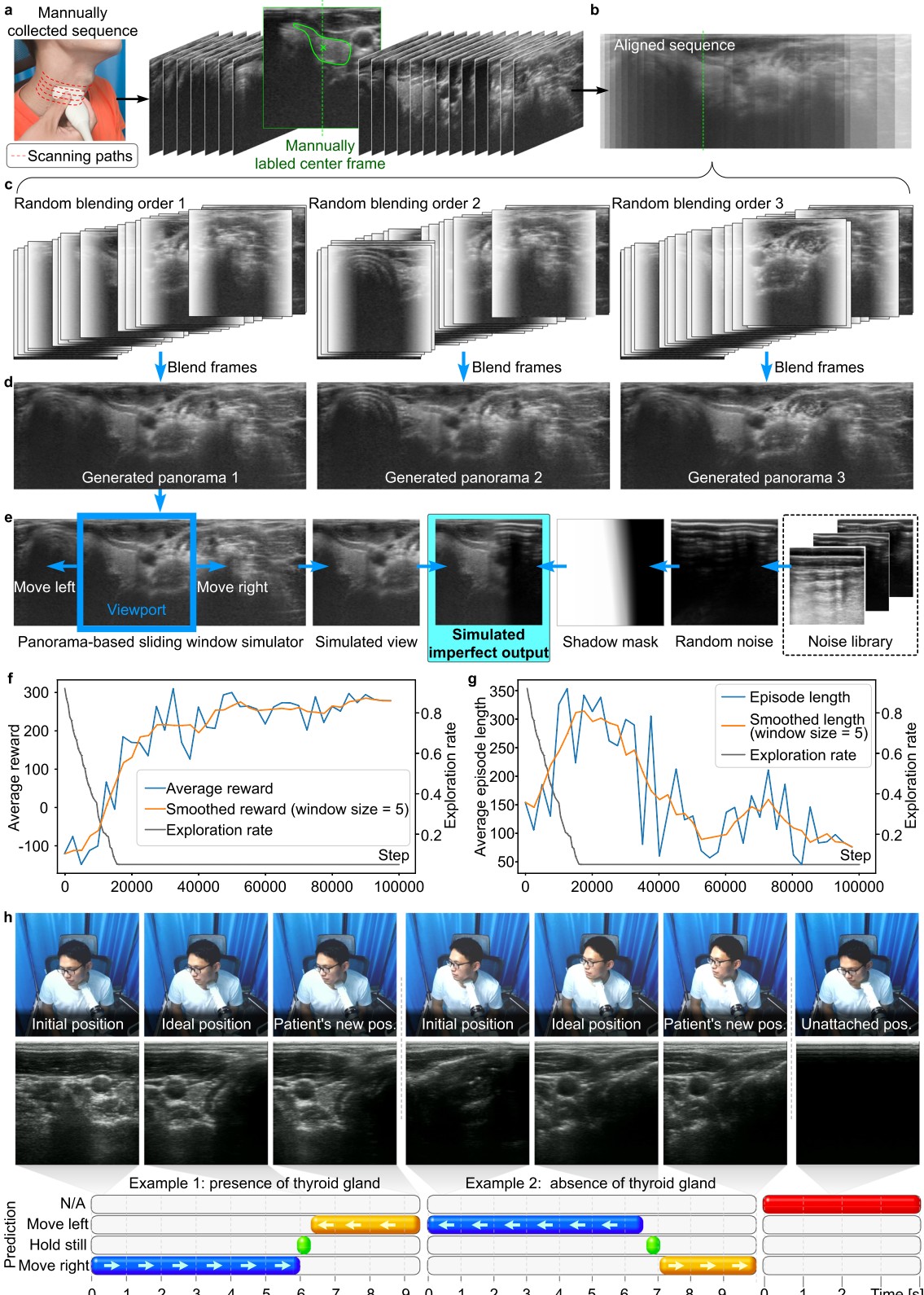

**Fig. 3 | Training process of DQN learning in panoramic thyroid environment.**
**a** A sequence of image is collected and labeled manually. **b** The alignment of
sequence image into panorama. **c** Blending images in a random sequence.
**d** Generated panorama. **e** The simulation of panorama-base sliding window.
**f** Average reward vs. step. **g** Average episode length vs. step. **h** Trained model's
prediction vs. time (please refer to Supplementary Movie 1 for additional details).
Source data are provided as a Source Data file.

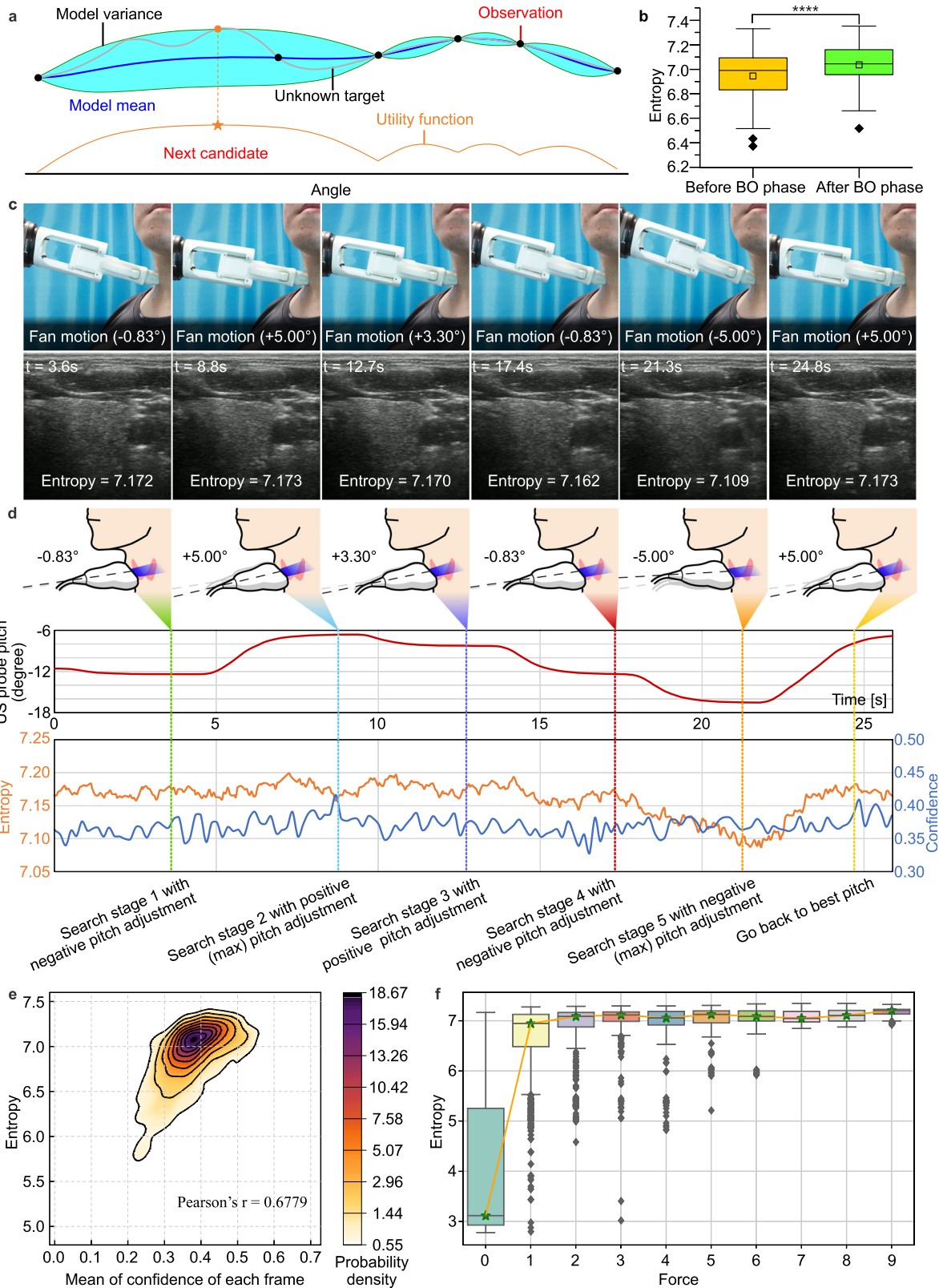

**Fig. 4 | Bayesian orientation (BO) optimization. a** Overview of Bayesian optimization. Example illustrating a Gaussian process surrogate model fitting to data derive from a unknown target and its expected utility function maximizing to select the next candidate. **b** The boxplot displaying the significant differences in image entropy values after BO phase for participants ($n = 89$); the top, middle, and bottom boundaries of the boxplots represent 25th, 50th, and 75th percentile, respectively; the small squares represent the mean. Outliers defined when value larger than 1.5*IQR + 75th percentile; ****$p < 0.0005$, $p$ value obtained with a paired, two-sided $t$-test. **c**, **d** Probe orientation optimization procedure shown the five decisions made by BO with image entropy as loss function (please refer to Supplementary Movie 2 for additional details). **e** The relationship between image entropy and mean confidence[46] that characterize the contact condition of US image, as seen in a positive correlation between two evaluation metrics. **f** Force values greater than 2 N do not cause major changes in the median entropy. The 13,256 pairs of data from 9 participants ($n = 9$) were divided into 10 groups and playback sampled 800 times for each group; the top, middle, and bottom boundaries of the boxplots represent 25th, 50th, and 75th percentile, respectively; the stars represent the median. Outliers defined when value larger than 1.5*IQR + 75th percentile. Source data are provided as a Source Data file.

involving nine participants. Each participant was instructed to apply varying levels of force on the US probe, positioned at the robot's end, while ensuring safety under continuous manual monitoring. Figure 4f demonstrates that when the contact force between the probe and the skin exceeds 2 N, the median of the image entropy value remains stable. Taking scanning comfort and safety into account, we set the maximum contact force at 4 N.

## Fully autonomous robotic ultrasound imaging

In actual clinical practice, sonographers frequently employ a combination of multi-view scanning methods to conduct comprehensive and detailed diagnosis of suspected nodules. Drawing inspiration from this clinical experience, once FARUS finds a suspected nodule in the transverse scan, the longitudinal scan will also be performed for further investigation. As displayed in Fig. 5a, we have achieved full autonomous scanning based on the fusion of force and visual information. Figure 5b shows the robotic scanning procedure for the right thyroid lobe of one participant. During the transverse scanning, we can see how the thyroid area increases first and then decreases, beginning with the appearance of the upper pole and ending with the disappearance of the lower pole. Moreover, shadows were avoided and thyroid lobes were centered with control strategy. The contact force, probe position and probe orientation are shown in Fig. 5c. As illustrated in Fig. 5c, noticeable fluctuations in the force value are observed during the initial 20 s of the transverse scan. This phenomenon arises as the US probe transitions from the middle of the thyroid toward the upper pole, following Bayesian optimization. The contact between the probe and the skin is affected by the surrounding thyroid cartilage at the upper pole, causing an unstable change in force during this movement. A similar instability is also observed during the scanning of the lower pole of the thyroid. Analysis of the probe's position change reveals that FARUS maintains a constant speed in the $Z$-axis direction throughout the transverse scanning process. However, non-uniform adjustments occur in the Y and X directions, which are related to centering the thyroid gland. In the longitudinal scanning process, if multiple nodules are detected, their image features and location information are combined to determine whether these nodules were scanned during the transverse scanning process. This determination is achieved by matching the position of the nodules within the thyroid.

For the evaluation of the US image quality, four evaluation metrics including confidence map[46], centering error, orientation error and image entropy were used to characterize the contact condition, thyroid visibility, orientation performance, and texture details of US image, respectively. From Fig. 5d–g, in the first 20 s, the entropy value and confidence value of the image increase as the probe makes contact with the patient, while the centering error and orientation error decrease. After 25 s, the mean value of the centering error of the image decreases to 0, indicating the completion of the thyroid search process and the image centering process. In the subsequent Bayesian optimization process, the entropy increase point is not distinctly visible due to variations in the optimization time for each patient. The centering error remain relatively stable between 25 and 100 s. After 150 s, the mean square error of confidence map, orientation error and image entropy increases, which is associated with the additional OPS phase.

## ACR TI-RADS risk stratification using FARUS

In this study, we present the FARUS system, designed for scanning, detection, and classification of nodules in a sample of 19 patients. The ACR TI-RADS[47] is employed for nodule classification based on their US characteristics. Additionally, we conduct a comparative analysis between FARUS-generated classifications and evaluations provided by professional sonographer. Figure 6a displays five nodules that demonstrate complete agreement with the sonographer's diagnosis. Results from the scoring and classification process by FARUS,

following ACR TI-RADS criteria, are in Fig. 6b. To assess the echogenicity, composition, and echogenic foci of the thyroid nodule, we analyze the distribution of pixels in the thyroid gland and the nodule (Fig. 6c). In Fig. 6b, the nodule's composition is categorized as solid, leading to a score of 2, while its echogenicity is classified as hypo, also resulting in a score of 2. The nodule demonstrates a well-defined boundary and a regular shape, contributing to a score of 0. Additionally, the comparison of height and weight exceeds 1, resulting in a score of 3. Lastly, no echogenic foci are observed within the nodule, leading to a score of 0. The total score amounts to 7, leading to the classification of the nodule as level 5 or highly suspicious. Another example of a nodule has a mixed composition, leading to a score of 1, and its echogenicity as hypo, resulting in a score of 2. The nodule's characteristic features, including a clear boundary and regular shape, warrant a score of 0 points. Additionally, the comparison of height and weight yields a score of 0 as it is less than 1. Furthermore, no presence of peripheral calcification within the nodule leading to a score of 0. Consequently, the cumulative score amounts to 3, classifying the nodule as level 3 or mildly suspicious.

According to doctor's evaluation, 17 individuals were found to have nodules, while 2 individuals showed no presence of nodules. Our developed system, FARUS, identified 13 individuals as having nodules and 6 individuals as having no nodules. The scoring and recommended management of 24 nodules among the 13 individuals detected by both FARUS and the doctor were compared. Each nodule detected by FARUS was matched to the doctor's report based on its location and shape. Table 1 shows the comparison between FARUS and doctor scoring and the classification of thyroid nodules (please refer to Supplementary Data 1 for more details). Among the 24 nodules assessed, 10 were diagnosed with the same score distribution by both the doctor and the developed FARUS. This can be attributed to the US image's appropriate brightness and contrast, resulting in well-distributed pixels. Additionally, 8 nodules exhibited a score difference of 1, primarily attributed to discrepancies in echogenicity or composition classifications. There were four nodules with a score difference of 2 and another with a score difference of 4, possibly caused by different classifications in both echogenicity and composition. Nodule size variations in the scan may be attributed to differences in patient positioning: the doctor scans in a supine position, while FARUS scans in an upright sitting position due to security concerns. The doctor's report indicates that out of the 19 patients, one patient has two nodules requiring fine-needle aspiration (FNA) and follow-up, respectively. For this patient, FARUS results were consistent with the doctor's assessment. In the cases of the nodule #5, diagnostic inconsistency arose from variations in echo characteristics, leading to divergent recommended management.

The main reason for the different classifications by FARUS is the use of a probe different from the one used by the doctor. The doctor's probe consistently produces a specific color of the thyroid gland for each patient, while the brightness and contrast of US images produced by our probe may vary across different patients with varying ages and weights. The proposed FARUS system classified nodules, with complete agreement in 10 out of 24 cases with the doctor's diagnosis. Furthermore, for the remaining nodules, the discrepancies were primarily limited to 1 score difference, mainly arising from variations in echogenicity and composition. Current volunteer participants typically have low-risk nodules and clinical FNA is not recommended. Although FARUS has demonstrated feasibility and potentiality in nodule detection and data collection for ACR TI-RADS classification, further clinical studies are essential to assess its safety as a screening tool for probably or definitely malignant nodules.

## Discussion

Current US examination relies on sonographers to perform scanning operations. Patients often need to make an appointment for US examinations, resulting in long waiting times and often delays in

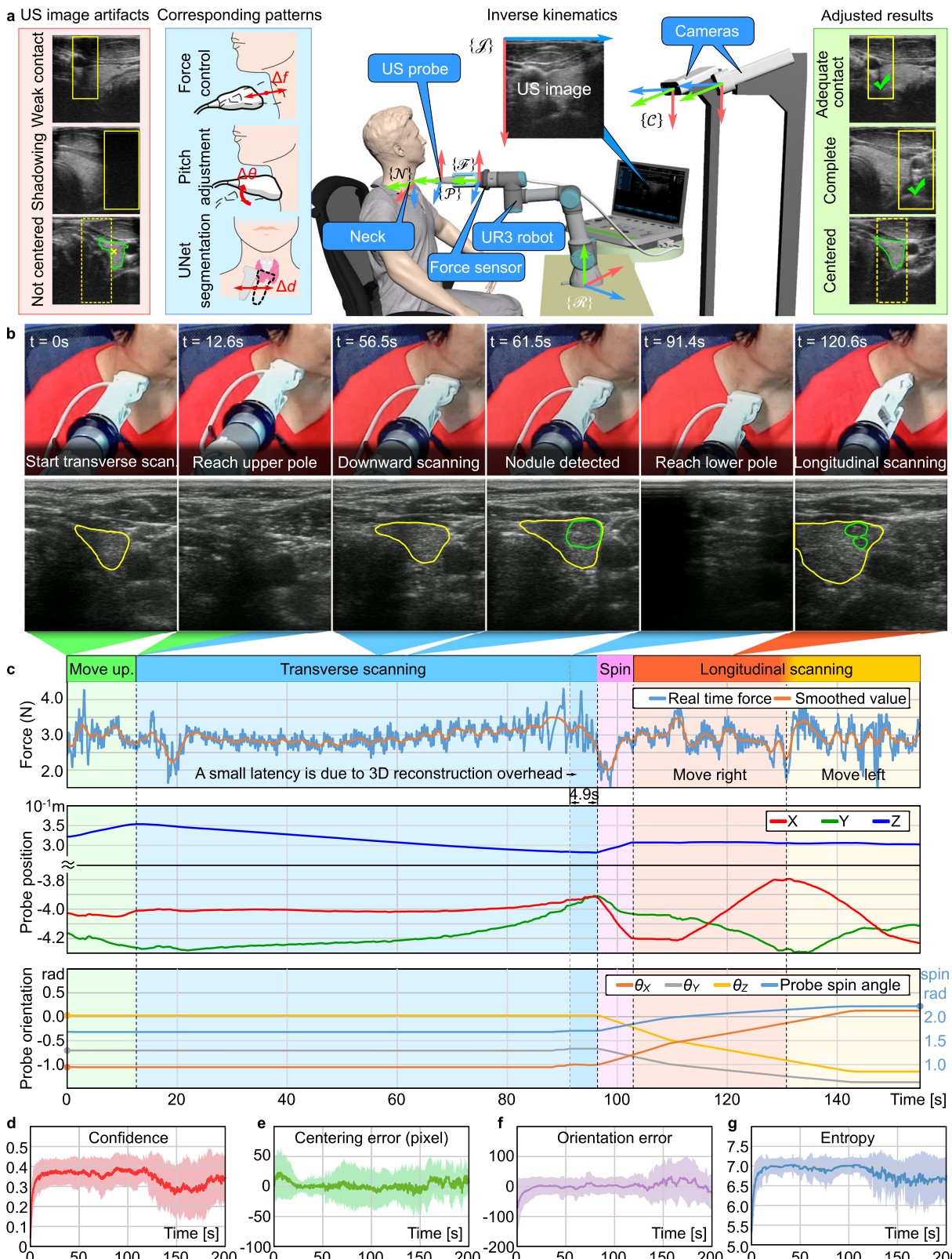

**Fig. 5 | Fully autonomous ultrasound imaging. a** The control strategy of FARUS. The autonomous scanning is achieved based on the fusion of force and visual information. **b, c** Evolution of force, probe position, and probe orientation for a patient with nodules during the experiment, which included the transverse scanning phase and longitudinal scanning phase. During the transverse scanning phase, the orientation of the probe at the end of the robot remained unchanged. Guided by the thyroid gland segmentation network, FARUS maintained the thyroid lobe in the center while moving the probe from the upper pole to the lower pole of the thyroid. When the nodule segmentation network detected a suspected nodule, the location of the suspected nodule was recorded. In the longitudinal phase, FARUS re-scanned the suspected nodules from a different angle to determine whether they were lesions (please refer to Supplementary Movie 3 for additional details). **d–g** Performance of the scanned image quality of FARUS on 19 patients. The performance metrics for evaluating the US image quality included confidence map, centering error, orientation error and image entropy. The shadowed area represents mean ± SD over the different experiments, while the curves inside the shadowed areas are the average results. Source data are provided as a Source Data file.

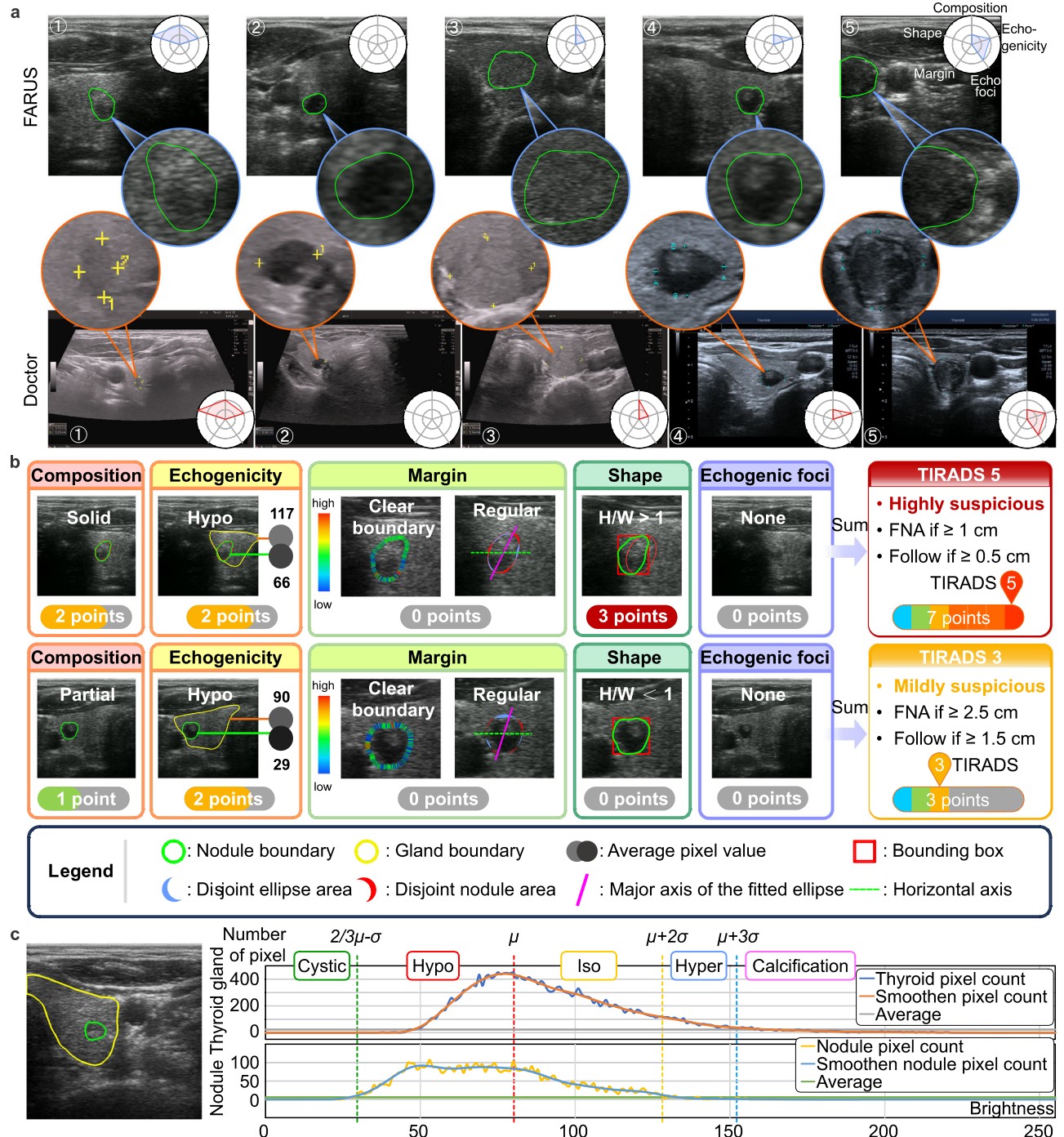

**Fig. 6 | Scoring and classification of thyroid nodules based on ACR-TIRADS.**
**a** Indicative nodules identified with FARUS and the respective US images provided by doctors. **b** Examples of two separate nodules, with accompanying explanations of their TIRADS scores. **c** Correlation between number of pixels over brightness in thyroid image. Source data are provided as a Source Data file.

treatment. At the same time, last decades have seen a rise in thyroid nodules. The detection and diagnosis of thyroid nodules still rely heavily on the expertise and experience of doctors. Compared with the traditional manual diagnosis or remote diagnosis, this fully autonomous robotic system adopts a patient-centered concept, allowing the patient to be examined in a more comfortable way. Furthermore, FARUS is suitable for rapid screening in out-patient clinics and remote low-level centers as it enables scan autonomously without intervention.

In this study, we developed an automated US diagnostic robot with artificial intelligence, which can accurately diagnose thyroid nodules. It is expected to be equipped not only in specialized hospitals,

but also in clinics and remote areas. This non-invasive, rapid, and accurate screening strategy can provide an early warning of thyroid nodule development. The system operates on an autonomous scanning mode, a notable advantage of which is the no need of direct contact between medical staff and patients. This configuration effectively minimizes the risk of transmitting infectious diseases between patients and healthcare providers.

Currently, most US robotic systems studies do not include comparisons with doctors, nor do they evaluate participants' satisfaction. As shown in Table S1, we recruited 14 sonographers from 7 hospitals to make evaluations of our acquired transverse and longitudinal images,

**Table 1 | Comparison of thyroid nodule scoring and recommendation between FARUS and doctor based on ACR TI-RADS**

| No | Name | Nodule | Size | Diagnosed by | Composition | Echogenicity | Margin | Shape | Echogenicity foci | Total | Difference | Recommendation |
|---|---|---|---|---|---|---|---|---|---|---|---|---|
| 1 | Patient 1 | #1 | 3.9 mm | FARUS | 2 | 2 | 0 | 3 | 0 | 7 | 0 | No FNA |
| | | | 3.2 mm | Doctor | 2 | 2 | 0 | 3 | 0 | 7 | | No FNA |
| 2 | Patient 2 | #2 | 3.0 mm | FARUS | 2 | 2 | 0 | 0 | 0 | 4 | 2 | No FNA |
| | | | 4.5 mm | Doctor | 1 | 1 | 0 | 0 | 0 | 2 | | No FNA |
| 3 | Patient 3 | #3 | 6.6 mm | FARUS | 0 | 0 | 0 | 0 | 0 | 0 | 0 | No FNA |
| | | | 6.9 mm | Doctor | 0 | 0 | 0 | 0 | 0 | 0 | | No FNA |
| 4 | Patient 3 | #4 | 5.9 mm | FARUS | 2 | 1 | 0 | 0 | 0 | 3 | 1 | No FNA |
| | | | 6.1 mm | Doctor | 1 | 1 | 0 | 0 | 0 | 2 | | No FNA |
| 5 | Patient 4 | #5 | 12.4 mm | FARUS | 2 | 2 | 0 | 0 | 0 | 4 | 1 | Follow-up |
| | | | 12.8 mm | Doctor | 2 | 1 | 0 | 0 | 0 | 3 | | No FNA |
| 6 | Patient 5 | #6 | 4.2 mm | FARUS | 2 | 2 | 0 | 0 | 0 | 4 | 4 | No FNA |
| | | | 3.4 mm | Doctor | 0 | 0 | 0 | 0 | 0 | 0 | | No FNA |
| 7 | Patient 5 | #7 | 6.9 mm | FARUS | 2 | 1 | 0 | 0 | 0 | 3 | 0 | No FNA |
| | | | 6.9 mm | Doctor | 1 | 2 | 0 | 0 | 0 | 3 | | No FNA |
| 8 | Patient 6 | #8 | 6.2 mm | FARUS | 1 | 2 | 0 | 0 | 0 | 3 | 1 | No FNA |
| | | | 5.5 mm | Doctor | 1 | 1 | 0 | 0 | 0 | 2 | | No FNA |
| 9 | Patient 7 | #9 | 6.7 mm | FARUS | 1 | 2 | 0 | 0 | 0 | 3 | 0 | No FNA |
| | | | 7.8 mm | Doctor | 1 | 2 | 0 | 0 | 0 | 3 | | No FNA |
| 10 | Patient 7 | #10 | 7.3 mm | FARUS | 2 | 2 | 0 | 0 | 0 | 4 | 1 | No FNA |
| | | | 7.2 mm | Doctor | 1 | 2 | 0 | 0 | 0 | 3 | | No FNA |
| 11 | Patient 7 | #11 | 6.6 mm | FARUS | 2 | 2 | 0 | 0 | 0 | 4 | 2 | No FNA |
| | | | 6.2 mm | Doctor | 1 | 1 | 0 | 0 | 0 | 2 | | No FNA |
| 12 | Patient 7 | #12 | 1.8 mm | FARUS | 2 | 2 | 0 | 0 | 0 | 4 | 2 | No FNA |
| | | | 2.4 mm | Doctor | 1 | 1 | 0 | 0 | 0 | 2 | | No FNA |
| 13 | Patient 7 | #13 | 3.1 mm | FARUS | 2 | 2 | 0 | 0 | 0 | 4 | 2 | No FNA |
| | | | 2.8 mm | Doctor | 1 | 1 | 0 | 0 | 0 | 2 | | No FNA |
| 14 | Patient 8 | #14 | 7.8 mm | FARUS | 1 | 2 | 0 | 0 | 0 | 3 | 1 | No FNA |
| | | | 7.6 mm | Doctor | 1 | 1 | 0 | 0 | 0 | 2 | | No FNA |
| 15 | Patient 8 | #15 | 4.3 mm | FARUS | 1 | 1 | 0 | 0 | 0 | 2 | 0 | No FNA |
| | | | 4.4 mm | Doctor | 1 | 1 | 0 | 0 | 0 | 2 | | No FNA |
| 16 | Patient 8 | #16 | 7.1 mm | FARUS | 1 | 1 | 0 | 0 | 0 | 2 | 0 | No FNA |
| | | | 7.5 mm | Doctor | 1 | 1 | 0 | 0 | 0 | 2 | | No FNA |
| 17 | Patient 8 | #17 | 1.7 mm | FARUS | 1 | 1 | 0 | 0 | 0 | 2 | 0 | No FNA |
| | | | 1.9 mm | Doctor | 1 | 1 | 0 | 0 | 0 | 2 | | No FNA |
| 18 | Patient 8 | #18 | 3.1 mm | FARUS | 1 | 1 | 0 | 0 | 0 | 2 | 0 | No FNA |
| | | | 4.1 mm | Doctor | 1 | 1 | 0 | 0 | 0 | 2 | | No FNA |
| 19 | Patient 9 | #19 | 18.6 mm | FARUS | 1 | 2 | 0 | 0 | 0 | 3 | 0 | Follow-up |
| | | | 19.1 mm | Doctor | 1 | 2 | 0 | 0 | 0 | 3 | | Follow-up |
| 20 | Patient 9 | #20 | 10.4 mm | FARUS | 1 | 2 | 0 | 0 | 2 | 5 | 0 | FNA |
| | | | 10.6 mm | Doctor | 1 | 2 | 0 | 0 | 2 | 5 | | FNA |
| 21 | Patient 10 | #21 | 5.8 mm | FARUS | 1 | 2 | 0 | 0 | 0 | 3 | 1 | No FNA |
| | | | 8.2 mm | Doctor | 1 | 1 | 0 | 0 | 0 | 2 | | No FNA |
| 22 | Patient 11 | #22 | 3.0 mm | FARUS | 1 | 2 | 0 | 0 | 0 | 3 | 1 | No FNA |
| | | | 3.6 mm | Doctor | 1 | 1 | 0 | 0 | 0 | 2 | | No FNA |
| 23 | Patient 12 | #23 | 17.0 mm | FARUS | 2 | 1 | 0 | 0 | 0 | 3 | 0 | No FNA |
| | | | 18.2 mm | Doctor | 2 | 1 | 0 | 0 | 0 | 3 | | No FNA |
| 24 | Patient 13 | #24 | 8.9 mm | FARUS | 2 | 1 | 0 | 0 | 0 | 3 | 1 | No FNA |
| | | | 9.7 mm | Doctor | 1 | 1 | 0 | 0 | 0 | 2 | | No FNA |

assigning scores quantified into five levels ranging from 1 to 5. Specifically, a score of 1 denoted "very poor", 2 denoted "poor", 3 denoted "medium", 4 denoted "good", and 5 denoted "very good" image quality. It shows that the images acquired by the FARUS have good quality, centrality and integrity. According to Fig. 7, most participants who took part in the scanning felt safe with our system and experienced no adverse reactions, such as pain and other discomfort. Certain participants expressed feelings of anxiety regarding the robot scan and harbored concerns regarding its safety. The majority of the participants expressed that US robots cannot replace doctors, primarily due to doctors' possession of a vast repository of medical knowledge, which the robots lack at the current moment. Moreover, doctors occupy a pivotal position as esteemed medical experts and trustworthy sources of aid.

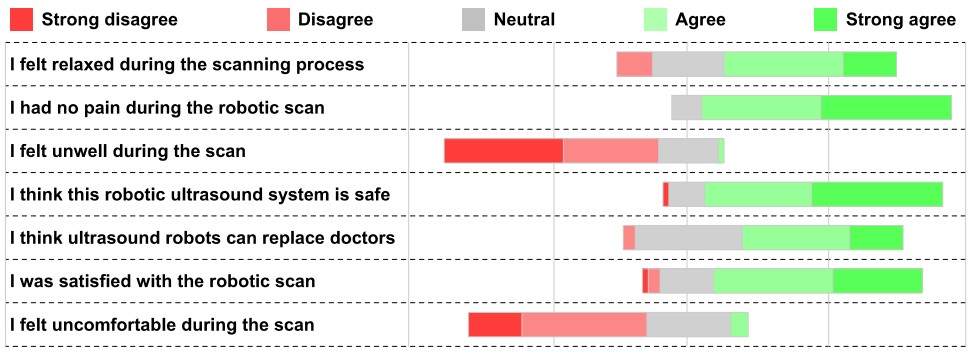

**Fig. 7 | Subjective Evaluation of FARUS Scanning Through Questionnaire Assessment.** A clear trend of agreement or disagreement is more obvious when the entire bar (100%) is shifted left or right ($n = 70$). Source data are provided as a Source Data file.

To compare the image quality of robotic US scans with that of manual scans, we invited five doctors with between 3 and 15 years of experience to scan the thyroids of 13 participants. In Fig. 8a, the FARUS completed transverse and longitudinal scanning in the same manner as a doctor. In the OPS phase, FARUS can detect suspected nodules as well as cover the carotid arteries. During the IPS phase, the FARUS was able to scan a continuous area from the upper pole to the lower pole of the thyroid gland, while ensuring that the thyroid gland was centered. As shown in Fig. 8b–e, there is still variation among doctors when using entropy, center error, mean confidence, and left-right intensity symmetry (LRIS) as image evaluation metrics. Specifically, LRIS refers to the ratio of gray scale distribution between the left and right sides of US images. Therefore, we performed a comparative analysis of five doctors as a group with robot performance. Figure 8f–i shows some similarity in the evaluation metrics between the robot and the doctors for both the IPS and OPS phase. The centering error of the IPS phase of robotic US scans is smaller than that observed in scans conducted by five doctors. This difference can be attributed to the robot control process. Additionally, differences in scanning method and range may also contribute to differences in entropy value, mean confidence, and image grayscale distribution between robot and doctors.

We also compared the probe motion of the robotic scanning with that of manual scanning. The probe motion data during doctors' scanning process were recorded with a probe motion sensor. As compared to manual scanning, the robotic scanning was more stable in terms of force and velocity, as in Table S2. In general, humans were more efficient than FARUS. The FARUS took $213.0 \pm 85.3$ s to complete a single thyroid lobe scan for 70 participants, while five doctors spent $67.2 \pm 27.6$ s to complete the scan for 13 participants. These differences can be attributed to dynamic path planning and force-controlled feedback, the former to compensate the motion and the latter to ensure participants' comfort. Moreover, a conservative speed is implemented in our FARUS system to ensure the safety of participants.

The FARUS comprises three primary stages: scanning, detection, and classification, each of which plays a crucial role in influencing FARUS's performance. During the detection stage, the main challenge faced by FARUS in detecting nodules is primarily attributed to the size and echogenicity of the nodules. Table 2 shows missed and possible false positive thyroid nodules detected by FARUS. The results of this study indicate that smaller nodules, such as #25, #26, #27, #28, #29 and #30 present greater difficulty for FARUS to detect. Similarly, nodules with isogenic properties, such as #31 and #32, also pose challenges for FARUS's detection capabilities. Moreover, the FARUS identified some nodules for which the doctor did not (#33, #34, #35 and #36). We sought opinions from multiple doctors and a firm conclusion could not be reached by doctors. In an exercise of prudence, we considered these occurrences as possibly false positives.

The FARUS needs improvement in the future due to its limitations, especially for small-scale and low-contrast nodules. The existing nodule dataset lacks sufficient diversity in terms of nodule size. Artifacts in ultrasound images were not considered in our current algorithm. Additionally, work needs to be done to incorporate video streams.

## Methods

### Human participants and safety

All experiments with human participants were performed with the approval of the Guangzhou First People's Hospital Review Board (K-2021-131-04). Our researchers explained the entire process to all participants, and all participants signed an informed consent form. Furthermore, we obtained consent from participants confirming their understanding of the open access nature of this journal.

With the approval of the ethical review committee, we recruited of three distinct groups of participants. The participants were over 18 years old, and of both sexes. None of the participants had following cases: (1) neck trauma or failure to heal after surgery; (2) inability to maintain a stable head position;(3) history of US gel allergies; (4) history of surgical resection of both thyroid lobes. The first group predominantly comprised college students, and we manually collected thyroid US data from 66 volunteers within this group using handheld US equipment. Simultaneously, we employed FARUS system to autonomously scan 70 volunteers (20–30 years of age, 19 females, 41 males), 13 of whom were scanned manually by 5 doctors. To address the limitations of handheld US equipment in accurately diagnosing nodules, we opted to employ portable US equipment to gather two additional sets of data. The second set of data was obtained from thyroid US scans of 29 middle-aged and elderly individuals within the community, chosen specifically to facilitate the training of the nodule segmentation network. Finally, the third group was composed of 19 patients (age $53.05 \pm 5.90$ years old, 12 females, 7 males) who underwent robotic thyroid scanning. This group played a pivotal role in verifying the diagnostic performance of our FARUS system. The recruitment bias primarily arises from the geographic scope of recruitment, limited to a specific region in China.

To ensure the safety of the participants, these five approaches were implemented in the FARUS system: (1) All the participants needed to complete an US gel allergy test before thyroid scanning. (2) The FARUS system used the UR3 collaborative robot, which will stop by itself in the event of a collision and a safety staff has been monitoring the operation of the robot arm. (3) We limited the working area of the robot arm ($R < 45$ cm), and once it crosses the working area, it will be forcibly stopped. (4) We set the contact force between 2.0 N and 4.0 N to ensure adequate contact with the skin and to prevent pressure-induced distortion of the thyroid anatomy. If the force exceeds 4.5 N, the robot will be automatically stopped. 5. The chair that participants used during scanning had wheels so that participants could move back if they felt uncomfortable.

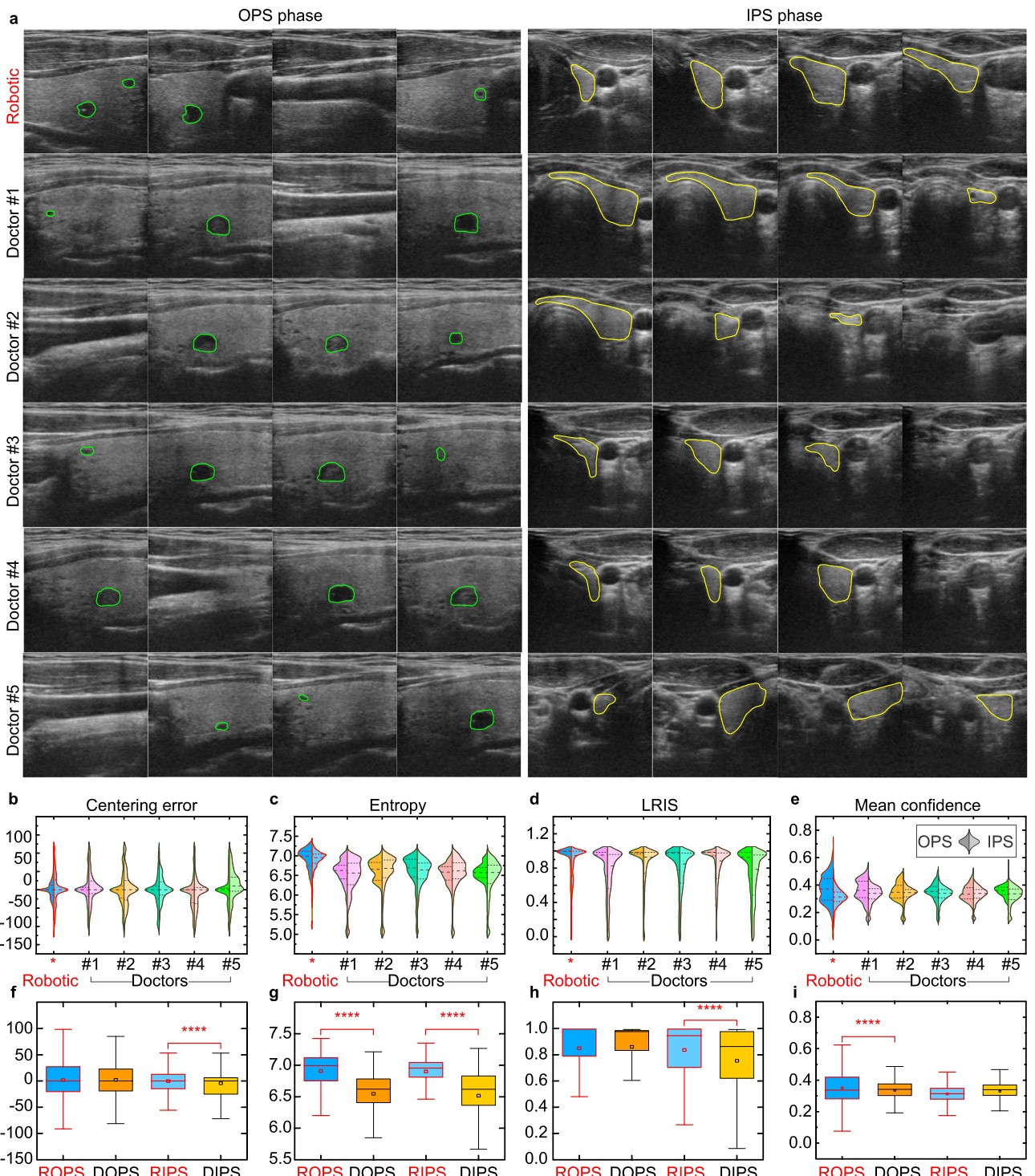

**Fig. 8 | Evaluation of US scans collected by the FARUS and five doctors. a** The in-plane/out-of-plane US image sequences acquired by the FARUS and five experi-enced doctors on the same two participants. The green contour in the US images represent the segmented suspicious nodules by our proposed VariaNet, while the yellow contour represent the segmented thyroid lobe by our pre-trained gland segmentation model. **b–e** Violin plots illustrating the four evaluation metrics between FARUS and five doctors. Dashed line indicates median; dotted lines indi-cate 25th and 75th percentiles; $n = 13$ participants. **f–i** Boxplots displaying the four evaluation metrics between FARUS and five doctors. ROPS robotic out-of-plane scanning, DOPS doctor out-of-plane scanning, RIPS robotic in-plane scanning, DIPS doctor in-plane scanning, LRIS left–right intensity symmetry. The top, middle, and bottom boundaries of the boxplots represent 25th, 50th, and 75th percentile, respectively; the small squares represent the mean. Outliers defined when value larger than 1.5*IQR + 75th percentile; $n = 13$ participants; $p$ value obtained with bootstrap tests; ****$p < 0.0005$. Source data are provided as a Source Data file.

## Thyroid modeling

For the estimation of the thyroid scanning range, we obtained the head and neck CT images from the Southern Hospital of Southern Medical University. Those contrast-enhanced CT images were used to establish the 3D model of the thyroid. We selected 500 contrast-enhanced CT sequences from patients with thickness ranging from 0.625 mm to 0.9 mm for modeling. This included 250 male and 250 female patients, aged between 7 and 82. The CT images were acquired with Philips

**Table 2 | Missed and possible false positives thyroid nodules detected by FARUS**

| No | Name | Nodule | Size | Diagnosed by | Composition | Echogenicity | Margin | Shape | Echogenicity foci | Total | Recommendation |
|----|------|--------|------|-------------|-------------|--------------|--------|-------|-------------------|-------|----------------|
| 1 | Patient 14 | #25 | 4 mm | Doctor | 2 | 1 | 0 | 0 | 0 | 3 | No FNA |
| 2 | Patient 14 | #26 | 3.7 mm | Doctor | 2 | 2 | 0 | 0 | 0 | 4 | No FNA |
| 3 | Patient 15 | #27 | 2 mm | Doctor | 1 | 2 | 0 | 0 | 0 | 3 | No FNA |
| 4 | Patient 9 | #28 | 3 mm | Doctor | 1 | 1 | 0 | 0 | 0 | 2 | No FNA |
| 5 | Patient 13 | #29 | 3 mm | Doctor | 0 | 0 | 0 | 0 | 0 | 0 | No FNA |
| 6 | Patient 16 | #30 | 2.3 mm | Doctor | 0 | 0 | 0 | 0 | 0 | 0 | No FNA |
| 7 | Patient 17 | #31 | 4.7 mm | Doctor | 1 | 1 | 0 | 0 | 0 | 2 | No FNA |
| 8 | Patient 8 | #32 | 23 mm | Doctor | 2 | 1 | 0 | 0 | 0 | 3 | No FNA |
| 9 | Patient 14 | #33 | 4.5 mm | FARUS | 2 | 1 | 0 | 0 | 0 | 3 | No FNA |
| 10 | Patient 14 | #34 | 7.0 mm | FARUS | 2 | 2 | 0 | 0 | 0 | 4 | No FNA |
| 11 | Patient 2 | #35 | 5.5 mm | FARUS | 2 | 1 | 0 | 0 | 0 | 3 | No FNA |
| 12 | Patient 2 | #36 | 11.0 mm | FARUS | 1 | 2 | 0 | 3 | 0 | 6 | Follow-up |

Gemini TF PET/CT scanner, and the resolution of the acquired DICOM data is $512 \times 512$. We used samples that represented all age group populations and covered people with body mass index (BMI) ranging from 17.1 to 27.8.

We defined 60 marker points to label the thyroid gland (see Fig. S1). They were 2 points above and below the isthmus of the thyroid; 8 points in the sagittal plane of the left and right lobes of the thyroid; and 2 points on the left and right of the entire thyroid cross-section; for the left and right lobes of the thyroid above the isthmus. The height is divided into five equal parts, and 4 points are marked on the upper, lower, left, and right sides of the 4 cross-sections, for a total of 32 points; After manually marking all the key points, our custom Python program will generate a 3D model and measure the average measurements of the various parts. Through this model, we obtained the morphological data of the thyroid. The average height of thyroid gland was 4.76 (±0.57) cm, and the average unilateral leaf width and thickness was 1.48 (±0.34) cm and 1.52 (±0.22) cm, respectively. Through regression analysis, we found that the width, length, thickness, and volume of the thyroid gland had very low correlations with gender, age, height, and weight. For this reason, the scan length was set to 6.47 cm based on the $3\sigma$ rule, and the scan width was set to 5.48 cm that was the probe width of 4.0 cm plus the average thyroid width.

**Thyroid gland and nodule segmentation**

We introduce a novel architecture called VariaNet, specifically designed for thyroid and nodule segmentation, as illustrated in Fig. 2a. The VariaNet architecture consists of two main models: the first model is responsible for thyroid gland segmentation, utilizing ResNet18[48] as the encoder and UNet as the decoder. Our training dataset, SCUTG8K, is employed for training this model, using the dice loss as loss function. The second model, focused to nodule segmentation, uses ResNeXt 50 as the encoder and UNet as the decoder, integrating a custom loss function. Due to dataset variations from different US probes, the training process for thyroid nodule segmentation is divided into two stages. Initially, we use TN3K[49] and our dataset, SCUTN10K, as the thyroid nodule image dataset for training the model. The second phase involves the application of transfer learning. During this phase, we use our dataset, SCUTN1K, which is generated using our specific probe, to train the thyroid nodule model.

To improve the segmentation performance of solid nodules, we introduce two tailor-made loss functions. Firstly, the feature loss function assigns higher weights to the isogenic part within the nodules' area. This attention mechanism, represented by the Gaussian mask of the image, allows VariaNet to focus on the isogenic region within the nodule, consequently enhancing the segmentation results. The definition of the feature loss is as follows:

$$L_{feat} = 1 - \frac{\widehat{pred} \cap gt}{\widehat{pred} \cup gt}$$
$$\widehat{pred} = W^{Gauss} \cdot N_{pred} \tag{1}$$
$$W^{Gauss} = 1 - e^{-\frac{(gt - g_{avg})^2}{2g_{std}^2}}$$

where $N_{pred}$ represent the prediction of the nodule mask, and $gt$ denote the ground truth of the nodule mask. The variables $g_{avg}$ and $g_{std}$ correspond to the mean and standard value, respectively, of the gland lobe intensity derived from the pseudo label, as illustrated in Fig. 2a. The second loss function used is called distance loss. The primary objective of this loss function is to assign increased weights to the false positives present in the model's prediction mask. The Distance Loss is constructed based on a distance mask that is calculated using the distances to the pseudo gland label. It is defined as:

$$L_{dist} = W^{dist} \cdot G_{pred} \tag{2}$$

$$W^{dist} = \begin{cases} 0 & p \in G_{pred} \\ min(x_b, y_b) \in Boundary \sqrt{(x_p - x_b)^2 + (y_p - y_b)^2} & p \notin G_{pred} \end{cases} \tag{3}$$

where $G_{pred}$ represent the prediction of the gland mask, and $(x_b, y_b)$ denote the boundary of the gland mask. The fundamental insight underlying this loss function is that the thyroid nodule mask must be spatially confined within or *precisely* aligned with the boundaries of the thyroid gland mask. Figure 2a shows feature loss presented by the gaussian mask, IoU loss and Distance Loss presented by Distance mask. By combining feature loss, distance loss and IoU loss, we develop iso-hybrid loss for segmentation in the three-level aspect echogenicity-, position- and map-level, which able to capture three aspects of segmentation. The iso-hybrid loss is defined as:

$$L_{IsoHy} = L_{iou} + \alpha L_{feat} + \beta L_{dist} \tag{4}$$

The framework is implemented using PyTorch 1.10.0 with CUDA 11.3 support. When the ImageNet pre-trained encoder is accessible, it is employed to initialize the model's weights. During training, a batch size of 8 is utilized, and the ADAM optimizer is employed to optimize the model for a total of 40 epochs.

## Scan planning

We used the Microsoft Azure Kinect DK depth camera to produce an initial scanning plan. We set up a Kinect camera at a distance of about 1.2 m from the seat, a height of about 1.5 m, and a 45° angle to the front of the seat. The camera acquisition parameters were set as: the color camera was set to a resolution of 1280 × 720, and the field of view was 90° × 59°; the depth camera was set to WFOV 2 × 2 binned, the resolution was 512 × 512, and the field of view was 120° × 120°; The frame rate was set to 30 FPS. We estimated the thyroid location using the neck and head skeleton joint points. Given variations in thyroid anatomy among populations, we introduced a process to locate the thyroid lobe. This involved utilizing ultrasound guidance and image segmentation for precise determination of its position.

The robotic scanning involved switching between IPS and OPS phases, as mentioned in the "System design for autonomous ultrasound imaging" section. The IPS phase started with an optimized initial orientation achieved through Bayesian optimization. Inspired by the IPS procedure introduced in ref. 34, the probe then scanned upward and downward until both ends of the thyroid lobe were no longer visible in the segmentation image. During the IPS phase, VariaNet recorded suspicious nodule locations, guiding the subsequent OPS phase. To simulate doctors' multi-view scanning, the OPS phase begins with a 60° probe rotation, followed by movement along the principal axis, covering a range of 12 mm. The image-based probe adjustment was achieved by calculating the intensity distribution of the US image as well as the center of mass in the segmentation mask, which enabled to prevent shadowing and keep the thyroid lobe in the center of the image.

## Robotic scanning

The platform consisted of a UR3 robot that was a six-degree-of-freedom robotic arm with a repeatability of 0.1 mm, a working radius of 500 mm, and a maximum load of 3 kg. We used the real-time port for monitoring, which fed back the status of the robot arm at a speed of 125 Hz. For general robotic arm commands, such as motion commands, we sent them through the secondary port, which received commands at 10 Hz. Moreover, we used SolidWorks to design a fixture for the probe, which was 3D printed with photosensitive resin. To avoid excessive pressure on the participants, we attached the US probe to the robot flange with four flexible spring-loaded connection (Fig. S2). And the length of the entire robot end effector was 242 mm. We used a SonoStar C5 Laptop Color Doppler Ultrasound System. This US depth was between 18 mm and 184 mm, the US frequency range was 6.5 MHz to 10 MHz. The original US data had a resolution of 512 × 512 and were stored in the AVI format.

Prior to scanning, participants assumed a seated position on a chair and adjusted their posture. Subsequently, the Kinect camera captured a depth map, which was then employed to generate a 3D point cloud of the environment. This point cloud underwent analysis for real-time detection and tracking of the human body joints by the Kinect body-tracking algorithm[50]. Then the position of the thyroid was estimated based on spatial information derived from tracked human skeleton joints. The robot approaches the pre-estimated position at a speed of 10 mm/s. When the distance between the probe and participant is approximately less than 200 mm, the speed of robot end effector will be reduced to 5 mm/s. The FARUS started scanning procedure when the probe reaches the pre-determined position and the force is in the range of 2.0 N to 4.0 N. During the scanning process, the participant was allowed to move slightly and the robot was able to adapt to the participant's movement through the force control and image servo. However, the scanning will be terminated if FARUS detects that the participant moves more than 2 cm left or right within 1 s, or 5 cm forward or backward. We chose the Robotiq FT300-S as the force/torque sensor. Its maximum range was $\mathbf{F}_x = \mathbf{F}_y = \mathbf{F}_z = \pm 300N$, and the sensor signal noise was 0.1 N. The control terminal read force information in real time at a speed of 100 Hz.

## Control strategy

During robotic scanning, the control strategy we developed allows us to adjust the position and orientation of the probe autonomously. The US probe has a total of six degrees of freedom that can be adjusted. As illustrated in Fig. 5a, force control can be used to control one degree of freedom: the amount of translation of the robot in the $\mathbf{Y}_p$ direction. The degree of freedom of US probe rotation around the $\mathbf{Y}_p$ axis is controlled by switching between transverse and longitudinal views. Bayesian optimization is applied to control the rotational degrees of freedom of the probe around the $\mathbf{Z}_p$ axis. The remaining three degrees of freedom are determined by the US image. Degrees of freedom in the $\mathbf{X}_p$ axis and $\mathbf{Z}_p$ axis translation of the US probe are controlled by the image segmentation results. While the degree of freedom of US probe rotation around the $\mathbf{X}_p$ axis is controlled by image intensity distribution.

Keeping the contact force applied to the body under control is vital to the safety of the participants. The external forces $\boldsymbol{f}$ and torques $\boldsymbol{\tau}$ measured in the force sensor frame $\mathcal{F}$ is a 6-D vector $\mathbf{H} = (\boldsymbol{f}^T \boldsymbol{\tau}^T)$. The force/torque vector in the probe frame $\mathcal{P}$ can be written as:

$$^p\mathbf{H}_p = {}^p\mathbf{F}_f(\mathbf{H} - {}^f\mathbf{F}_g {}^g\mathbf{H}_g) \tag{5}$$

where $^g\mathbf{H}_g \in \mathbb{R}^6$ is the gravity force of the US probe, and represent the transformation matrix from frame $\mathcal{F}$ to frame $\mathcal{P}$ and the transformation matrix from the probe's inertial frame to frame $\mathcal{F}$, respectively. Our objective is to control the force along the y-axis in the probe frame $\mathcal{P}$, we define the target contact force as:

$$t_f = \mathbf{S}_y {}^p\mathbf{H}_p \tag{6}$$

where $\mathbf{S}_y = (0\ 1\ 0\ 0\ 0\ 0)$. According to the visual servo control law $\dot{s} = \boldsymbol{L}_s \boldsymbol{v}_c$[51], the control law for the force control task is given by:

$$v_f = -\frac{\lambda_f}{k}(\boldsymbol{S}_y {}^p\mathbf{F}_f(\mathbf{H} - {}^f\mathbf{F}_g {}^g\mathbf{H}_g) - t_f^*)\boldsymbol{S}_y^T \tag{7}$$

where $\lambda_f$ is the force control gain, $t_f^*$ is the desired force value and $k$ is the human tissues stiffness that is usually set in [125, 500] N/m according to ref. 52.

The sweeping trajectory $T$ is defined by $N$ waypoints $\{w_0, w_1, \ldots, w_n\}$ and a speed $\mathbf{v}$ to travel along the trajectory. The waypoint $w_0$ is defined as the position where the probe orientation is optimized after Bayesian optimization procedure. In the case of scanning toward the upper or lower poles of the thyroid, the next waypoint $w_i$ would be a stride from the previous waypoint $w_{i-1}$. The position and orientation of the probe is adjusted after each step of movement by: (1) a translation along the $\mathbf{Y}_p$ axis to achieve sufficient contact between the probe and the neck. (2) a translation along the $\mathbf{Z}_p$ axis to ensure that the segmented thyroid is located in the center of US image. (3) A sideway correction to avoid the shadowing. Similar to ref. 34, the image-based sideway correction is carried out in an iterative manner:

$$\mathbf{R}_{step} = \begin{bmatrix} 1 & 0 & 0 \\ 0 & cos(\theta_{step}) & -sin(\theta_{step}) \\ 0 & sin(\theta_{step}) & cos(\theta_{step}) \end{bmatrix} \tag{8}$$

$$\mathbf{R}_n = \mathbf{R}_{step} \cdot \mathbf{R}_{n-1} \tag{9}$$

where $\mathbf{R}_{step}$ is the rotation matrix defined by the rotation adjustment step of $\theta_{step}$ in the $\mathbf{Y}_p$-$\mathbf{Z}_p$ plane, $\mathbf{R}_n$ represents the probe orientation after $n$ adjustments.

## Thyroid nodules scoring and classification

The ACR TI-RADS is a risk stratification system designed to help radiologists categorize thyroid nodules based on their US characteristics[47].

It provides a standardized way to assess the risk of malignancy for thyroid nodules, which can then guide the management and follow-up recommendations for patients. Figure 6b illustrates a sample outcome obtained from the ACR-TIRADS calculation:

(1) Echogenicity, composition and echogenic foci: to assess the echogenicity, composition, and echogenic foci of the thyroid nodule, we analyze the distribution of pixels in the thyroid gland and the nodule, as depicted in Fig. 6c. Initially, we create a smooth line graph representing the pixel distribution in the thyroid gland and calculate the line average of the number of pixels. By selecting thyroid gland pixels above this line average, we determine the mean and standard deviation. Using the same method, we identify thyroid nodule pixels and calculate the percentage of echogenicity, composition, and presence of echogenic foci based on specific boundaries, such as cystic, hypo-genicity, iso-genicity, hyper-genicity, and calcification.

(2) Margin and shape: the determination of the thyroid nodule's margin involves two factors. Firstly, we consider the boundary line of the nodule, assessing distinctive pixels on its outer and inner edges. This factor is calculated by determining the mean standard deviation of 10 perpendicular pixels along each pixel of the nodule's boundary line (5 inner pixels + 1 mid pixel + 5 outer pixels). Secondly, we analyze the shape of the boundary line by calculating the intersection over union (IoU) between the nodule and the fitted ellipse. The aspect ratio is another important shape characteristic, indicating the ratio of depth to width and revealing whether the nodule appears taller-than-wide. This aspect ratio provides valuable insight into the tumor's growth pattern and is associated with an elevated risk of malignancy if it is higher.

## Statistics and reproducibility
No statistical method was used to predetermine sample size. To assess the robotic image quality of US scans, we employed the FARUS system to autonomously scan 70 college students ($n = 70$). Additionally, to investigate the diagnostic performance of our FARUS system, robotic thyroid scanning was performed on 19 patients ($n = 19$). Overall, the robotic thyroid scans were successfully performed on 89 participants. Additionally, a transparent reporting of a multivariable prediction model for individual prognosis or diagnosis (TRIPOD) checklist is provided as Supplementary Table S3.

## Reporting summary
Further information on research design is available in the Nature Portfolio Reporting Summary linked to this article.

## Data availability
The main data generated in this study have been deposited in the github: https://github.com/Ciel04sk/SCUT_Thyroid_DataSet/tree/main[53]. The graph and table data within this work are provided in the Supplementary Information/Source Data file. Source data are provided with this paper.

## Code availability
The code for this study have been deposited in the github: https://github.com/Ciel04sk/SCUT_Thyroid_DataSet/tree/main[53].

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

## Acknowledgements

This project was funded by National Key Research and Development Program of China (No. 2023YFC3603803), TCL Charity Foundation Young Scholars Project (No. 2024001), Guangdong Basic and Applied Basic Research Foundation (2023B1515120064), Dreams Foundation of Jianghuai Advance Technology Center (No. 2023ZM01Z013) and Guangdong Basic and Applied Basic Research Foundation (2022A1515010140, 2023A1515011248). We extend our special appreciation to Chelvyn Christson Immanuel for his insightful and beneficial suggestion regarding image segmentation.

## Author contributions

K. Su, J. Liu, X. Ren, Y. Huo and G. Du contributed to the conception of the study; K. Su, J. Liu and W. Zhao performed the experiment; K. Su, Y. Huo, G. Du and P. X. Liu contributed significantly to analysis and manuscript preparation; K. Su, Y. Huo and P. X. Liu performed the data analyses and wrote the manuscript; X. Wang, B. Liang, D. Li and P. X. Liu helped perform the analysis with constructive discussions.

## Competing interests

The authors declare no competing interests.
