## [Peer Review File · Nature Communications]

A Fully Autonomous Robotic Ultrasound System for Thyroid ScanningREVIEWER COMMENTS

Reviewer #1 (Remarks to the Author):

This paper proposes a novel robotic ultrasound scanning system for thyroid nodule detection. The presented approach functions fully autonomously. The method combines tasks such as an automatic approach to the neck, US image-based navigation, thyroid nodule detection, and visualization of the nodules from two views, in-plane, and out-of-plane. The system functions successfully and shows superior results compared to manual nodule detection performed by different doctors. Furthermore, the authors report that the volunteers felt comfortable with the system.

Major correction in referencing prior work:

The current embedding of this manuscript in the scientific landscape is misleading. Most of the in-plane navigation cannot be seen as a contribution by the authors but was already introduced by Zielke et al. (RSV: robotic sonography for thyroid volumetry, IEEE RA-L 2022). The authors mention this paper in the related work section, showing that they are aware of this publication. However, this fact is not presented correctly in the methodology and contribution. Therefore, the authors must ensure that their contributions are correctly embedded into the scientific landscape, and that previous work is acknowledged sufficiently in an updated manuscript version and wherever it is used or is related to, to make the contributions and novelty of this work clearer.

Validity:

The proposed method was evaluated against the performance of several doctors and through participant feedback. Both evaluations show promising results. The conclusions that were drawn are also coherent. Both doctor and volunteer groups were sufficiently large for the evaluation. In the future, testing the system on actual patients would be interesting, and discussing this limitation in the paper would be helpful. However, this is not a reason to reject the current paper.

Significance:

The paper proposes an entire autonomous robotic ultrasound scanning framework for thyroid nodule detection. A major concern is that the in-plane automatic navigation is the same method proposed by Zielke et al. (RSV: robotic sonography for thyroid volumetry, IEEE RA-L 2022). This image-based navigation is, therefore, not novel and cannot be seen as a contribution. It is also of concern that the authors did not reference this existing work appropriately. The authors should update this reference. However, on top of the existing work, the authors combine previous literature with a key frame-based automatic approach, nodule detection, and an out-of-plane rotation for multi-view analysis of the nodule. Therefore, the contributions are still relevant for the community but have to be updated based on the existing state of the art.

Data and Methodology:

The quality and quantity of the data are well presented. Furthermore, the gathered data is presented nicely. Based on the comments in the Significance section, the presentation of the methodology has to be adapted to credit the current state of the art. However, the overall method is sound in itself. Interestingly, the system functions fully automatically, the system can adapt the applied force on the run, and the probe orientation is adapted automatically to analyze nodules. It would be helpful if the authors could clarify some points regarding the method:

- Why was only one side of the neck scanned?
- In general, I like the idea of having a sitting patient instead of a lying patient. However, Figure 4 looks like the setup did not enable a big neck support allowing for a lot of patient movement. This patient movement was discussed slightly, but it would be great to have a more consistent discussion on this topic.
- Did the study cohort include participants with several nodules? In that case, how does the

system handle multiple nodules? Moreover, how does it make sure not to visit the same nodule several times? Are the nodules somehow flagged after being detected?

- Page 10: Did the authors use data augmentation during training?
- Page 18: How were the two skeleton joint points defined?

Analytical Approach

Evaluations were performed and presented in detail. However, more explanation can be added to some results, as mentioned in the other sections. Statistical tests were performed for different evaluations. Moreover, violin and box plots depict the necessary evaluation results. Discussing the results depicted in Figure 6b-i in more detail would be very valuable.

Suggested Improvements:

Here I am suggesting some additional improvements. I suggest the authors adapt these changes and those mentioned in the other sections.

- Page 1: 'The outcoming images vary greatly among doctors, even the same doctor can produce different scan results under different conditions.' I would suggest that the authors add a reference for this.
- Page 2: Robotic systems of level 3 should be compared against robotic assistant systems - I would add: and clinical standard practice.
- Page 15: 'In this study, we developed an automated US diagnostic robot with artificial intelligence, which can accurately diagnose thyroid nodules.'. The system detects but does not diagnose nodules. The authors should change this sentence.
- Figure 3 and 6: The US images are tiny and hard to analyze.
- Figure 3: It would be great to describe the graphs in more detail.
- Page 15: 'The system is based on an unmanned automatic scanning method,...' - 'unmanned' sounds weird in this context. I suggest replacing the word.
- Page 18: It would be nice to have an image depicting the marker points on the thyroid CT.

Clarity and Context:

The provided results cover different aspects of robotic ultrasound scanning and nicely compare against manual US acquisitions. This comparison to the manual task is performed by several doctors and analyzed in detail. This is highly appreciated as comparisons to clinical practice are very valuable. Furthermore, including an evaluation from the participants on the robotic scanning process is appreciated. However, some results should be explained more explicitly and in more detail.

- Page 10: Please provide more information about the volunteers (age, gender, weight, etc.).
- Figure 7: The plot shows that the participants do not believe the system can replace doctors. It would be interesting to analyze the reasons why this is the case.
- Figure 6f-i: The notion of 'p-value = 0****' is not common. I suggest that the authors add an explanation. Furthermore, ROPS, DOPS, RIPS, and DIPS are not defined in the caption.
- Table 1: why is the force exerted by the robot marked as the best force?
- Tables 2 and 3 require significantly more information. What do these tables show? Which questionnaire was used? The rows seem to correlate to each physician, but what do the numbers inside the table relate to? Which metric is the image quality given in? This information should be added to the table caption and inside the text.

References:

The manuscript references previous literature. However, some literature is missing, and some articles are referenced incorrectly. Furthermore, the related work section requires extensive revision. Some concrete points are mentioned below:

- The following very relevant reference is missing: Kojcev, R., Khakzar, A., Fuerst, B., Zettinig, O., Fahkry, C., DeJong, R., Richmon, J., Taylor, R., Sinibaldi, E., & Navab, N. (2017). On the reproducibility of expert-operated and robotic ultrasound acquisitions. *International journal of computer assisted radiology and surgery*, 12(6), 1003–1011. <https://doi.org/10.1007/s11548-017-1561-1>
- Page 1: 'As these requirements cannot always be fulfilled, US examination produces

inconsistency and unreliability results due to insufficient standardized procedure [9, 11–14].¹ The references mentioned here are incorrect. [9] mentions high variability due to manual operation but does not discuss standardized procedures. [11] states that US procedures can be highly standardized but require manual execution. Similar to other references, [12] mentions that 'results rely on the doctors' skill and experience'. [13] states that 3D US volumes can be analyzed with a standard workflow after acquisition. I suggest that the authors double-check these references. [14] does not mention standardized procedures. I suggest that the authors remove these references and add fitting ones.

- Page 2: Reference 30 is misplaced. This work does not require experts to design the acquisition procedure.

- Page 6: 'Unlike other examinations with less deformation, the optimal probe orientation for soft tissue (eg, thyroid) scanning is not always normal [40].'¹ Here the reference is misplaced because [40] proposes a system for normal probe positioning.

Conclusion

Based on the mentioned points, I suggest the paper undergoes a major revision.

Reviewer #2 (Remarks to the Author):

A. Summary of the key results

The authors proposed a fully autonomous robotic ultrasound system for thyroid scanning, which using ultrasound images for real-time thyroid nodule detection, force feedback and depth image for robot motion control. They compared the system with manual scans on human participants, and also evaluate participants' satisfaction.

This work completed the clinical application of fully automated robotic ultrasound systems and achieve human-like fusion of ultrasound images, visual and force information. However, the innovation in this work is not apparent, it lies more in the integration of existing work into a cohesive system.

B. Originality and significance: if not novel, please include reference

As previously stated, this work emphasized the comprehensive character of the robotic ultrasound system. The contribution of this work to system implementation and clinical evaluation is significant as a fully automated robotic system, but it's hard to find novelty in the methodology compared with current works. For example in path planning, Huang et al. proposed a algorithm of using depth image for path planning [1], Z. Wang et al. using different views of depth images to plan 3D scanning path[2].

C. Data & methodology: validity of approach, quality of data, quality of presentation

The authors evaluated the system using reliable experimental data and presented the results effectively. However, the validity of some experimental data needs further confirmation.

As the authors claim, they used Kinect Azure DK RGBD camera to reconstructing the point cloud surface of the human neck and calculated the initial orientation of the probe for robotic scanning. However, the accuracy of this camera is difficult to calculate the exact normal vector, especially when the surface of the human neck is covered with enough ultrasound gel. Besides, the transparent and reflective properties of ultrasound gels can affect the work of depth cameras. Also, the movement of the volunteers will change the initial point based the previous data, so dynamic identification and tracking is required here.

D. Appropriate use of statistics and treatment of uncertainties

Yes, they did. The authors obtained the head and neck CT images from the Southern Hospital of Southern Medical University for the estimation of the thyroid scanning range. Their samples represented all age group populations and covered people with body mass index (BMI) ranging from 17.1 to 27.8. Also, to build the AI models for thyroid segmentation and nodule detection, they collected 118,137 thyroid images manually from the volunteers.

E. Conclusions: robustness, validity, reliability

This work evaluated the time spent and participants' satisfaction of the system and ultrasound

specialists in the scanning and diagnostic process. However, as they mentioned, the NDT: Nodule detection time of the system is a hundred times faster than the doctors' time cost, which is very counterintuitive. And they didn't mention how to identify the time cost of nodule detection for doctors in this paper. This comparison is not reliable.

F. Suggested improvements: experiments, data for possible revision

The authors had done necessary experiments in this paper. However, as they mentioned in the discussion, the contact force is different between groups of people. As the imaging quality of ultrasound images is influenced by the contact force, they should consider the robustness of ultrasound image processing algorithms under varying contact forces.

G. References: appropriate credit to previous work?

The authors focus more on work related to fully automated medical robots and ignored the few recent advances in the area of fully automatic robotic ultrasound systems, such as Z. Wang et al.[2], Z. Jiang et al.[3], J. Zhan et al.[4], F. Suligoj et al.[5].

H. Clarity and context: lucidity of abstract/summary, appropriateness of abstract, introduction and conclusions

The authors provided a detailed description of the experimental equipment, the assembly method, and the performance of the AI model. This paper had clear summary and reasonable conclusions. However, they described the evolution of automated medical robotic systems in detail such as Da Vinci Surgical System, but there is little background on robotic ultrasound systems. Besides, a sweep can take up to 4-5 minutes as shown in this paper, making it challenging to guarantee that the participants don't move. The authors didn't go into detail about how the system guarantees safety after detecting movement and how it continues to complete the sweep. This limitation needs to be mentioned in the discussion.

[1] Q. Huang, B. Wu, J. Lan, and X. Li, "Fully automatic three-dimensional ultrasound imaging based on conventional B-scan," *IEEE Trans. Biomed. Circuits Syst.*, vol. 12, no. 2, pp. 426–436, Apr. 2018.

[2] Z. Wang et al., "Full Coverage Path Planning and Stable Interaction Control for Automated Robotic Breast Ultrasound Scanning," *IEEE Trans. Ind. Electron.*, pp. 1–10, 2022.

[3] Z. Jiang et al., "Autonomous robotic screening of tubular structures based only on real-time ultrasound imaging feedback," *IEEE Trans. Ind. Electron.*, 2021.

[4] J. Zhan, J. Cartucho and S. Giannarou, "Autonomous tissue scanning under free-form motion for intraoperative tissue characterisation", *Proc. IEEE Int. Conf. Robot. Autom. (ICRA)*, pp. 11147-11154, 2020.

[5] F. Suligoj, C. M. Heunis, J. Sikorski, and S. Misra, "RobUSt—An autonomous robotic ultrasound system for medical imaging," *IEEE Access*, vol. 9, pp. 67456–67465, May. 2021.

Reviewer #3 (Remarks to the Author):

In this research, they provided a novel fully autonomous robotic ultrasound system (FARUS) for thyroid scanning on humans. But I still have some questions:

1. I would like to know the comparison of the size, shape, localization, calcification, and border of the nodules detected by FARUS and clinicians. Is FARUS better at detecting some specific nodules or all nodules?

2. As we all know, identifying malignant nodules is of great value for clinicians. Thyroid Imaging Reporting and Data Systems (TIRADS) is a 5-point classification to determine the risk of cancer in thyroid nodules based on ultrasound characteristics. Could FARUS assess the malignant risk of the nodules according to TIRADS?

Reviewer #1

This paper proposes a novel robotic ultrasound scanning system for thyroid nodule detection. The presented approach functions fully autonomously. The method combines tasks such as an automatic approach to the neck, US image-based navigation, thyroid nodule detection, and visualization of the nodules from two views, in-plane, and out-of-plane. The system functions successfully and shows superior results compared to manual nodule detection performed by different doctors. Furthermore, the authors report that the volunteers felt comfortable with the system.

Response to comment

We thank the reviewer for the positive evaluation and valuable comments on our manuscript.

Major correction in referencing prior work:

The current embedding of this manuscript in the scientific landscape is misleading. Most of the in-plane navigation cannot be seen as a contribution by the authors but was already introduced by Zielke et al. (RSV: robotic sonography for thyroid volumetry, IEEE RA-L 2022). The authors mention this paper in the related work section, showing that they are aware of this publication. However, this fact is not presented correctly in the methodology and contribution. Therefore, the authors must ensure that their contributions are correctly embedded into the scientific landscape, and that previous work is acknowledged sufficiently in an updated manuscript version and wherever it is used or is related to, to make the contributions and novelty of this work clearer.

Response to comment

We would like to sincerely thank the reviewer for the valuable suggestions.

- We are very sorry that this work by Zielke et al. (RSV: robotic sonography for thyroid volumetry, IEEE RA-L 2022) was not properly introduced in the relevant work section but has been corrected.

1) Introduction in manuscript (page 2)

Previous texts

Most existing autonomous systems are still at level 2 [27–30], which rely on an expert to design the acquisition procedure.

[30] Zielke, J., Eilers, C., Busam, B., Weber, W., Navab, N., Wendler, T.: Rsv: Robotic sonography for thyroid volumetry. IEEE Robotics and Automation Letters 7(2), 3342–3348 (2022).

Revised texts

In an effort to apply online image-guided methods to define the scanning trajectory and derive clinically relevant information out of the 3D reconstructed image, **Zielke et al. [34] implemented in-plane navigation** specifically designed for robotic sonography in thyroid volumetry. **However, in actual clinical practice, sonographers often use a combination of multiple views, such as transverse and longitudinal scans**, for the identification and diagnosis of both benign and malignant pathologies [35].

[34] Zielke, J., Eilers, C., Busam, B., Weber, W., Navab, N., Wendler, T.: Rsv: Robotic sonography for thyroid volumetry. *IEEE Robotics and Automation Letters* 7(2), 3342–3348 (2022).

[35] Kharchenko, V.P., Kotlyarov, P.M., Mogutov, M.S., Alexandrov, Y.K., Sencha, A.N., Patrunov, Y.N., Belyaev, D.V.: *Ultrasound Diagnostics of Thyroid Diseases*. Springer, New York (2010).

- Now, we have clearly known that it is inappropriate to include in-plane navigation as one of the contributions of this study. In this regard, we have reorganized the structure of this study to make the contribution and innovation of this work clearer.

2) Introduction in manuscript (page 3)

Previous texts

In order to eliminate these obstacles, we designed a fully autonomous robotic ultrasound system (FARUS), as shown in Fig. 1. To the best of our knowledge, this is the first fully autonomously ultrasound robotic system for thyroid scanning. As our first contribution, we used an autonomous robot to perform an enhanced thyroid scan. Necessary fundamental autonomous implementations are listed below. 1) General scanning range determination based on CT thyroid image data. 2) Dynamic path planning based on image control and force control. 3) Probe orientation adjustment based on Bayesian optimization. 4) Central region position tracking based on image segmentation. 5) Possible nodular lesion detection in ultrasound images based on object detection. In the new ultrasound scanning setting, the examination procedure has outstanding isolation and excellent stability, which can assist and replace the work of medical staff in dangerous environments such as high contagion, with great potential to serve the great needs in ultrasound screening in pandemic crisis, such as COVID-19.

The second contribution of this work is the ability to implement autonomous scanning strategies under the soft tissue, which presents various technical and workflow challenges. Unlike other examinations with less deformation such as orthopedics

[40,41], the optimal orientation for soft tissue is not necessarily the normal. Ultrasonography of the thyroid presents multiple challenges—including the small size of the thyroid lobe and the need for precise coverage of the lobe; soft tissue deformation; force sensitivity; image quality optimization; and the effect of motion on trajectory planning. Here, we have developed machine learning, computer vision and advanced control technologies to meet the needs of the thyroid scanning. In this process, depending on the fusion of sensing technology and vision, we obtained high-quality ultrasound images comparable to those manually collected by physicians, achieving accurate and real-time identification of thyroid nodules.

Revised texts

In order to eliminate these obstacles, we designed a fully autonomous robotic ultrasound system (FARUS), as shown in Fig. 1. To the best of our knowledge, **this is the first-in-human study of fully autonomous robotic US scanning for thyroid. In conventional US examinations, the process involves a division of responsibilities between sonographer and radiologist. However, this FARUS system combines the expertise of both roles into a single autonomous unit.** Here, we achieved a human-like fusion of both in-plane and out-of-plane scanning, allowing for comprehensive scanning of the thyroid region, and providing a detailed evaluation of the anatomy. The system overcomes the challenge of locating thyroid targets through the implementation of a reinforcement learning strategy. It enables probe orientation optimization with Bayesian optimization. Furthermore, the FARUS system uses deep learning techniques for real-time segmentation of the thyroid gland and potential nodules. In general, this system provides a convenient screening tool integrating intelligent nodule detection, lesion localization, automatic measurement, and benign and malignant classification for thyroid nodule ultrasound, which can improve the detection rate of nodules and the ability to distinguish benign and malignant nodules and reduce the work burden for doctors.

The second contribution of **our study lies in the practical realization and clinical application of the FARUS that achieves high-quality US images in comparison with those manually collected by physicians, achieving accurate and real-time detection of thyroid nodules.** We substantiated the validity of our approach by conducting a comparative evaluation of FARUS-driven diagnostic results for thyroid nodules against the established hospital benchmark. We have conducted extensive evaluations and provided evidence of the system's performance and reliability. Our work addresses the gap between existing research and clinical application by demonstrating the successful deployment of this system in a real-world clinical setting.

Previous framework

- System design for autonomous ultrasound imaging
- Thyroid search and probe orientation optimization
- Fully autonomous in-plane ultrasound imaging
- Fully autonomous out-of-plane ultrasound imaging

Revised framework

- System design for autonomous ultrasound imaging
- Deep learning for gland and nodule segmentation
- Thyroid search and probe orientation optimization
- Fully autonomous robotic ultrasound imaging
- Thyroid nodule imaging reporting with FARUS

- We substantiated the validity of our approach by conducting a comparative evaluation of FARUS-driven diagnostic results for thyroid nodules against the established hospital benchmark, as shown in the figure below. FARUS proved to be a reliable and effective tool for nodule diagnosis.

The results of the scoring and classification process according to the ACR-TIRADS are presented in **Fig. 6**. The **Fig. 6b** shows the nodule's composition is categorized as solid, leading to a score of 2, while its echogenicity is classified as hypo, also resulting in a score of 2. The nodule demonstrates a well-defined boundary and a regular shape, contributing to a score of 0. Additionally, the comparison of height and weight exceeds 1, resulting in a score of 3. Lastly, no echogenic foci are observed within the nodule, leading to a score of 0. The total score amounts to 7, leading to the classification of the nodule as level 5 or highly suspicious. **Figure 6b** illustrates the nodule's composition as partial, leading to a score of 1, and its echogenicity as hypo, resulting in a score of 2. The nodule's characteristic features, including a clear boundary and regular shape, warrant a score of 0 points. Additionally, the comparison of height and weight yields a score of 0 as it is less than 1. Furthermore, no presence of peripheral calcification within the nodule leading to a score of 0. Consequently, the cumulative score amounts to 3, classifying the nodule as level 3 or mildly suspicious.

In this research, we introduce the FARUS method, which aims to scan, detect, and classify nodules gathered from 19 individuals. Furthermore, we compare the data of each patient with the evaluations provided by doctor. According to the doctor's evaluation, 17 individuals were found to have nodules, while 2 individuals showed no presence of nodules. Our proposed system, FARUS, successfully identified 13 individuals as having nodules and 6 individuals as not having nodules. To analyze the accuracy of FARUS, we compared the scoring and classification of 17 nodules

among the 13 individuals successfully detected by both FARUS and the doctor. Each nodule was matched to the doctor's report based on its location and shape. **Table 1** displays a comparison between FARUS and doctor scoring and the classification of thyroid nodules. Among the 17 nodules assessed, 5 were diagnosed with the same score by both the doctor and the proposed FARUS system. **Figure 6a** illustrates scoring and classification of nodule diagnosed by doctor and FARUS. This can be attributed to the ultrasound image's appropriate brightness and contrast, resulting in well-distributed pixels. Additionally, 8 nodules exhibited a score difference of 1, primarily attributed to discrepancies in echogenicity or composition classifications. Furthermore, there was one nodule with a score difference of 2 and another with a score difference of 4, caused by differing classifications in both echogenicity and composition. Moreover, one nodule displayed a score difference of 3 due to variations in shape classification.

The main reason for the different classifications by FARUS is the use of a probe different from the one used by the doctor. The doctor's probe consistently produces a specific color of the thyroid gland for each patient, while the brightness and contrast of ultrasound images produced by our probe may vary across different patients with varying ages and weights. The proposed FARUS system demonstrated adequate accuracy in classifying nodules, with 5 out of 17 cases showing complete agreement with the doctor's diagnosis. Furthermore, for the remaining nodules, the discrepancies were primarily limited to 1 score difference, mainly arising from variations in echogenicity and composition. Overall, FARUS proved to be a reliable and effective tool for nodule classification.

Newly added Table 1

Table 1 Comparison of thyroid nodule scoring and classification between FARUS and doctor

No	Name	Nodule	Diagnosed by	Size	Composition	Echogenicity	Margin	Shape	Echogenicity Foci	Total	Difference	Reason
1	PATIENT_1L	#1	FARUS Doctor	6.1*5.3 R 3.2*2.6*2.9 R	2 2	2 2	0 0	3 3	0 0	7 7	0	Echogenicity,
2	PATIENT_2R	#2	FARUS Doctor	2.1*3.2 4.5*3.7*3 R	2 1	2 1	0 0	0 0	0 0	4 2	2	Echogenicity,
3	PATIENT_3R	#3	FARUS Doctor	4.6*5.8 R 6*4*7 R	0 0	0 0	0 0	0 0	0 0	0 0	0	
4	PATIENT_3R	#4	FARUS Doctor	2.5*4.1 R 6*4.5*5.6 R	2 1	1 1	0 0	0 0	0 0	3 2	1	Composition
5	PATIENT_4L	#5	FARUS Doctor	9.8*11.9 L 13*8*11 L	2 2	2 1	0 0	0 0	0 0	4 3	1	Echogenicity
6	PATIENT_5L	#6	FARUS Doctor	2.6*4.2 L 3*2*2.8 L	2 0	2 0	0 0	0 0	0 0	4 0	4	Echogenicity,
7	PATIENT_5R	#7	FARUS Doctor	3.8*4.8 R 5*1.6*7 R	2 1	1 2	0 0	0 0	0 0	3 3	0	Echogenicity,
8	PATIENT_6L	#8	FARUS Doctor	4.9*6.2 L 5*3*6 L	1 1	2 1	0 0	0 0	0 0	3 2	1	Echogenicity
9	PATIENT_7L	#9	FARUS Doctor	6.1*6.7 L 7*5*8 L	1 1	2 2	0 0	0 0	0 0	3 3	0	
10	PATIENT_7R	#10	FARUS Doctor	3.8*6.2 R 6*4*7 R	2 1	2 2	0 0	0 0	0 0	4 3	1	Composition
11	PATIENT_8R	#11	FARUS Doctor	6.5*8.6 R 7.5*5.4*8 R	1 1	2 1	0 0	0 0	0 0	3 2	1	Echogenicity
12	PATIENT_9L	#12	FARUS Doctor	11.1*10.7 L 11*9*19 L	1 1	2 2	0 0	3 0	0 0	6 3	3	Shape
13	PATIENT_9L	#13	FARUS Doctor	9.3*0.99 L 10*10*11 L	1 1	2 2	0 0	0 0	2 2	5 5	0	
14	PATIENT_10R	#14	FARUS Doctor	5.1*5.2 R 8*5*8 R	1 1	2 1	0 0	0 0	0 0	3 2	1	Echogenicity
15	PATIENT_11R	#15	FARUS Doctor	2.2*3.6 R 3*2*3 R	1 1	2 1	0 0	0 0	0 0	3 2	1	Echogenicity
16	PATIENT_12L	#16	FARUS Doctor	8.5*11.8 L 14*11*18 L	2 2	1 1	0 0	0 0	0 0	3 3	0	
17	PATIENT_13R	#17	FARUS Doctor	8.6*9.5 R 10*8*7.5 R	2 1	1 1	0 0	0 0	0 0	3 2	1	Composition

Newly added Figure 6

Fig. 6 Scoring and classification of thyroid nodules based on ACR-TIRADS. a Comparison of the thyroid nodules found by FARUS and doctor. **b**, Result of scoring and classification diagnosed by FARUS based on ACR-TIRADS. **c**, Correlation between numbers of pixels over brightness in thyroid image.

- We introduce a novel architecture called VariaNet, specifically designed for thyroid and nodule segmentation, as illustrated in Fig. 2a. To benefit from the prior knowledge provided by the thyroid lobe mask, we generate thyroid lobe pseudo labels during the training phase from nodule images. Subsequently, we incorporated spatial and feature constraints to enhance nodule segmentation based on these pseudo-labels. The spatial constraint ensures that the segmented

nodules remain in proximity to the thyroid lobe, while the feature constraint allocates greater weights to regions within the nodule mask that share analogous gray values with the thyroid lobe. By incorporating these advancements, VariaNet demonstrates high performance in thyroid and nodule segmentation.

Newly added Figure 2

Fig. 2 Thyroid gland and nodule segmentation with deep learning networks. **a**, Thyroid and nodule segmentation network based on pre-training and weakly supervision. The feature and spatial losses are used to provide prior knowledge to the network considering the diversity of nodule samples and the space constrain between thyroid lobes. **b**, Thyroid lobe segmentation based on pre-training. **c**, Our proposed VariaNet and its variants predict results for different types of nodules. **d**, Comparisons with the state-of-the-art segmentation models on our nodule testset. **e**, ROC curve of different algorithms on our nodule testset.

Validity:

The proposed method was evaluated against the performance of several doctors and through participant feedback. Both evaluations show promising results. The conclusions that were drawn are also coherent. Both doctor and volunteer groups were sufficiently large for the evaluation. In the future, testing the system on actual patients would be interesting, and discussing this limitation in the paper would be helpful. However, this is not a reason to reject the current paper.

Response to comment

First of all, we would like to thank the reviewers for their recognition of our work.

With the approval of the ethical review committee, we recruited 19 patients. First of all, we scanned both sides of the thyroid gland of 19 participants with FARUS, and at the same time, these participants underwent nodule diagnosis at The Third Affiliated Hospital of Sun Yat-Sen University Lingnan Hospital(<https://lnyy.zssy.com.cn/>) to verify the robotic scans with the classification of benign and malignant nodules.

Secondly, the robot automatic ultrasonic system provides a convenient screening tool integrating intelligent nodule detection, lesion localization, automatic measurement, and benign and malignant classification for thyroid nodule ultrasound, which can improve the detection rate of nodules and the ability to distinguish benign and malignant nodules and reduce the work burden for doctors.

Importantly, the system has been tested in clinical trials with actual patients, and the results are as follows:

- The patient's clinical nodule diagnosis report and robot scanning scene are shown as follow(The source file of diagnosis report and scanning scene video is stored in:https://drive.google.com/drive/folders/1WOe_0WVlKOJLhRJJXSVz4PmCtsebJo9p?usp=sharing):

Images and data of patients redacted due to lack of consent

- The patient's nodule segmentation results are shown below (source file is stored in:https://drive.google.com/drive/folders/1WOe_0WV1KOJLhRJJXSVz4PmCtsebJo9p?usp=sharing):

The ultrasound images scanned by the robot	The segmentation result of the thyroid gland and the nodule	The report of the doctor manually detecting the nodule
	 02Left		 03Right		 04Right		 07Left	

07Right

09Left

09Right

10Right

11Left

11Left

12Left

12Right

- The benign and malignant classification results of patients' nodules are shown in the figure below:

In this research, we introduce the FARUS method, which aims to scan, detect, and classify nodules gathered from 19 individuals. Furthermore, we compare the data of

each patient with the evaluations provided by doctor. According to the doctor's evaluation, 17 individuals were found to have nodules, while 2 individuals showed no presence of nodules. Our proposed system, FARUS, successfully identified 13 individuals as having nodules and 6 individuals as not having nodules.

To analyze the accuracy of FARUS, we compared the scoring and classification of 17 nodules among the 13 individuals successfully detected by both FARUS and the doctor. Each nodule was matched to the doctor's report based on its location and shape. **Table 1** displays a comparison between FARUS and doctor scoring and the classification of thyroid nodules.

Newly added Table 1

Table 1 Comparison of thyroid nodule scoring and classification between FARUS and doctor

No	Name	Nodule	Diagnosed by	Size	Composition	Echogenicity	Margin	Shape	Echogenicity Foci	Total	Difference	Reason
1	PATIENT_1.L	#1	FARUS Doctor	6.1*5.3 R 3.2*2.6*2.9 R	2 2	2 2	0 0	3 3	0 0	7 7	0	Echogenicity,
2	PATIENT_2.R	#2	FARUS Doctor	2.1*3.2 4.5*3.7*3 R	2 1	2 1	0 0	0 0	0 0	4 2	2	Echogenicity,
3	PATIENT_3.R	#3	FARUS Doctor	4.6*5.8 R 6*4*7 R	0 0	0 0	0 0	0 0	0 0	0 0	0	
4	PATIENT_3.R	#4	FARUS Doctor	2.5*4.1 R 6*4.5*5.6 R	2 1	1 1	0 0	0 0	0 0	3 2	1	Composition
5	PATIENT_4.L	#5	FARUS Doctor	9.8*11.9 L 13*8*11 L	2 2	2 1	0 0	0 0	0 0	4 3	1	Echogenicity
6	PATIENT_5.L	#6	FARUS Doctor	2.6*4.2 L 3*2*2.8 L	2 0	2 0	0 0	0 0	0 0	4 0	4	Echogenicity,
7	PATIENT_5.R	#7	FARUS Doctor	3.8*4.8 R 5*1.6*7 R	2 1	1 2	0 0	0 0	0 0	3 3	0	Echogenicity,
8	PATIENT_6.L	#8	FARUS Doctor	4.9*6.2 L 5*3*6 L	1 1	2 1	0 0	0 0	0 0	3 2	1	Echogenicity
9	PATIENT_7.L	#9	FARUS Doctor	6.1*6.7 L 7*5*8 L	1 1	2 2	0 0	0 0	0 0	3 3	0	
10	PATIENT_7.R	#10	FARUS Doctor	3.8*6.2 R 6*4*7 R	2 1	2 2	0 0	0 0	0 0	4 3	1	Composition
11	PATIENT_8.R	#11	FARUS Doctor	6.5*8.6 R 7.5*5.4*8 R	1 1	2 1	0 0	0 0	0 0	3 2	1	Echogenicity
12	PATIENT_9.L	#12	FARUS Doctor	11.1*10.7 L 11*9*19 L	1 1	2 2	0 0	3 0	0 0	6 3	3	Shape
13	PATIENT_9.L	#13	FARUS Doctor	9.3*0.99 L 10*10*11 L	1 1	2 2	0 0	0 0	2 2	5 5	0	
14	PATIENT_10.R	#14	FARUS Doctor	5.1*5.2 R 8*5*8 R	1 1	2 1	0 0	0 0	0 0	3 2	1	Echogenicity
15	PATIENT_11.R	#15	FARUS Doctor	2.2*3.6 R 3*2*3 R	1 1	2 1	0 0	0 0	0 0	3 2	1	Echogenicity
16	PATIENT_12.L	#16	FARUS Doctor	8.5*11.8 L 14*11*18 L	2 2	1 1	0 0	0 0	0 0	3 3	0	
17	PATIENT_13.R	#17	FARUS Doctor	8.6*9.5 R 10*8*7.5 R	2 1	1 1	0 0	0 0	0 0	3 2	1	Composition

Among the 17 nodules assessed, 5 were diagnosed with the same score by both the doctor and the proposed FARUS system. **The figure below** illustrates scoring and classification of nodule diagnosed by doctor and FARUS. This can be attributed to the ultrasound image's appropriate brightness and contrast, resulting in well-distributed pixels. Additionally, 8 nodules exhibited a score difference of 1, primarily attributed to discrepancies in echogenicity or composition classifications. Furthermore, there was one nodule with a score difference of 2 and another with a score difference of 4, caused by differing classifications in both echogenicity and composition. Moreover, one nodule displayed a score difference of 3 due to variations in shape classification.

Supplemental Figure

Significance:

The paper proposes an entire autonomous robotic ultrasound scanning framework for thyroid nodule detection. A major concern is that the in-plane automatic navigation is the same method proposed by Zielke et al. (RSV: robotic sonography for thyroid volumetry, IEEE RA-L 2022). This image-based navigation is, therefore, not novel and

cannot be seen as a contribution. It is also of concern that the authors did not reference this existing work appropriately. The authors should update this reference. However, on top of the existing work, the authors combine previous literature with a key frame-based automatic approach, nodule detection, and an out-of-plane rotation for multi-view analysis of the nodule. Therefore, the contributions are still relevant for the community but have to be updated based on the existing state of the art.

Response to comment

We thank the reviewer for the constructive comment.

We have revised Introduction to reference this existing work proposed by Zielke et al. (RSV: robotic sonography for thyroid volumetry, IEEE RA-L 2022) appropriately. Meanwhile, we have reorganized the structure of this study to make the contribution and innovation of this work clearer.

For more details, please refer to our responses for comment (**Major correction in referencing prior work**).

Data and Methodology:

The quality and quantity of the data are well presented. Furthermore, the gathered data is presented nicely. Based on the comments in the Significance section, the presentation of the methodology has to be adapted to credit the current state of the art. However, the overall method is sound in itself. Interestingly, the system functions fully automatically, the system can adapt the applied force on the run, and the probe orientation is adapted automatically to analyze nodules. It would be helpful if the authors could clarify some points regarding the method:

Response to comment

We thank the reviewer for the positive evaluation and valuable comments on our manuscript. The followings are the point-to-point replies to the comments.

Q1) Why was only one side of the neck scanned?

Response to comment

A1) Thank you for the constructive comment. In the previous experiment, from the perspective of scanning technology, only the thyroid segmentation effect and the nodule detection rate were considered to be the same on both sides of the scan, so we only scanned one side. In the current clinical trial, in order to have a more comprehensive grasp of the patient's condition from a clinical point of view, we scan both sides.

Q2) In general, I like the idea of having a sitting patient instead of a lying patient. However, Figure 4 looks like the setup did not enable a big neck support allowing for a lot of patient movement. This patient movement was discussed slightly, but it would be great to have a more consistent discussion on this topic.

Response to comment

A2) We thank the reviewer for the insightful comment.

In this paper, we employ reinforcement learning techniques to dynamically adjust the robotic arm's position, facilitating the attainment of an optimal horizontal alignment for capturing thyroid images. Motivation for Reinforcement Learning (RL) lies in its ability to enable machines to learn and make decisions in dynamic, uncertain environments without the need for labeled datasets. FARUS system has limitations in compensating for large-scale movements. Although it incorporates reinforcement learning and force control to allow for small patient movements, excessive movements can cause the FARUS system to lose tracking and stop. Kindly direct your attention to the updated Part 3 within the Results section for more comprehensive information.

Q3) Did the study cohort include participants with several nodules? In that case, how does the system handle multiple nodules? Moreover, how does it make sure not to visit the same nodule several times? Are the nodules somehow flagged after being detected?

Response to comment

A3) We appreciate your comment regarding the multiple nodules.

In our experiment, once detected, multiple nodules are identified through the segmentation of the prediction mask. The connected components within the prediction mask are analyzed, leading to the determination of the area and centroid point position for each nodule in the mask. Additionally, the probe's displacement in each frame is considered, as it contributes to the observed changes in the generated ultrasound (US) image.

There are two primary criteria for flagging nodules in the segmented prediction mask within a frame sequence: Intersection over Union (IoU) and centroid point position. The IoU and centroid point position are computed between the current frame and the subsequent frame, and a comparison is made between them. If the nodule area in the current frame (A) intersects with the nodule area in the next frame (B) and the

displacement of A and B is less than the probe's displacement, then nodule B is labeled as belonging to the same nodule as A.

Q4) Page 10: Did the authors use data augmentation during training?

Response to comment

A4) We thank the reviewer for the insightful comment. To enhance model robustness, generalization ability, and overall performance, we applied various data augmentation transformations to our training samples.

In our previous work, we utilized the UNet algorithm for thyroid segmentation and the YoloV3 algorithm for detecting suspected nodules. For the UNet segmentation task, we implemented transformation methods such as flips, shifts, random brightness contrast, and crop area with mask. The adjustment of image brightness and contrast, in particular, allows us to simulate volunteers with different ages, and scanning statuses of contact between the US probe and human neck, aiding the model in recognizing patterns in diverse forms. For suspicious nodules detection task, we also applied data augmentation to training data, including random flipping, scaling, translation, and random rotation. These data augmentation methods can simulate nodules of different sizes but similar characteristics, and can also simulate the movement of the nodules during robotic scanning.

Our current work aims to achieve the diagnosis of benign and malignant nodules. To facilitate nodule diagnosis and TIRADS grade calculation, we have refined the original nodule detection task into a nodule segmentation task. As a result, our current work involves two segmentation tasks: thyroid lobe segmentation and nodule segmentation. For both thyroid lobe and nodule segmentation task, we adopted a data enhancement approach similar to the one used in the previous Unet segmentation process. This includes flips, shifts, random brightness contrast, and crop area with mask.

Q5) Page 18: How were the two skeleton joint points defined?

Response to comment

A5) We thank the reviewer for the insightful comment.

Anatomically, the thyroid gland is approximately between the clavicle and Adam's apple. Although Kinect may not provide direct thyroid location data, but we can estimate the thyroid 's position based on the location of the neck and the head.

Previously, we approximately defined the position of the thyroid gland as a quarter point between the key points of the head and neck. However, taking into account the deflection of the neck during human scanning, we adjusted the position by translating the key point by 1cm.

In the current experiment, we designate 4 skeletal points that are provided by kinect camera body tracking SDK as robot initial posture decision and scanning guidance points, as shown in the figure below. The 3D vector connecting the left clavicle to the right clavicle point is employed as the horizontal axis, while the 3D vector from the 'neck' point to the 'head' point is used as rough reference for neck orientation. We generate three planes for each scanning side, all planes intersect to the 'head' and 'neck' points but with varying angles. These planes intersect the skin surface of the neck, creating three 3D curves. Among these curves, the middle one is selected as the scanning path, while the curves on both sides determine the probe's yaw angle decision.

Schematic Diagram

Analytical Approach:

Evaluations were performed and presented in detail. However, more explanation can be added to some results, as mentioned in the other sections. Statistical tests were performed for different evaluations. Moreover, violin and box plots depict the necessary evaluation results. Discussing the results depicted in Figure 6b-i in more detail would be very valuable.

Response to comment

We thank the reviewer for the detailed comment. According to the reviewer's comments, we have discussed the results in Figure 6b-i in detail, and **the previous Figure 6 is Figure 7**, and the text is revised as follows:

1) Discussion in manuscript (page 20)

Previous texts

In the Fig. 6a, the FARUS completed transverse and longitudinal scanning in the same manner as a doctor. In the OPS phase, FARUS can detect suspected nodules as well as cover the carotid arteries. During the IPS phase, the FARUS was able to scan a continuous area from the upper pole to the lower pole of the thyroid gland, while ensuring that the thyroid gland was centered. The manually controlled centering by the doctors was worse than by the FARUS, as seen in Fig. 6b and Fig.

6f. In addition, there was a large variation in centering performance between doctors, indicating much variation in the sweeping skills between doctors. For the OPS phase, the variance of the robot's centering was larger than doctors', because the scanned region of the FARUS were wider than doctors. The entropy value of the doctors' image was significantly lower than the FARUS (Fig. 6c, g), this is because the Bayesian optimization process adjusts the probe angle to maximize the image entropy. As seen from Fig. 6d and Fig. 6h, the FARUS outperformed the physician in the OPS case, because the doctor sweeping process might switch the scanning mode resulting in less LRIS. The mean confidence of doctors was better compared to the FARUS (Fig. 6e, i), which is caused by the additional sweeping of the thyroid isthmus by the doctors.

Revised Figure 7

Revised texts

From **Fig. 7 b** and **f**, the centering error of the IPS phase of FARUS US scans is much smaller than that observed in scans conducted by five doctors. This difference can be attributed to the robot control process, which ensures centering through thyroid segmentation. Furthermore, the centering error of FARUS in the

IPS phase is smaller than that in the OPS phase. This discrepancy arises due to the relatively larger area of the thyroid gland in the longitudinal views, often extending to the image's borders. **Figure 7c** and **g** illustrate that the entropy value of FARUS's US scan images is significantly higher compared to those captured by doctors. This dissimilarity arises from the differences in scan coverage by doctors, as they may simultaneously examine both thyroid lobes, leading to potential inclusion of the isthmus region of the thyroid gland. Additionally, FARUS exhibit lower variance in image entropy, possibly due to the robot's more continuous scanning process and

the stability of force control. From **Fig. 7d** and **7h**, it is evident that the contact status of robotic scans and manual scans are similar, indicating an even contact of the probe throughout the scanning process. However, a substantial difference is observed between FARUS and doctors in the IPS stage, which could be attributed to variations in scanning force and methods. The robot conducts scans in a seated position, while doctors may have participants lean back, resulting in dissimilarities in scanning techniques and approaches, leading to increased variance in physicians' scanning contact status. **Figures 7e** and **i** reveal that the mean confidence of FARUS's US scan images is smaller than that of doctors. This discrepancy arises because the mean confidence metric reflects the contact condition of the image at the pixel level, which is influenced by the scan range and tissue anatomy characteristics. For instance, the proximity of the esophagus and trachea to the thyroid gland can impact mean confidence values. Even if the contact is good, scanning these adjacent structures can decrease the mean confidence. Furthermore, the differences in scan range between FARUS and the doctor contribute to variations in mean confidence distributions observed in the images.

Suggested Improvements:

Here I am suggesting some additional improvements. I suggest the authors adapt these changes and those mentioned in the other sections.

Response to comment

We thank the reviewer for the insightful comment. We have carefully addressed all the points, as shown below.

Q1) Page 1: 'The outcoming images vary greatly among doctors, even the same doctor

can produce different scan results under different conditions.' I would suggest that the authors add a reference for this.

Response to comment

A1) As suggested by the reviewer, we added references [10] to support the statement in the paper that manual acquisition of ultrasound images cannot obtain reliable and reproducible results. The relevant description in reference [10] is as follows:

“In most clinical applications, 2D high-resolution US imaging is used, requiring **manual repositioning of the transducer**. The interpretation of anatomy represented in US images is based on in-depth knowledge, training and experience. **This leads to a strong operator dependence and a possible variability of repeated measurements performed by one operator.**”

1) Introduction in manuscript (page 1)

Previous texts

The outcoming images vary greatly among doctors, even the same doctor can produce different scan results under different conditions.

Revised texts

The acquisition of US images can exhibit variability between different doctors, even a single doctor potentially producing varying scan results under different conditions [10].

Newly added references

[10] Kojcev, R., Khakzar, A., Fuerst, B., Zettinig, O., Fahkry, C., DeJong, R., Richmon, J., Taylor, R., Sinibaldi, E., Navab, N.: On the reproducibility of expert-operated and robotic ultrasound acquisitions. International journal of computer assisted radiology and surgery 12, 1003–1011 (2017).

Q2) Page 2: Robotic systems of level 3 should be compared against robotic assistant systems - I would add: and clinical standard practice.

Response to comment

A2) We would like to thank the reviewer for raising the question.

Robotic assistant systems allow the robot and the on-site clinician to have shared control over the probe motion [1]. Moreover, the safety of the patient can be better monitored by the on-site clinician and the scanning protocols can be easily modified through patient-clinician communication. Therefore, the unique features of the human-robot cooperation systems make them more likely to adapt to clinical practice [2].

Compared with the robot assisted system, clinical effectiveness of robotic systems at level 3 needs to be further investigated. **Safety is a paramount issue for the robotic systems of level 3 to improve clinical acceptance of this technology. Compared with the systems under continuous control of a on-site clinician, the increased autonomy in the robotic systems of level 3 may bring a higher risk of patient injury due to malfunction** [3]. Although most existing the robotic systems of level 3 have incorporated force sensing to improve patient safety, excessive force may be accidentally exerted onto the patient due to sensor or controller failure [4]. Using more sensors to monitor the system may improve the safety and reliability of the acquisition, but it would also increase the burden of data processing.

References

- [1] D. R. Swerdlow, K. Cleary, E. Wilson, B. Azizi-Koutenaei, and R. Monfaredi, "Robotic arm-assisted sonography: Review of technical developments and potential clinical applications," *Amer. J. Roentgenol.*, vol. 208, no. 4, pp. 733–738, 2017.
- [2] K. Li, Y. Xu and M. Q. . -H. Meng, "An Overview of Systems and Techniques for Autonomous Robotic Ultrasound Acquisitions," in *IEEE Transactions on Medical Robotics and Bionics*, vol. 3, no. 2, pp. 510-524, May 2021.
- [3] G.-Z. Yang et al., "Medical robotics—Regulatory, ethical, and legal considerations for increasing levels of autonomy," *Sci. Robot.*, vol. 2, no. 4, p. 8638, 2017.
- [4] R. Tsumura and H. Iwata, "Robotic fetal ultrasonography platform with a passive scan mechanism," *Int. J. Comput. Assist. Radiol. Surg.*, vol. 15, pp. 1323–1333, Feb. 2020.

Q3) Page 15: 'In this study, we developed an automated US diagnostic robot with artificial intelligence, which can accurately diagnose thyroid nodules.'. The system detects but does not diagnose nodules. The authors should change this sentence.

Response to comment

A3) We are very sorry for not appropriate to use the word (diagnose) in the previous manuscript. **However, this system currently provides a convenient screening tool integrating intelligent nodule detection and benign and malignant classification for thyroid nodule ultrasound.** Therefore, in the current manuscript, the authors will keep the words in that sentence unchanged.

- The benign and malignant classification results of patients' nodules are shown in the figure below:

In this research, we introduce the FARUS method, which aims to scan, detect, and classify nodules gathered from 19 individuals. Furthermore, we compare the data of each patient with the evaluations provided by doctor. According to the doctor's evaluation, 17 individuals were found to have nodules, while 2 individuals showed no presence of nodules. Our proposed system, FARUS, successfully identified 13 individuals as having nodules and 6 individuals as not having nodules.

To analyze the accuracy of FARUS, we compared the scoring and classification of 17 nodules among the 13 individuals successfully detected by both FARUS and the doctor. Each nodule was matched to the doctor's report based on its location and shape. **Table 1** displays a comparison between FARUS and doctor scoring and the classification of thyroid nodules.

Newly added Table 1

Table 1 Comparison of thyroid nodule scoring and classification between FARUS and doctor

No	Name	Nodule	Diagnosed by	Size	Composition	Echogenicity	Margin	Shape	Echogenicity Foci	Total	Difference	Reason
1	PATIENT_1L	#1	FARUS Doctor	6.1*5.3 R 3.2*2.6*2.9 R	2 2	2 2	0 0	3 3	0 0	7 7	0	Echogenicity,
2	PATIENT_2R	#2	FARUS Doctor	2.1*3.2 4.5*3.7*3 R	2 1	2 1	0 0	0 0	0 0	4 2	2	Echogenicity,
3	PATIENT_3R	#3	FARUS Doctor	4.6*5.8 R 6*4*7 R	0 0	0 0	0 0	0 0	0 0	0 0	0	
4	PATIENT_3R	#4	FARUS Doctor	2.5*4.1 R 6*4.5*5.6 R	2 1	1 1	0 0	0 0	0 0	3 2	1	Composition
5	PATIENT_4L	#5	FARUS Doctor	9.8*11.9 L 13*8*11 L	2 2	2 1	0 0	0 0	0 0	4 3	1	Echogenicity
6	PATIENT_5L	#6	FARUS Doctor	2.6*4.2 L 3*2*2.8 L	2 0	2 0	0 0	0 0	0 0	4 0	4	Echogenicity,
7	PATIENT_5R	#7	FARUS Doctor	3.8*4.8 R 5*1.6*7 R	2 1	1 2	0 0	0 0	0 0	3 3	0	Echogenicity,
8	PATIENT_6L	#8	FARUS Doctor	4.9*6.2 L 5*3*6 L	1 1	2 1	0 0	0 0	0 0	3 2	1	Echogenicity
9	PATIENT_7L	#9	FARUS Doctor	6.1*6.7 L 7*5*8 L	1 1	2 2	0 0	0 0	0 0	3 3	0	
10	PATIENT_7R	#10	FARUS Doctor	3.8*6.2 R 6*4*7 R	2 1	2 2	0 0	0 0	0 0	4 3	1	Composition
11	PATIENT_8R	#11	FARUS Doctor	6.5*8.6 R 7.5*5.4*8 R	1 1	2 1	0 0	0 0	0 0	3 2	1	Echogenicity
12	PATIENT_9L	#12	FARUS Doctor	11.1*10.7 L 11*9*19 L	1 1	2 2	0 0	3 0	0 0	6 3	3	Shape
13	PATIENT_9L	#13	FARUS Doctor	9.3*0.99 L 10*10*11 L	1 1	2 2	0 0	0 0	2 2	5 5	0	
14	PATIENT_10R	#14	FARUS Doctor	5.1*5.2 R 8*5*8 R	1 1	2 1	0 0	0 0	0 0	3 2	1	Echogenicity
15	PATIENT_11R	#15	FARUS Doctor	2.2*3.6 R 3*2*3 R	1 1	2 1	0 0	0 0	0 0	3 2	1	Echogenicity
16	PATIENT_12L	#16	FARUS Doctor	8.5*11.8 L 14*11*18 L	2 2	1 1	0 0	0 0	0 0	3 3	0	
17	PATIENT_13R	#17	FARUS Doctor	8.6*9.5 R 10*8*7.5 R	2 1	1 1	0 0	0 0	0 0	3 2	1	Composition

Among the 17 nodules assessed, 5 were diagnosed with the same score by both the doctor and the proposed FARUS system. **The figure below** illustrates scoring and classification of nodule diagnosed by doctor and FARUS. This can be attributed to the ultrasound image's appropriate brightness and contrast, resulting in well-distributed pixels. Additionally, 8 nodules exhibited a score difference of 1, primarily attributed to discrepancies in echogenicity or composition classifications. Furthermore, there was one nodule with a score difference of 2 and another with a score difference of 4, caused by differing classifications in both echogenicity and composition. Moreover, one nodule displayed a score difference of 3 due to variations in shape classification.

Supplemental Figure

Q4) Figure 3 and 6: The US images are tiny and hard to analyze.

Response to comment

A4) We would like to thank the reviewer for raising the question. According to the comments of the reviewers, we have re-cropped and typesetted the ultrasound pictures, and **the previous Figure 3 and Figure 6 are Figure 4 and Figure 7**. The results are as follows:

1) Results in manuscript (page 13)

Previous Figure 3

Fig. 3 Bayesian orientation optimization. **a**, Overview of Bayesian optimization (BO). Example illustrating a Gaussian process surrogate model fitting to data derive from a unknown target and its expected utility function maximizing to select the next candidate. **b**, The boxplot displaying the significant differences in image entropy values after BO phase for 70 participants. **c**, Probe orientation optimization procedure shown the five decisions made by BO with image entropy as loss function. **d**, US image entropy over time for five iterations of BO phase, during which the force value is between 2.2 N to 2.7 N to exclude the effect of the contact force on image entropy. **e**, The relationship between image entropy and mean confidence that characterize the contact condition of US image[47], as seen in a positive correlation between two evaluation metrics **f**, The relationship between the confidence level of the nodule detection frame and the image entropy value, showing that most thyroid nodules were located within images with entropy values exceeding 7.0.

Revised Figure 4

Fig. 4 Bayesian orientation optimization. **a**, Overview of Bayesian optimization (BO) Example illustrating a Gaussian process surrogate model fitting to data derive from a unknown target and its expected utility function maximizing to select the next candidate **b**, The boxplot displaying the significant differences in image entropy values after BO phase for 89 participants. **c**, Probe orientation optimization procedure shown the five decisions made by BO with image entropy as loss function. **e**, The relationship between image entropy and mean confidence that characterize the contact condition of US image[46], as seen in a positive correlation between two evaluation metrics **f**, During thyroid ultrasound scans, the median entropy remained stable for force values above 2.0 N, with no significant changes when the force exceeded 3.0 N.

2) Discussion in manuscript (page 20)

Previous Figure 6

Fig. 6 Evaluation of US scans collected by the FARUS and five doctors. **a**, The in-plane/out-of-plane US image sequences acquired by the FARUS and five experienced doctors on the same two participants. The green contour in the US images represent the segmented thyroid lobe region by the UNet network, while the green bounding boxes represent the detected suspicious nodules. **b-e**, Violin plots illustrating the four evaluation metrics between FARUS and five doctors. **f-i**, Boxplots displaying the four evaluation metrics between FARUS and five doctors.

Revised Figure 7

Q5) Figure 3: It would be great to describe the graphs in more detail.

Response to comment

A5) We would like to thank the reviewer for raising the question. In the revised manuscript, and **the previous Figure 3 is Figure 4**. Figure 4 has been described in detail as follows:

1) Results in manuscript (page 12)

Previous texts

In many instances, Bayesian optimization (BO) outperforms expert as well as other state-of-the-art global optimization algorithms. Bayesian optimization constructs a surrogate model for the objective function and quantifies the uncertainty through Bayesian inference. This surrogate model determines where the next candidate will be, as in Fig. 3a. During the BO phase, the position of the US probe remains constant; therefore, probe orientation and contact force are the two major factors in the entropy value of the US image. The force value is set between 2.2 N and 2.7 N to exclude the effect of the contact force on the image entropy, as in Fig. 3d. when the probe orientation remains constant and the force value exceeds 2.8 N, the entropy value of the image keeps stable, indicating sufficient conditions for contact force between the US probe and the skin. We consider a limited budget of $N = 5$ iterations to speed up the BO phase. The entropy of the US image increases from 6.93 to 7.16 in 5 iterations, and the max entropy corresponding to the optimal orientation was reached at the second iteration (Fig. 3c). When the number of iterations reached 5, a higher drop in performance was observed due to its exploration nature (Fig. 3c). The significant differences in image entropy values were observed after the BO phase for 70 participants (Fig. 3b). Furthermore, the omitted thyroid nodules might be detected in the US image during the optimization procedure (Fig. 3c). As illustrated in Fig. 3e, there is a positive relationship between the image entropy and confidence map that evaluates the contact condition at each pixel [47]. As the entropy value consumes less computation, it allows for a real-time control of image quality. Additionally, we can see that most of the US images with nodules have entropy values above 7.0 (Fig. 3f), indicating that the higher entropy value provides more favorable conditions for thyroid nodules detection.

Revised Figure 4

Fig. 4 Bayesian orientation optimization. **a**, Overview of Bayesian optimization (BO). Example illustrating a Gaussian process surrogate model fitting to data derive from a unknown target and its expected utility function maximizing to select the next candidate. **b**, The boxplot displaying the significant differences in image entropy values after BO phase for 89 participants. **c**, Probe orientation optimization procedure shown the five decisions made by BO with image entropy as loss function. **e**, The relationship between image entropy and mean confidence that characterize the contact condition of US image[46], as seen in a positive correlation between two evaluation metrics **f**, During thyroid ultrasound scans, the median entropy remained stable for force values above 2.0 N, with no significant changes when the force exceeded 3.0 N.

Revised texts

In many instances, Bayesian optimization (BO) outperforms expert as well as other state-of-the-art global optimization algorithms. Bayesian optimization constructs a surrogate model for the objective function and quantifies the uncertainty through Bayesian inference. This surrogate model determines where the next candidate will be, as in **Fig. 4a**. During the BO phase, the position of the US probe remains constant; therefore, probe orientation and contact force are the two major factors in the entropy value of the US image. To further explore the influence of contact force on image quality, we conducted an investigation involving 9 participants. Each participant was instructed to apply varying levels of force on the US probe, positioned at the robot's end, while ensuring safety under continuous manual monitoring. **Fig. 4 e** demonstrates that when the contact force between the probe and the skin exceeds 2 N, the median of the image entropy value remains stable. Taking scanning comfort and safety into account, we set the maximum contact force at 4 N. We consider a limited budget of $N = 5$ iterations to speed up the BO phase. The entropy of the US image varies from 7.172 to 7.173 in 5 iterations, and the max entropy corresponding to the optimal orientation was reached at the second iteration (**Fig. 4c**). When the number of iterations reached 5, a higher drop in performance was observed due to its exploration nature (**Fig. 4c**). The significant differences in image entropy values were observed after the BO phase for 89 participants (**Fig. 4b**). During this experiment, the probe was initially set at a relatively optimal angle, resulting in a less noticeable entropy increase before and after Bayesian optimization. However, between 13.4 s and 21.3 s, the US probe underwent a 10-degree angle adjustment, revealing that the entropy value of the US image responded more sensitively to angle changes compared to image confidence. It was speculated that this sensitivity could be attributed to the level of contact between the skin and the probe. As illustrated in **Fig. 4e**, there is a positive relationship between the image entropy and confidence map that evaluates the contact condition at each pixel [46]. As the entropy value consumes less computation, it allows for a real-time control of image quality.

Q6) Page 15: 'The system is based on an unmanned automatic scanning method,...' - 'unmanned' sounds weird in this context. I suggest replacing the word.

Response to comment

A6) We would like to thank the reviewer for raising the question. We are very sorry for not appropriate to use the word (unmanned') in the previous manuscript. We have modified the above statement as follows:

1) Discussion in manuscript (page 21)

Previous texts

The system is based on an unmanned automatic scanning method, so the advantage is that medical staffs do not have to contact with the participant, which can prevent the medical staff from being infected by the participant's infectious diseases.

Revised texts

The system operates on an autonomous scanning method, a significant benefit of which is the absence of direct contact between medical personnel and the participant. This arrangement effectively reduces the risk of transmission of any infectious diseases from the participant to the healthcare providers.

Q7) Page 18: It would be nice to have an image depicting the marker points on the thyroid CT.

Response to comment

A7) We would like to thank the reviewer for raising the question. We defined 60 marker points to label the thyroid gland, as shown in the **Figure 9**. They were 2 points above and below the isthmus of the thyroid; 8 points in the sagittal plane of the left and right lobes of the thyroid; and 2 points on the left and right of the entire thyroid cross-section; for the left and right lobes of the thyroid above the isthmus The height is divided into five equal parts, and 4 points are marked on the upper, lower, left, and right sides of the 4 cross-sections, for a total of 32 points; After manually marking all the key points, our custom Python program will generate a 3D model and measure the average measurements of the various parts. Through this model, we obtained the morphological data of the thyroid. The average height of thyroid gland was 4.76 (± 0.57) cm, and the average unilateral leaf width and thickness was 1.48 (± 0.34) cm and 1.52 (± 0.22) cm, respectively. Through regression analysis, we found that the width, length, thickness, and volume of the thyroid gland had very low correlations with gender, age, height, and weight. For this reason, the scan length was set to 6.47 cm based on the 3σ rule, and the scan width was set to 5.48 cm that was the probe width of 4.0 cm plus the average thyroid width.

Newly added Figure 9

Fig. 9 Three-dimensional model of thyroid gland based on CT image reconstruction

Clarity and Context:

The provided results cover different aspects of robotic ultrasound scanning and nicely compare against manual US acquisitions. This comparison to the manual task is performed by several doctors and analyzed in detail. This is highly appreciated as comparisons to clinical practice are very valuable. Furthermore, including an evaluation from the participants on the robotic scanning process is appreciated. However, some results should be explained more explicitly and in more detail.

Response to comment

We thank the reviewer for the positive evaluation on our manuscript.

Q1) Page 10: Please provide more information about the volunteers (age, gender, weight, etc.).

Response to comment

A1) In the preceding experiment, we regrettably failed to record the weight and age information of 70 volunteers. However, in the present study, we have successfully recruited 29 volunteers and meticulously documented their age, height, weight, and gender, as depicted in the figure below:

Supplemental Figure

Distribution of Height, Age, and Weight for 29 Ultrasound Volunteers

Volunteer	Age	Height(cm)	Weight(kg)
#1	55	155	75
#2	60	150	60
#3	65	155	65
#4	60	150	60
#5	65	155	65
#6	60	150	60
#7	65	155	65
#8	60	150	60
#9	65	155	65
#10	60	150	60
#11	65	155	65
#12	60	150	60
#13	65	155	65
#14	60	150	60
#15	65	155	65
#16	60	150	60
#17	65	155	65
#18	60	150	60
#19	65	155	65
#20	60	150	60
#21	65	155	65
#22	60	150	60
#23	65	155	65
#24	60	150	60
#25	65	155	65
#26	60	150	60
#27	65	155	65
#28	60	150	60
#29	65	155	65

Q2) Figure 7: The plot shows that the participants do not believe the system can replace doctors. It would be interesting to analyze the reasons why this is the case.

Response to comment

A2) We thank the reviewer for the insightful comment.

From an objective perspective:

- **The system cannot deal with complex information**

Besides the data captured by medical devices, doctors need to make complex decisions based on multimodal information during the diagnostic process. Robots, whose strength lies in their powerful data-processing capabilities, are sometimes powerless in such situations.

- **The system learns mostly from humans**

Robots acquiring diagnostic capabilities through a deep learning process, which means that it is mostly dependent on human experience. We can only assume that robots can't perform better than doctors since there's no standard statistical method for comparative calculations.

From the subjective perspective:

- **The system has no compassion**

Even though the system will provide excellent solutions, it is hard to mimic compassion. Because the key to compassion is the process of building trust: listening to the participant, focusing on the participant's needs, expressing compassion and responding in a way that makes the participant feel understood.

- **The system is not always safe**

The increased autonomy of robotic systems can lead to a higher risk of injury to patients due to malfunctions, such as sensors or controllers that may accidentally exert excessive force on patients. Also, in some cases human operation will always be faster, safer, more reliable than the system, or cheaper than the technology.

Q3) Figure 6f-i: The notion of 'p-value = 0****' is not common. I suggest that the authors add an explanation. Furthermore, ROPS, DOPS, RIPS, and DIPS are not defined in the caption.

Response to comment

A3) We thank the reviewer for the detailed comment.

The notion of 'p-value = 0****' was an oversight in the plotting of the statistics and has been corrected in the new manuscript.

ROPS, DOPS, RIPS, and DIPS are explained as follows:

IPS: in-plane scanning; OPS: out-of-plane scanning;
R stands for robotic; D is for doctor.

So,

ROPS: robotic out-of-plane scanning; DOPS: doctor out-of-plane scanning;

RIPS: robotic in-plane scanning; DIPS: doctor in-plane scanning.

Q4) Table 1: why is the force exerted by the robot marked as the best force?

Response to comment

A4) We thank the reviewers for their helpful comments. We take into account that the force applied by the robot has the lowest standard deviation, so we think that this force is the best. However, from a clinical point of view, skin tissue is elastic, and the standard deviation of force to compare robots and doctors is not rigorous and appropriate. Therefore, we only retained the comparison of the force between the robot and the doctor in the revised manuscript, and whether it has the best force is no longer discussed. **The previous Table 1 is Table 2.**

1) Discussion in manuscript (page 21)

Previous table

Table 1 Comparison of contact forces, probe motion transitions, nodule detection time, and total scanning time used for acquisition between the FARUS and professional doctors (Mean±SD).

	Robot	Doctor#1	Doctor#2	Doctor#3	Doctor#4	Doctor#5
F	2.53±0.52	1.98±0.89	2.51±0.67	2.43±0.61	2.53±0.71	2.51±0.71
V	3	11.7 ±2.1	11.4±1.8	12.6±2.5	12.2±2.2	12.4±1.9
ω	9.17	19.86±6.38	23.31±10.49	22.03±9.80	22.73±9.03	22.56±8.48
NDT	0.02	3.12±1.50	3.60±1.76	3.51±1.55	3.29±1.47	3.23±1.41
TST	213.0±85.3	69.9±26.6	73.8±27.1	58.2±26.6	65.2±32.3	64.1±21.5

Note: F: Force (N); Acc: Nodule detection accuracy; V: Velocity (mm/s); ω : Angular velocity (degree/s); NDT: Nodule detection time (s); TST : Total scanning time (s)

Revised table 2

Table 2 Comparison of contact forces, probe motion transitions, total scanning time used for acquisition between the FARUS and professional doctors (Mean±SD).

	Robot	Doctor#1	Doctor#2	Doctor#3	Doctor#4	Doctor#5
F	2.53±0.52	1.98±0.89	2.51±0.67	2.43±0.61	2.53±0.71	2.51±0.71
V	3	11.7 ±2.1	11.4±1.8	12.6±2.5	12.2±2.2	12.4±1.9
ω	9.17	19.86±6.38	23.31±10.49	22.03±9.80	22.73±9.03	22.56±8.48
TST	213.0±85.3	69.9±26.6	73.8±27.1	58.2±26.6	65.2±32.3	64.1±21.5

Note: F: Force (N); V: Velocity (mm/s); ω : Angular velocity (degree/s); TST : Total scanning time (s)

Q5) Tables 2 and 3 require significantly more information. What do these tables show? Which questionnaire was used? The rows seem to correlate to each physician, but what do the numbers inside the table relate to? Which metric is the image quality given in? This information should be added to the table caption and inside the text.

Response to comment

A5) We thank the reviewer for the insightful comment.

In the current experiment, this system(FARUS) provides a convenient screening tool integrating intelligent nodule detection, lesion localization, automatic measurement, and benign and malignant classification for thyroid nodule ultrasound. Therefore, the applied algorithm and the study object (healthy volunteers transformed into patients) have changed, so the questionnaire for thyroid and nodule scanning is not appropriate here, and we will delete the previous Table 2.

The previous Table 3 is now Table 5, and the details of Table 5 are described as follows:

Currently, most US robotic systems studies do not include comparisons with professional clinicians, nor do they evaluate participants' satisfaction. As shown in Table 5, we recruited 14 sonographers from 7 hospitals to make subjective evaluations of our acquired transverse and longitudinal images. It shows that the images acquired by the FARUS have good quality, centrality and integrity.

Revised table 5

Table 5 IPS phase evaluation from fourteen doctors' questionnaire regarding image quality, scanning completeness and centering performance.

Image quality (Mean±SD)	Scanning completeness					Centering performance				
	1	2	3	4	5	1	2	3	4	5
4.91±0.33	0	0	16	43	11	0	0	0	5	65
3.71±0.59	0	0	43	24	3	0	0	3	43	24
4.04±0.31	0	0	0	39	31	0	0	1	51	18
3.07±0.85	0	7	38	23	2	0	15	29	23	3
4.43±0.57	0	0	11	32	27	0	0	16	27	27
4.94±0.23	0	0	0	0	70	0	0	0	2	68
4.54±0.60	0	0	4	36	30	0	0	5	27	38
3.71±0.48	0	0	2	31	37	0	0	12	33	25
5.00±0.00	0	0	0	0	70	0	0	0	0	70
3.91±0.33	0	0	40	29	1	0	0	41	27	2
5.00±0.00	0	0	0	23	47	0	0	0	4	66
4.63±0.51	0	0	0	18	52	0	0	0	2	68
4.50±0.63	0	0	5	14	51	0	0	0	7	63
4.26±0.58	0	0	7	21	42	0	0	1	25	44

References:

The manuscript references previous literature. However, some literature is missing, and some articles are referenced incorrectly. Furthermore, the related work section requires extensive revision. Some concrete points are mentioned below:

Response to comment

We thank the reviewer for the constructive comment. We have carefully addressed all the points, as shown below.

Q1) The following very relevant reference is missing: Kojcev, R., Khakzar, A., Fuerst, B., Zettinig, O., Fakhry, C., DeJong, R., Richmon, J., Taylor, R., Sinibaldi, E., & Navab, N. (2017). On the reproducibility of expertoperated and robotic ultrasound acquisitions. *International journal of computer assisted radiology and surgery*, 12(6), 1003 - 1011. <https://doi.org/10.1007/s11548-017-1561-1>.

Response to comment

A1) We would like to thank the reviewer for raising the question. We added the reference to support the statement in the paper that manual acquisition of ultrasound images cannot obtain reliable and reproducible results. The relevant description in reference is as follows:

“In most clinical applications, 2D high-resolution US imaging is used, requiring **manual repositioning of the transducer**. The interpretation of anatomy represented in US images is based on in-depth knowledge, training and experience. **This leads to a strong operator dependence and a possible variability of repeated measurements performed by one operator.**”

1) Introduction in manuscript (page 1)

Previous texts

The outcoming images vary greatly among doctors, even the same doctor can produce different scan results under different conditions.

Revised texts

The acquisition of US images can exhibit variability between different doctors, even a single doctor potentially producing varying scan results under different conditions [10].

Newly added references

[10] Kojcev, R., Khakzar, A., Fuerst, B., Zettinig, O., Fahkry, C., DeJong, R., Richmon, J., Taylor, R., Sinibaldi, E., Navab, N.: On the reproducibility of expert-operated and robotic ultrasound acquisitions. International journal of computer assisted radiology and surgery 12, 1003 - 1011 (2017).

Q2) Page 1: 'As these requirements cannot always be fulfilled, US examination produces inconsistency and unreliability results due to insufficient standardized procedure [9, 11-14].' The references mentioned here are incorrect. [9] mentions high variability due to manual operation but does not discuss standardized procedures. [11] states that US procedures can be highly standardized but require manual execution. Similar to other references, [12] mentions that 'results rely on the doctors' skill and experience'. [13] states that 3D US volumes can be analyzed with a standard workflow after acquisition. I suggest that the authors double-check these references. [14] does not mention standardized procedures. I suggest that the authors remove these references and add fitting ones.

Response to comment

A2) We would like to thank the reviewer for raising the question. We are very sorry that the reference mentioned here is incorrect.

- [9] mentions high variability due to manual operation but does not discuss standardized procedures. Therefore, we place the reference [9] in the following text.

1) Introduction in manuscript (page 1)

Previous texts

As these requirements cannot always be fulfilled, US examination produces inconsistency and unreliability results due to insufficient standardized procedure [9, 11-14].

Revised texts

However, US diagnostic qualities are heavily relied on the experience and techniques of the doctors [7 - 9]

- Following the reviewer's opinion, we carefully checked the above references, and these references are not appropriate here, so we have deleted the references [11-14] from the original manuscript. Then, we added references [11, 12] to support the statement in the paper that US examination produces inconsistency and unreliability results due to insufficient standardized procedure. The relevant description in reference [11, 12] is as follows:

“The mathematical model applies only to clear fluids contents, and this mathematical calculation of gastric fluid volume as well as the qualitative analysis of fluid contents requires turning the patient in the right lateral decubitus position, what was not feasible in a quarter of emergency patients in a previous study.”

“Neck US results are highly dependent on equipment, scanning protocol and training.

Optimal imaging requires equipment that meets specific technical standards and a trained operator working within a thyroid team.”

2) Introduction in manuscript (page 1)

Previous texts

As these requirements cannot always be fulfilled, US examination produces inconsistency and unreliability results due to insufficient standardized procedure [9, 11-14].

Revised texts

Furthermore, the standard practice of having patients assume a supine position and maintain stillness during examinations can present challenges, as these requirements may not always be met [11]. Consequently, operator-dependency and patient-specific factors introduce elements of inconsistency and unreliability into US examinations [12].

Previous references

[9] Jiang, Z., Grimm, M., Zhou, M., Esteban, J., Simson, W., Zahnd, G., Navab, N.: Automatic normal positioning of robotic ultrasound probe based only on confidence map optimization and force measurement. *IEEE Robotics and Automation Letters* 5(2), 1342–1349 (2020).

[11] Graumann, C., Fuerst, B., Hennersperger, C., Bork, F., Navab, N.: Robotic ultrasound trajectory planning for volume of interest coverage. In: 2016 IEEE International Conference on Robotics and Automation (ICRA), pp. 736–741 (2016).

[12] Mustafa, A.S.B., Ishii, T., Matsunaga, Y., Nakadate, R., Ishii, H., Ogawa, K., Saito, A., Sugawara, M., Niki, K., Takanishi, A.: Development of robotic system for autonomous liver screening using ultrasound scan ning device. In: 2013 IEEE International Conference on Robotics and Biomimetics (ROBIO), pp. 804–809 (2013).

[13] Chan, L., Fung, T., Leung, T., Sahota, D., Lau, T.: Volumetric (3d) imaging reduces inter-and intraobserver variation of fetal biometry measurements. *Ultrasound in Obstetrics and Gynecology: The Official Journal of the International Society of Ultrasound in Obstetrics and Gynecology* 33(4), 447–452 (2009) .

[14] Tsumura, R., Iwata, H.: Robotic fetal ultrasonography platform with a passive scan mechanism. *International Journal of Computer Assisted Radiology and Surgery* 15(8), 1323–1333 (2020) .

Newly added references

- [11] Bouvet, L., Chassard, D.: Ultrasound assessment of gastric contents in emergency patients examined in the full supine position: an appropriate composite ultrasound grading scale can finally be proposed. Springer(2020).
- [12] Leenhardt, L., Erdogan, M., Hegedus, L., Mandel, S., Paschke, R., Rago, T., Russ, G.: 2013 european thyroid association guidelines for cervical ultrasound scan and ultrasound-guided techniques in the postoperative management of patients with thyroid cancer. European thyroid journal 2(3), 147–159 (2013).

Q3) Page 2: Reference 30 is misplaced. This work does not require experts to design the acquisition procedure.

Response to comment

A3) We are very sorry that reference 30 was misplaced but has been corrected. The details as follows:

1) Introduction in manuscript (page 2)

Previous texts

Most existing autonomous systems are still at level 2 [27–30], which rely on an expert to design the acquisition procedure.

[30] Zielke, J., Eilers, C., Busam, B., Weber, W., Navab, N., Wendler, T.:Rsv: Robotic sonography for thyroid volumetry. IEEE Robotics and Automation Letters 7(2), 3342–3348 (2022).

Revised texts

In an effort to apply online image-guided methods to define the scanning trajectory and derive clinically relevant information out of the 3D reconstructed image, **Zielke et al. [34] implemented in-plane navigation** specifically designed for robotic sonography in thyroid volumetry. **However, in actual clinical practice, sonographers often use a combination of multiple views, such as transverse and longitudinal scans**, for the identification and diagnosis of both benign and malignant pathologies [35].

[34] Zielke, J., Eilers, C., Busam, B., Weber, W., Navab, N., Wendler, T.:Rsv: Robotic sonography for thyroid volumetry. IEEE Robotics and Automation Letters

7(2), 3342–3348 (2022).

[35] Kharchenko, V.P., Kotlyarov, P.M., Mogutov, M.S., Alexandrov, Y.K., Sencha, A.N., Patrunov, Y.N., Belyaev, D.V.: *Ultrasound Diagnostics of Thyroid Diseases*. Springer, New York (2010).

Q4) Page 6: 'Unlike other examinations with less deformation, the optimal probe orientation for soft tissue (eg, thyroid) scanning is not always normal [40].' Here the reference is misplaced because [40] proposes a system for normal probe positioning.

Response to comment

A4) We would like to thank the reviewer for raising the question. The reference [40] is not really appropriate here, so we have deleted the reference [40]. Because the original sentence is not clearly described, it will cause confusion to the reviewer. We have revised the sentence as follows:

1) Result in manuscript (page 12)

Previous texts

Unlike other examinations with less deformation, the optimal probe orientation for soft tissue (eg, thyroid) scanning is not always normal [40].

Revised texts

However, taking into account the limited resolution of the Kinect and potential participant movements during the scan, the pre-estimated normal vector cannot be directly applied as the normal vector for subsequent scans. To tackle this problem, Bayesian optimization algorithm with image entropy as the loss function was used to obtain a better probe orientation with very few adjustments.

Conclusion:

Based on the mentioned points, I suggest the paper undergoes a major revision.

Response to comment

We thank the reviewer for the positive evaluation and valuable comments on our manuscript.

Reviewer #2

A. Summary of the key results

The authors proposed a fully autonomous robotic ultrasound system for thyroid scanning, which using ultrasound images for real-time thyroid nodule detection, force feedback and depth image for robot motion control. They compared the system with manual scans on human participants, and also evaluate participants' satisfaction.

This work completed the clinical application of fully automated robotic ultrasound systems and achieve human-like fusion of ultrasound images, visual and force information. However, the innovation in this work is not apparent, it lies more in the integration of existing work into a cohesive system.

Response to comment

We thank the reviewer for raising this critical issue to make the innovation of our work more clear. In the revised version of the manuscript, we provide a new framework and a re-summary of the innovations rather than just a system that integrates existing work.

2) Introduction in manuscript (page 3)

Previous texts

In order to eliminate these obstacles, we designed a fully autonomous robotic ultrasound system (FARUS), as shown in Fig. 1. To the best of our knowledge, this is the first fully autonomously ultrasound robotic system for thyroid scanning. As our first contribution, we used an autonomous robot to perform an enhanced thyroid scan. Necessary fundamental autonomous implementations are listed below. 1) General scanning range determination based on CT thyroid image data. 2) Dynamic path planning based on image control and force control. 3) Probe orientation adjustment based on Bayesian optimization. 4) Central region position tracking based on image segmentation. 5) Possible nodular lesion detection in ultrasound images based on object detection. In the new ultrasound scanning setting, the examination procedure has outstanding isolation and excellent stability, which can assist and replace the work of medical staff in dangerous environments such as high contagion, with great potential to serve the great needs in ultrasound screening in pandemic crisis, such as COVID-19.

The second contribution of this work is the ability to implement autonomous scanning strategies under the soft tissue, which presents various technical and workflow challenges. Unlike other examinations with less deformation such as orthopedics [40,41], the optimal orientation for soft tissue is not necessarily the normal.

Ultrasonography of the thyroid presents multiple challenges—including the small size of the thyroid lobe and the need for precise coverage of the lobe; soft tissue deformation; force sensitivity; image quality optimization; and the effect of motion on trajectory planning. Here, we have developed machine learning, computer vision and advanced control technologies to meet the needs of the thyroid scanning. In this process, depending on the fusion of sensing technology and vision, we obtained high-quality ultrasound images comparable to those manually collected by physicians, achieving accurate and real-time identification of thyroid nodules.

Revised texts

In order to eliminate these obstacles, we designed a fully autonomous robotic ultrasound system (FARUS), as shown in Fig. 1. To the best of our knowledge, **this is the first-in-human study of fully autonomous robotic US scanning for thyroid. In conventional US examinations, the process involves a division of responsibilities between sonographer and radiologist. However, this FARUS system combines the expertise of both roles into a single autonomous unit.** Here, we achieved a human-like fusion of both in-plane and out-of-plane scanning, allowing for comprehensive scanning of the thyroid region, and providing a detailed evaluation of the anatomy. The system overcomes the challenge of locating thyroid targets through the implementation of a reinforcement learning strategy. It enables probe orientation optimization with Bayesian optimization. Furthermore, the FARUS system uses deep learning techniques for real-time segmentation of the thyroid gland and potential nodules. In general, this system provides a convenient screening tool integrating intelligent nodule detection, lesion localization, automatic measurement, and benign and malignant classification for thyroid nodule ultrasound, which can improve the detection rate of nodules and the ability to distinguish benign and malignant nodules and reduce the work burden for doctors.

The second contribution of **our study lies in the practical realization and clinical application of the FARUS that achieves high-quality US images in comparison with those manually collected by physicians, achieving accurate and real-time detection of thyroid nodules.** We substantiated the validity of our approach by conducting a comparative evaluation of FARUS-driven diagnostic results for thyroid nodules against the established hospital benchmark. We have conducted extensive evaluations and provided evidence of the system's performance and reliability. Our work addresses the gap between existing research and clinical application by demonstrating the successful deployment of this system in a real-world clinical setting.

Previous framework

- System design for autonomous ultrasound imaging
- Thyroid search and probe orientation optimization
- Fully autonomous in-plane ultrasound imaging
- Fully autonomous out-of-plane ultrasound imaging

Revised framework

- System design for autonomous ultrasound imaging
- Deep learning for gland and nodule segmentation
- Thyroid search and probe orientation optimization
- Fully autonomous robotic ultrasound imaging
- Thyroid nodule imaging reporting with FARUS

- We substantiated the validity of our approach by conducting a comparative evaluation of FARUS-driven diagnostic results for thyroid nodules against the established hospital benchmark, as shown in the figure below. FARUS proved to be a reliable and effective tool for nodule diagnosis.

The results of the scoring and classification process according to the ACR-TIRADS are presented in **Fig. 6**. The **Fig. 6b** shows the nodule's composition is categorized as solid, leading to a score of 2, while its echogenicity is classified as hypo, also resulting in a score of 2. The nodule demonstrates a well-defined boundary and a regular shape, contributing to a score of 0. Additionally, the comparison of height and weight exceeds 1, resulting in a score of 3. Lastly, no echogenic foci are observed within the nodule, leading to a score of 0. The total score amounts to 7, leading to the classification of the nodule as level 5 or highly suspicious. **Figure 6b** illustrates the nodule's composition as partial, leading to a score of 1, and its echogenicity as hypo, resulting in a score of 2. The nodule's characteristic features, including a clear boundary and regular shape, warrant a score of 0 points. Additionally, the comparison of height and weight yields a score of 0 as it is less than 1. Furthermore, no presence of peripheral calcification within the nodule leading to a score of 0. Consequently, the cumulative score amounts to 3, classifying the nodule as level 3 or mildly suspicious.

In this research, we introduce the FARUS method, which aims to scan, detect, and classify nodules gathered from 19 individuals. Furthermore, we compare the data of each patient with the evaluations provided by doctor. According to the doctor's evaluation, 17 individuals were found to have nodules, while 2 individuals showed no presence of nodules. Our proposed system, FARUS, successfully identified 13 individuals as having nodules and 6 individuals as not having nodules. To analyze the accuracy of FARUS, we compared the scoring and classification of 17 nodules

among the 13 individuals successfully detected by both FARUS and the doctor. Each nodule was matched to the doctor's report based on its location and shape. **Table 1** displays a comparison between FARUS and doctor scoring and the classification of thyroid nodules. Among the 17 nodules assessed, 5 were diagnosed with the same score by both the doctor and the proposed FARUS system. **Figure 6a** illustrates scoring and classification of nodule diagnosed by doctor and FARUS. This can be attributed to the ultrasound image's appropriate brightness and contrast, resulting in well-distributed pixels. Additionally, 8 nodules exhibited a score difference of 1, primarily attributed to discrepancies in echogenicity or composition classifications. Furthermore, there was one nodule with a score difference of 2 and another with a score difference of 4, caused by differing classifications in both echogenicity and composition. Moreover, one nodule displayed a score difference of 3 due to variations in shape classification.

The main reason for the different classifications by FARUS is the use of a probe different from the one used by the doctor. The doctor's probe consistently produces a specific color of the thyroid gland for each patient, while the brightness and contrast of ultrasound images produced by our probe may vary across different patients with varying ages and weights. The proposed FARUS system demonstrated adequate accuracy in classifying nodules, with 5 out of 17 cases showing complete agreement with the doctor's diagnosis. Furthermore, for the remaining nodules, the discrepancies were primarily limited to 1 score difference, mainly arising from variations in echogenicity and composition. Overall, FARUS proved to be a reliable and effective tool for nodule classification.

Newly added Table 1

Table 1 Comparison of thyroid nodule scoring and classification between FARUS and doctor

No	Name	Nodule	Diagnosed by	Size	Composition	Echogenicity	Margin	Shape	Echogenicity Foci	Total	Difference	Reason
1	PATIENT_1.L	#1	FARUS Doctor	6.1*5.3 R 3.2*2.6*2.9 R	2 2	2 2	0 0	3 3	0 0	7 7	0	Echogenicity,
2	PATIENT_2.R	#2	FARUS Doctor	2.1*3.2 4.5*3.7*3 R	2 1	2 1	0 0	0 0	0 0	4 2	2	Echogenicity,
3	PATIENT_3.R	#3	FARUS Doctor	4.6*5.8 R 6*4*7 R	0 0	0 0	0 0	0 0	0 0	0 0	0	
4	PATIENT_3.R	#4	FARUS Doctor	2.5*4.1 R 6*4.5*5.6 R	2 1	1 1	0 0	0 0	0 0	3 2	1	Composition
5	PATIENT_4.L	#5	FARUS Doctor	9.8*11.9 L 13*8*11 L	2 2	2 1	0 0	0 0	0 0	4 3	1	Echogenicity
6	PATIENT_5.L	#6	FARUS Doctor	2.6*4.2 L 3*2*2.8 L	2 0	2 0	0 0	0 0	0 0	4 0	4	Echogenicity,
7	PATIENT_5.R	#7	FARUS Doctor	3.8*4.8 R 5*1.6*7 R	2 1	1 2	0 0	0 0	0 0	3 3	0	Echogenicity,
8	PATIENT_6.L	#8	FARUS Doctor	4.9*6.2 L 5*3*6 L	1 1	2 1	0 0	0 0	0 0	3 2	1	Echogenicity
9	PATIENT_7.L	#9	FARUS Doctor	6.1*6.7 L 7*5*8 L	1 1	2 2	0 0	0 0	0 0	3 3	0	
10	PATIENT_7.R	#10	FARUS Doctor	3.8*6.2 R 6*4*7 R	2 1	2 2	0 0	0 0	0 0	4 3	1	Composition
11	PATIENT_8.R	#11	FARUS Doctor	6.5*8.6 R 7.5*5.4*8 R	1 1	2 1	0 0	0 0	0 0	3 2	1	Echogenicity
12	PATIENT_9.L	#12	FARUS Doctor	11.1*10.7 L 11*9*19 L	1 1	2 2	0 0	3 0	0 0	6 3	3	Shape
13	PATIENT_9.L	#13	FARUS Doctor	9.3*0.99 L 10*10*11 L	1 1	2 2	0 0	0 0	2 2	5 5	0	
14	PATIENT_10.R	#14	FARUS Doctor	5.1*5.2 R 8*5*8 R	1 1	2 1	0 0	0 0	0 0	3 2	1	Echogenicity
15	PATIENT_11.R	#15	FARUS Doctor	2.2*3.6 R 3*2*3 R	1 1	2 1	0 0	0 0	0 0	3 2	1	Echogenicity
16	PATIENT_12.L	#16	FARUS Doctor	8.5*11.8 L 14*11*18 L	2 2	1 1	0 0	0 0	0 0	3 3	0	
17	PATIENT_13.R	#17	FARUS Doctor	8.6*9.5 R 10*8*7.5 R	2 1	1 1	0 0	0 0	0 0	3 2	1	Composition

Newly added Figure 6

Fig. 6 Scoring and classification of thyroid nodules based on ACR-TIRADS. a Comparison of the thyroid nodules found by FARUS and doctor. **b**, Result of scoring and classification diagnosed by FARUS based on ACR-TIRADS. **c**, Correlation between numbers of pixels over brightness in thyroid image.

B. Originality and significance: if not novel, please include reference

As previously stated, this work emphasized the comprehensive character of the robotic ultrasound system. The contribution of this work to system implementation and clinical

evaluation is significant as a fully automated robotic system, but it's hard to find novelty in the methodology compared with current works. For example in path planning, Huang et al. proposed a algorithm of using depth image for path planning [1], Z. Wang et al. using different views of depth images to plan 3D scanning path[2].

Response to comment

Generally, existing systems typically rely on global information of the target tissue acquired from preoperative medical images or surface information obtained from external sensors [27, 30, 31]. Given the inherent variability in individual anatomy and the dynamic of human motion, executing a scanning task based on a predetermined trajectory presents significant challenges. To address these difficulties, Z. Jiang et al. [32] integrated the feedback of segmented images into the control process. J. Zhan et al. [33] have proposed a visual servoing framework for motion compensation. **However, these methods usually assume that the target features exist in the ultrasound image, once the features are lost, the control methods may fail.**

In an effort to apply online image-guided methods to define the scanning trajectory and derive clinically relevant information out of the 3D reconstructed image, Zielke et al. [34] implemented in-plane navigation specifically designed for robotic sonography in thyroid volumetry. **However, in actual clinical practice, sonographers often use a combination of multiple views, such as transverse and longitudinal scans, for the identification and diagnosis of both benign and malignant pathologies [35].**

Newly added references

[27] Huang, Q., Wu, B., Lan, J., Li, X.: Fully automatic three-dimensional ultrasound imaging based on conventional b-scan. IEEE transactions on biomedical circuits and systems 12(2), 426–436 (2018).

[30] Wang, Z., Zhao, B., Zhang, P., Yao, L., Wang, Q., Li, B., Meng, M.Q.-H., Hu, Y.: Full-coverage path planning and stable interaction control for automated robotic breast ultrasound scanning. IEEE Transactions on Industrial Electronics 70(7), 7051–7061 (2023). <https://doi.org/10.1109/TIE.2022.3204967>.

[31] Suligoj, F., Heunis, C.M., Sikorski, J., Misra, S.: Robust—an autonomous robotic ultrasound system for medical imaging. IEEE Access 9, 67456–67465 (2021). <https://doi.org/10.1109/ACCESS.2021.3077037>.

[32] Jiang, Z., Li, Z., Grimm, M., Zhou, M., Esposito, M., Wein, W., Stechele, W., Wendler, T., Navab, N.: Autonomous robotic screening of tubular structures based only on real-time ultrasound imaging feedback. IEEE Transactions on Industrial Electronics 69(7), 7064–7075 (2022). <https://doi.org/10.1109/TIE.2021.3095787>.

- [33] Zhan, J., Cartucho, J., Giannarou, S.: Autonomous tissue scanning under free-form motion for intraoperative tissue characterisation. In: 2020 IEEE International Conference on Robotics and Automation (ICRA), pp.11147–11154 (2020). IEEE.
- [34] Zielke, J., Eilers, C., Busam, B., Weber, W., Navab, N., Wendler, T.: Rsv: Robotic sonography for thyroid volumetry. *IEEE Robotics and Automation Letters* 7(2), 3342–3348 (2022).
- [35] Kharchenko, V.P., Kotlyarov, P.M., Mogutov, M.S., Alexandrov, Y.K., Sencha, A.N., Patrunov, Y.N., Belyaev, D.V.: *Ultrasound Diagnostics of Thyroid Diseases*. Springer, New York (2010).

In this paper, we employ reinforcement learning techniques to dynamically adjust the robotic arm's position, facilitating the attainment of an optimal horizontal alignment for capturing thyroid images. Motivation for Reinforcement Learning (RL) lies in its ability to enable machines to learn and make decisions in dynamic, uncertain environments without the need for labeled datasets.

Revised texts

An important problem in robotic thyroid scanning is locating the thyroid gland on the body surface. We present a coarse-to-fine approach to locate the thyroid. In the coarse estimation step, the region of neck is identified by human skeleton keypoints. Notably, the thyroid lobe may not be visible based on the location predicted by skeletal data (**Fig. 2b**), because the anatomy of the neck varies greatly in different populations [43]. Consequently, we added the fine-tuning step to further locate the thyroid lobe. To enable robotic scan in such a case, we used reinforcement learning to determine thyroid location.

Figure 3 shows the training process of DQN learning with panoramic environment. The process starts from sequences of data are collected and labelled manually. We labeled each sequence of thyroid images a goal or an ideal position for model to learn as shown in **Fig 3a**. after that each of sequence of image will be aligned and attached to be a panorama as shown in **Fig 3b**. Then a bunch sequence of image will be randomized and generated in to panoramas as shown in **Fig 3c-d**. there is a sliding window will slide according to the action given by the agent. To mimic the real environment when sometimes the probe is not attached fully, we combine simulated view with the random noise as shadow mask to produced imperfect thyroid image that mimic image that is produced when the probe is not fully attached as shown in **Fig 3e**. **Figure 3f** presents the results of the training evaluation for reinforcement learning. During the exploration stage, which comprises the initial 30,000 steps. **Fig. 3g** illustrates a progressive increase in the mean reward, which indicates the gradual enhancement of RL model's performance in the given task as it continues to learn over time. **Figure 3h** depicts the process of thyroid scanning facilitated by the DQN Learning model in the context of FARUS. The

DQN Learning model enables the robotic arm to execute movements based on the input received from ultrasound images. As illustrated in **Fig 3h**, the model accurately predicts the required movements to guide the robotic arm effectively.

In the initial and fourth images, the probe is initially attached to the patient's neck. Subsequently, the model predicts the appropriate directions for the robotic arm to move in each frame. The blue bar represents the model's predictions for movement to attain the ideal position. The green bar signifies the model's prediction to maintain a stationary position once the ideal position is reached. Conversely, the yellow bar represents the model's predictions when the patient is already in motion. The DQN Learning model instructs the robotic arm to move right based on the first thyroid image until the arm reaches the position displaying the ideal thyroid image, as shown in the second thyroid image. Notably, our proposed FARUS can effectively guide the robotic arm even when the patient is in motion. For instance, at 6.3 seconds, when the patient's neck moves left, the model adjusts the robotic arm's position accordingly, resulting in the leftward movement depicted in the third thyroid image. In the fourth thyroid image, the robotic arm is attached to the mid-neck region, which is not the correct location for detecting the presence of thyroid in the neck. The model guides the robotic arm to move left until it reaches the ideal position. Moreover, even in the absence of thyroid presence in the fifth image, the model has learned to predict the ideal position, leading the robotic arm to halt at this position. The DQN Learning model in FARUS showcases its capability to accurately guide the robotic arm during thyroid scanning, even in the presence of patient movement or the absence of thyroid.

Newly added Figure 3

Fig. 3 Training process of DQN learning in panoramic thyroid environment. a A sequence of image is collected and labeled manually. **b**, The alignment of sequence image into panorama. **c**, Blending images in a random sequence. **d**, Generated panorama. **e**, The simulation of panorama-base sliding window. **f**, Average reward vs step. **g**, Average episode length vs step. **h**, Trained model's prediction vs time.

C. Data & methodology: validity of approach, quality of data, quality of

presentation

The authors evaluated the system using reliable experimental data and presented the results effectively. However, the validity of some experimental data needs further confirmation. As the authors claim, they used Kinect Azure DK RGBD camera to reconstructing the point cloud surface of the human neck and calculated the initial orientation of the probe for robotic scanning. However, the accuracy of this camera is difficult to calculate the exact normal vector, especially when the surface of the human neck is covered with enough ultrasound gel. Besides, the transparent and reflective properties of ultrasound gels can affect the work of depth cameras. Also, the movement of the volunteers will change the initial point based the previous data, so dynamic identification and tracking is required here.

Response to comment

First of all, we agree with the author that the accuracy of the Kinect Azure DK RGBD camera is difficult to calculate an accurate normal vector, especially when the surface of the human neck is covered with enough ultrasonic gel when scanning.

Secondly, the normal vector is not accurate for the following reasons:

- The camera hardware precision is not high. Therefore, in the rough estimation step, it is inaccurate to identify the neck region by the skeleton key points and then determine the probe direction by the point cloud normal vector.
- Solving the normal vector before the probe touches the person may be inconsistent with the normal vector when the probe touches the person, because the person may move before the probe touches the person.

Therefore, the above reasons cause the probe position is not accurate, the Angle is not accurate.

Finally, the solution is as follows:

- For inaccurate position: We use reinforcement learning to coarsely locate the thyroid in the absence of visible thyroid through intelligent path planning. Then, we add fine-tuning steps, using image segmentation to guide the probe to the exact location of the thyroid.
- For the Angle is not accurate, the Bayesian optimization method is used to adjust the probe Angle, so as to adjust the normal vector of ultrasonic scanning.

D. Appropriate use of statistics and treatment of uncertainties

Yes, they did. The authors obtained the head and neck CT images from the Southern Hospital of Southern Medical University for the estimation of the thyroid scanning range. Their samples represented all age group populations and covered people with body mass index (BMI) ranging from 17.1 to 27.8. Also, to build the AI models for thyroid segmentation and nodule detection, they collected 118,137 thyroid images manually from the volunteers.

Response to comment

Thank the reviewers for their recognition and affirmation of our work.

In the current trial, with the approval of the ethical review committee, we recruited 19 patients. First of all, we scanned both sides of the thyroid gland of 19 participants with FARUS, and at the same time, these participants underwent nodule diagnosis at The Third Affiliated Hospital of Sun Yat-Sen University Lingnan Hospital(<https://lnyy.zssy.com.cn/>) to verify the robotic scans with the classification of benign and malignant nodules.

Importantly, the system has been tested in clinical trials with actual patients.

E. Conclusions: robustness, validity, reliability

This work evaluated the time spent and participants' satisfaction of the system and ultrasound specialists in the scanning and diagnostic process. However, as they mentioned, the NDT: Nodule detection time of the system is a hundred times faster than the doctors' time cost, which is very counterintuitive. And they didn't mention how to identify the time cost of nodule detection for doctors in this paper. This comparison is not reliable.

Response to comment

In this paper, we took the time when the doctor detected the nodule and stayed in this frame (the nodule position did not change, as shown in the video sequence below) as the time for the doctor to detect the nodule, considering that the doctor needs to stay in this position to check the nodule boundary, size and other information. At present, it seems that the index of nodule detection time may be inaccurate and inappropriate to compare robots and doctors, and we have deleted this index in the revised manuscript.

F. Suggested improvements: experiments, data for possible revision

The authors had done necessary experiments in this paper. However, as they mentioned in the discussion, the contact force is different between groups of people. As the imaging quality of ultrasound images is influenced by the contact force, they should consider the robustness of ultrasound image processing algorithms under varying contact forces.

Response to comment

The issue mentioned by the reviewer about the robustness of ultrasonic image processing algorithms under different contact forces has been discussed in Figure 4 of the revised manuscript, with specific details as follows:

Revised Figure 4

Revised texts

In many instances, Bayesian optimization (BO) outperforms expert as well as other state-of-the-art global optimization algorithms. Bayesian optimization constructs a surrogate model for the objective function and quantifies the uncertainty through Bayesian inference. This surrogate model determines where the next candidate will be, as in **Fig. 4a**. During the BO phase, the position of

the US probe remains constant; therefore, probe orientation and contact force are the two major factors in the entropy value of the US image. To further explore the influence of contact force on image quality, we conducted an investigation involving 9 participants. Each participant was instructed to apply varying levels of force on the US probe, positioned at the robot's end, while ensuring safety under continuous manual monitoring. **Fig. 4 e** demonstrates that when the contact force between the probe and the skin exceeds 2 N, the median of the image entropy value remains stable. Taking scanning comfort and safety into account, we set the maximum contact force at 4 N. We consider a limited budget of $N = 5$ iterations to speed up the BO phase. The entropy of the US image varies from 7.172 to 7.173 in 5 iterations, and the max entropy corresponding to the optimal orientation was reached at the second iteration (**Fig. 4c**). When the number of iterations reached 5, a higher drop in performance was observed due to its exploration nature (**Fig. 4c**). The significant differences in image entropy values were observed after the BO phase for 89 participants (**Fig. 4b**). During this experiment, the probe was initially set at a relatively optimal angle, resulting in a less noticeable entropy increase before and after Bayesian optimization. However, between 13.4 s and 21.3 s, the US probe underwent a 10-degree angle adjustment, revealing that the entropy value of the US image responded more sensitively to angle changes compared to image confidence. It was speculated that this sensitivity could be attributed to the level of contact between the skin and the probe. As illustrated in **Fig. 4e**, there is a positive relationship between the image entropy and confidence map that evaluates the contact condition at each pixel [46]. As the entropy value consumes less computation, it allows for a real-time control of image quality.

G. References: appropriate credit to previous work?

The authors focus more on work related to fully automated medical robots and ignored the few recent advances in the area of fully automatic robotic ultrasound systems, such as Z. Wang et al.[2], Z. Jiang et al.[3], J. Zhan et al.[4] , F. Suligoj et al.[5].

[1] Q. Huang, B. Wu, J. Lan, and X. Li, "Fully automatic three-dimensional ultrasound imaging based on conventional B-scan," *IEEE Trans. Biomed. Circuits Syst.*, vol. 12, no. 2, pp. 426–436, Apr. 2018.

[2] Z. Wang et al., "Full Coverage Path Planning and Stable Interaction Control for Automated Robotic Breast Ultrasound Scanning," *IEEE Trans. Ind. Electron.*, pp. 1–10, 2022.

[3] Z. Jiang et al., "Autonomous robotic screening of tubular structures based only on real-time ultrasound imaging feedback", *IEEE Trans. Ind. Electron.*, 2021.

[4] J. Zhan, J. Cartucho and S. Giannarou, "Autonomous tissue scanning under free-form motion for intraoperative tissue characterisation", *Proc. IEEE Int. Conf. Robot. Autom. (ICRA)*, pp. 11147-11154, 2020.

[5] F. Suligoj, C. M. Heunis, J. Sikorski, and S. Misra, "RobUS_t–An autonomous

robotic ultrasound system for medical imaging,” IEEE Access, vol. 9, pp. 67456–67465, May. 2021.

Response to comment

We appreciate the few recent advances in the field of fully automatic robotic ultrasound systems mentioned by reviewers, such as Wang Z. et al.[2], Jiang Z. et al.[3], J. Zhan et al.[4], F. Suligoj et al.[5]. In the revised version of the manuscript, these works have been added to the Introduction section.

2) Introduction in manuscript (page 2)

Revised texts

With the improvement of the autonomy of ultrasonic inspection of robots, substantial advancements in the field have been achieved, providing promising solutions to improve the accuracy and efficiency of US procedures. The prerequisite for robotic ultrasound acquisitions is to plan the scanning path to ensure the finding a desired imaging plane or covering a selected region of interest. Generally, existing systems typically rely on global information of the target tissue acquired from preoperative medical images or surface information obtained from external sensors [27, 30, 31]. Given the inherent variability in individual anatomy and the dynamic of human motion, executing a scanning task based on a predetermined trajectory presents significant challenges. To address these difficulties, Z. Jiang et al. [32] integrated the feedback of segmented images into the control process. J. Zhan et al. [33] have proposed a visual servoing framework for motion compensation. However, these methods usually assume that the target features exist in the ultrasound image, once the features are lost, the control methods may fail. In an effort to apply online image-guided methods to define the scanning trajectory and derive clinically relevant information out of the 3D reconstructed image, Zielke et al. [34] implemented in-plane navigation specifically designed for robotic sonography in thyroid volumetry. However, in actual clinical practice, sonographers often use a combination of multiple views, such as transverse and longitudinal scans, for the identification and diagnosis of both benign and malignant pathologies [35].

Newly added references

- [27] Q. Huang, B. Wu, J. Lan, and X. Li, “Fully automatic three-dimensional ultrasound imaging based on conventional B-scan,” IEEE Trans. Biomed. Circuits Syst., vol. 12, no. 2, pp. 426–436, Apr. 2018.
- [30] Z. Wang et al., “Full Coverage Path Planning and Stable Interaction Control for Automated Robotic Breast Ultrasound Scanning,” IEEE Trans. Ind. Electron., pp. 1–10, 2022.

[32] F. Suligoj, C. M. Heunis, J. Sikorski, and S. Misra, "RobUSt—An autonomous robotic ultrasound system for medical imaging," IEEE Access, vol. 9, pp. 67456–67465, May. 2021.

[33] Z. Jiang et al., "Autonomous robotic screening of tubular structures based only on real-time ultrasound imaging feedback", IEEE Trans. Ind. Electron., 2021.

[34] J. Zhan, J. Cartucho and S. Giannarou, "Autonomous tissue scanning under free-form motion for intraoperative tissue characterisation", Proc. IEEE Int. Conf. Robot. Autom. (ICRA), pp. 11147-11154, 2020.

H. Clarity and context: lucidity of abstract/summary, appropriateness of abstract, introduction and conclusions

The authors provided a detailed description of the experimental equipment, the assembly method, and the performance of the AI model. This paper had clear summary and reasonable conclusions. However, they described the evolution of automated medical robotic systems in detail such as Da Vinci Surgical System, but there is little background on robotic ultrasound systems. Besides, a sweep can take up to 4-5 minutes as shown in this paper, making it challenging to guarantee that the participants don't move. The authors didn't go into detail about how the system guarantees safety after detecting movement and how it continues to complete the sweep. This limitation needs to be mentioned in the discussion.

Response to comment

- We thank the reviewer for the insightful comment. As suggested by the reviewer, we have added details to the background of the robotic ultrasound system, as follows:

1) Introduction in manuscript (page 2)

Revised texts

Under the concept of the autonomy level of medical robots (Level 0-5), the autonomy level of ultrasonic inspection robots (Level 0-3) has been defined at present. Level 0 is defined as "manual probe manipulation", the proposed tele-echography systems [15–19]. The level 1 ultrasound robotic system utilizes visual servo technology to allow the robot to automatically track desired image features [20–23] and compensate for unnecessary patient movement during remote operation. Level 2 ultrasound robotic system is described as performing autonomous ultrasound acquisition along a manually planned path [24, 25]. The ultrasound robotic system in level 3 that can autonomously plan and perform ultrasound acquisition without any instruction from a human operator but requires

the supervision of an operator [26–29].

Newly added references

- [17] Wu, S., Wu, D., Ye, R., Li, K., Lu, Y., Xu, J., Xiong, L., Zhao, Y., Cui, A., Li, Y., Peng, C., Lv, F.: Pilot study of robot-assisted teleultrasound based on 5g network: A new feasible strategy for early imaging assessment during covid-19 pandemic. *IEEE Transactions on Ultrasonics, Ferroelectrics, and Frequency Control* 67(11), 2241–2248 (2020).
- [22] Fu, Y., Lin, W., Yu, X., Rodríguez-Andina, J.J., Gao, H.: Robot-assisted teleoperation ultrasound system based on fusion of augmented reality and predictive force. *IEEE Transactions on Industrial Electronics* 70(7), 7449–7456 (2023).
- [23] Martin, S.B.N.J.C.e.a. J.W.: Enabling the future of colonoscopy with intelligent and autonomous magnetic manipulation. *Nature Machine Intelligence* (2), 595–606 (2020).
- [28] Tan, J., Li, B., Leng, Y., Li, Y., Peng, J., Wu, J., Luo, B., Chen, X., Rong, Y., Fu, C.: Fully automatic dual-probe lung ultrasound scanning robot for screening triage. *IEEE Transactions on Ultrasonics, Ferroelectrics, and Frequency Control*, 1–1 (2022).
- [29] Li, K., Li, A., Xu, Y., Xiong, H., Meng, M.Q.-H.: Rl-tee: Autonomous probe guidance for transesophageal echocardiography based on attention augmented deep reinforcement learning. *IEEE Transactions on Automation Science and Engineering*, 1–13 (2023).

- In the context of limited participant movement, our FARUS system exhibits the ability to effectively compensate for such minor motion through a combination of image control, force control, and reinforcement learning. However, should the range of movement exceed acceptable limits, scanning operations will be terminated as a precautionary safety measure. Specifically, if FARUS detects participant movement beyond 2 cm to the left or right, or 5 cm forward or backward within one second, the scan will be halted promptly.
-

Reviewer #3

In this research, they provided a novel fully autonomous robotic ultrasound system (FARUS) for thyroid scanning on humans. But I still have some questions:

1. I would like to know the comparison of the size, shape, localization, calcification, and border of the nodules detected by FARUS and clinicians. Is FARUS better at detecting some specific nodules or all nodules?

Response to comment

We express our gratitude to the reviewer for inquiring about the performance of our proposed Fully Automated Robotic Ultrasound System (FARUS) in comparison to clinicians for detecting nodules.

We would like to reiterate the challenges faced by FARUS, which consists of three key stages: scanning, detection, and classification, each contributing significantly to the overall performance of the system. The primary limitation affecting FARUS's ability to effectively scan nodules stems from the disparity between its scanning methods and those employed by medical professionals. Unlike doctors, the robotic arm lacks the flexibility to perform scans from multiple angles, which may impact the thoroughness of the examination.

Moreover, a notable difference arises in the patient's positioning during the scanning process. While doctors typically scan patients in a supine position, FARUS scans patients in an upright sitting position. Scanning the thyroid in a supine position offers the advantage of ensuring relaxation of the thyroid and neck skin surface, facilitating more accessible and accurate scanning. However, employing the robotic arm for scanning patients in a lying-down position is not recommended due to safety considerations.

The detection and classification stages are significantly influenced by the quality of the probe being used. The images produced by our probe exhibit inconsistencies when compared to those produced by a doctor's probe. These inconsistencies are evident in terms of brightness and contrast in the images generated by our probe. Consequently, it is not appropriate to directly compare our proposed FARUS with the results obtained by clinicians. However, for relative comparison, our proposed FARUS exhibits better performance in detecting cystic, hypo-echogenic, and hyper-echogenic nodules with distinct contrast compared to the surrounding gland.

The size of the nodule affects the performance of FARUS. Smaller nodules are more challenging for FARUS to detect accurately. Moreover, FARUS faces increased difficulty in detecting nodules located near the carotid artery or the side

area of thyroid gland or, particularly when these nodules are hypogenic or cystic and exhibit higher heterogeneity that may be indistinct from the carotid artery. Our FARUS system demonstrates a comparable ability to clinicians in detecting variations in shape, calcification, or border characteristics of nodules."

2. As we all know, identifying malignant nodules is of great value for clinicians. Thyroid Imaging Reporting and Data Systems (TIRADS) is a 5-point classification to determine the risk of cancer in thyroid nodules based on ultrasound characteristics. Could FARUS assess the malignant risk of the nodules according to TIRADS?

Response to comment

We express our gratitude to the reviewer for their valuable comment. Through the utilization of our FARUS system, we are poised to provide significant contributions to the medical community, specifically by aiding clinicians in accurately determining the risk of cancer in thyroid nodules based on ultrasound characteristics. The FARUS system has the capability to assess the malignant risk of these nodules in accordance with the Thyroid Imaging Reporting and Data System (TIRADS) guidelines.

We substantiated the validity of our approach by conducting a comparative evaluation of FARUS-driven diagnostic results for thyroid nodules against the established hospital benchmark, as shown in the figure below. FARUS proved to be a reliable and effective tool for nodule diagnosis.

The results of the scoring and classification process according to the ACR-TIRADS are presented in **Fig. 6**. The **Fig. 6b** shows the nodule's composition is categorized as solid, leading to a score of 2, while its echogenicity is classified as hypo, also resulting in a score of 2. The nodule demonstrates a well-defined boundary and a regular shape, contributing to a score of 0. Additionally, the comparison of height and weight exceeds 1, resulting in a score of 3. Lastly, no echogenic foci are observed within the nodule, leading to a score of 0. The total score amounts to 7, leading to the classification of the nodule as level 5 or highly suspicious. **Figure 6b** illustrates the nodule's composition as partial, leading to a score of 1, and its echogenicity as hypo, resulting in a score of 2. The nodule's characteristic features, including a clear boundary and regular shape, warrant a score of 0 points. Additionally, the comparison of height and weight yields a score of 0 as it is less than 1. Furthermore, no presence of peripheral calcification within the nodule leading to a score of 0. Consequently, the cumulative score amounts to 3, classifying the nodule as level 3 or mildly suspicious.

In this research, we introduce the FARUS method, which aims to scan, detect, and classify nodules gathered from 19 individuals. Furthermore, we compare the data of

each patient with the evaluations provided by doctor. According to the doctor's evaluation, 17 individuals were found to have nodules, while 2 individuals showed no presence of nodules. Our proposed system, FARUS, successfully identified 13 individuals as having nodules and 6 individuals as not having nodules. To analyze the accuracy of FARUS, we compared the scoring and classification of 17 nodules among the 13 individuals successfully detected by both FARUS and the doctor. Each nodule was matched to the doctor's report based on its location and shape. **Table 1** displays a comparison between FARUS and doctor scoring and the classification of thyroid nodules. Among the 17 nodules assessed, 5 were diagnosed with the same score by both the doctor and the proposed FARUS system. **Figure 6a** illustrates scoring and classification of nodule diagnosed by doctor and FARUS. This can be attributed to the ultrasound image's appropriate brightness and contrast, resulting in well-distributed pixels. Additionally, 8 nodules exhibited a score difference of 1, primarily attributed to discrepancies in echogenicity or composition classifications. Furthermore, there was one nodule with a score difference of 2 and another with a score difference of 4, caused by differing classifications in both echogenicity and composition. Moreover, one nodule displayed a score difference of 3 due to variations in shape classification.

The main reason for the different classifications by FARUS is the use of a probe different from the one used by the doctor. The doctor's probe consistently produces a specific color of the thyroid gland for each patient, while the brightness and contrast of ultrasound images produced by our probe may vary across different patients with varying ages and weights. The proposed FARUS system demonstrated adequate accuracy in classifying nodules, with 5 out of 17 cases showing complete agreement with the doctor's diagnosis. Furthermore, for the remaining nodules, the discrepancies were primarily limited to 1 score difference, mainly arising from variations in echogenicity and composition. Overall, FARUS proved to be a reliable and effective tool for nodule classification.

Newly added Table 1

Table 1 Comparison of thyroid nodule scoring and classification between FARUS and doctor

No	Name	Nodule	Diagnosed by	Size	Composition	Echogenicity	Margin	Shape	Echogenicity Foci	Total	Difference	Reason
1	PATIENT_1L	#1	FARUS Doctor	6.1*5.3 R 3.2*2.6*2.9 R	2 2	2 2	0 0	3 3	0 0	7 7	0	Echogenicity,
2	PATIENT_2R	#2	FARUS Doctor	2.1*3.2 4.5*3.7*3 R	2 1	2 1	0 0	0 0	0 0	4 2	2	Echogenicity,
3	PATIENT_3R	#3	FARUS Doctor	4.6*5.8 R 6*4*7 R	0 0	0 0	0 0	0 0	0 0	0 0	0	
4	PATIENT_3R	#4	FARUS Doctor	2.5*4.1 R 6*4.5*5.6 R	2 1	1 1	0 0	0 0	0 0	3 2	1	Composition
5	PATIENT_4L	#5	FARUS Doctor	9.8*11.9 L 13*8*11 L	2 2	2 1	0 0	0 0	0 0	4 3	1	Echogenicity
6	PATIENT_5L	#6	FARUS Doctor	2.6*4.2 L 3*2*2.8 L	2 0	2 0	0 0	0 0	0 0	4 0	4	Echogenicity,
7	PATIENT_5R	#7	FARUS Doctor	3.8*4.8 R 5*1.6*7 R	2 1	1 2	0 0	0 0	0 0	3 3	0	Echogenicity,
8	PATIENT_6L	#8	FARUS Doctor	4.9*6.2 L 5*3*6 L	1 1	2 1	0 0	0 0	0 0	3 2	1	Echogenicity
9	PATIENT_7L	#9	FARUS Doctor	6.1*6.7 L 7*5*8 L	1 1	2 2	0 0	0 0	0 0	3 3	0	
10	PATIENT_7R	#10	FARUS Doctor	3.8*6.2 R 6*4*7 R	2 1	2 2	0 0	0 0	0 0	4 3	1	Composition
11	PATIENT_8R	#11	FARUS Doctor	6.5*8.6 R 7.5*5.4*8 R	1 1	2 1	0 0	0 0	0 0	3 2	1	Echogenicity
12	PATIENT_9L	#12	FARUS Doctor	11.1*10.7 L 11*9*19 L	1 1	2 2	0 0	3 0	0 0	6 3	3	Shape
13	PATIENT_9L	#13	FARUS Doctor	9.3*0.99 L 10*10*11 L	1 1	2 2	0 0	0 0	2 2	5 5	0	
14	PATIENT_10R	#14	FARUS Doctor	5.1*5.2 R 8*5*8 R	1 1	2 1	0 0	0 0	0 0	3 2	1	Echogenicity
15	PATIENT_11R	#15	FARUS Doctor	2.2*3.6 R 3*2*3 R	1 1	2 1	0 0	0 0	0 0	3 2	1	Echogenicity
16	PATIENT_12L	#16	FARUS Doctor	8.5*11.8 L 14*11*18 L	2 2	1 1	0 0	0 0	0 0	3 3	0	
17	PATIENT_13R	#17	FARUS Doctor	8.6*9.5 R 10*8*7.5 R	2 1	1 1	0 0	0 0	0 0	3 2	1	Composition

Newly added Figure 6

Fig. 6 Scoring and classification of thyroid nodules based on ACR-TIRADS. a Comparison of the thyroid nodules found by FARUS and doctor. **b**, Result of scoring and classification diagnosed by FARUS based on ACR-TIRADS. **c**, Correlation between numbers of pixels over brightness in thyroid image.

REVIEWER COMMENTS

Reviewer #1 (Remarks to the Author):

- The authors changed some of the contributions and focused on nodule diagnosis (including TIRADS score). This has improved the paper.
- The authors answered sufficiently to most of reviewer's comments.
- They extended the study, now also including patients. This is definitely valuable and more interesting.
- They explained their reasoning for having volunteers sit and detect patient movement. However, it is not clear which images they use or discard while they detect patient movement. Please explain this more clearly within the paper. This needs to be done during the minor revision.
- They now mention literature correctly in the related work. However, they still use some approaches for example in plane scanning and do not refer from which works in literature these are inspired by for the robot scanning methodology part. Please make sure that such inspirations are clearly mentioned and cited in the method section. These need to be clear and will not reduce the impact of the paper, which is on proving its hypothesis, but providing the citations at the right place are needed from an academic perspective. This needs to be done during the minor revision.
- The writing and explanations have improved but the authors should do another grammar and spelling check. This needs to be done during the minor revision.

Reviewer #5 (Remarks to the Author):

The authors have addressed comments of both reviewers properly.

The softened the language regarding the innovation. The paper still has several novelty aspects. Out-of-plane automatic scanning, automatic segmentation of gland and nodule, using real-time image segmentation information for closing the control loop and visual servoing are some of these innovations.

The authors enrolled more volunteers to address some of the comments. New references have been added to paper. 29 more volunteers were enrolled to collect age, height, weight, gender information. They also recruited 19 patients under approved IRB protocol to evaluate the system performance and showed the promise of their proposed technique.

The paper is well written. I do not have additional comments. I believe that the paper is ready for publication.

Reviewer #6 (Remarks to the Author):

Dear Authors

Your manuscript offers useful data on a novel fully autonomous robotic ultrasound system (FARUS) for thyroid scanning on humans.

I have additional observations:

1. Abstract: You mention that "Experimental results on human participants demonstrate that FARUS can perform high-quality ultrasound scans in comparison with manual scans obtained by clinicians. Moreover, FARUS proved to be a reliable and effective tool for nodule diagnosis."

One could state that your results do not fully support these statements. I agree that FARUS can

perform high-quality ultrasound scans and may identify some but not all nodules. You state (page 21FARUS's inability to scan nodules effectively....) and have shown that FARUS missed 8 nodules diagnosed by doctors and found 4 nodules not identified by doctors, as implied by tables 3 and 4. Moreover, you have shown discrepancies in TI-RADS classification between FARUS and Doctors (table 1). Consequently, I would recommend that you rephrase your conclusion and make statements based on your results, something like the following: "Experimental results on human participants demonstrated that FARUS can perform high-quality ultrasound scans, close to manual scans obtained by clinicians, may detect thyroid nodules and can provide data on nodule characteristics for ACR TI-RADS calculation."

2. Introduction: You mention "We substantiated the validity of our approach by conducting a comparative evaluation of FARUS-driven diagnostic results for thyroid nodules against the established hospital benchmark. We have conducted extensive evaluations and provided evidence of the system's performance and reliability. Our work addresses the gap between existing research and clinical application by demonstrating the successful deployment of this system in a real-world clinical setting." Comment: You might want to reconsider some of these statements because you have actually showed based on your results that FARUS can be useful as an initial diagnostic approach, especially in remote areas, with many discrepancies between FARUS and doctor-based results, maybe something like: "We investigated the validity of our approach by conducting a comparative evaluation of FARUS-driven diagnostic results for thyroid nodules against the established hospital benchmark. We have conducted extensive evaluations and studied the system's performance and reliability. Our work addresses the gap between existing research and clinical application by demonstrating the deployment of this system in a real-world clinical setting."

3. Page 13, you mention " 2.5 Thyroid nodule imaging reporting with FARUS

The results of the scoring and classification process according to the ACR TIRADS are presented in Fig. 6. "

Comment: This is not accurate. It is a display of examples of nodules, as seen by the two techniques and examples of scoring and classification according to the ACR TIRADS. In Fig 6a you display 6 indicative nodules, as identified with FARUS and the respective US images provided by doctors. In 6b you display examples of two separate nodules, and explain their TIRADS scores.

4. Your results of ACR TIRADS classification are summarized in table 1. You should add a column stating the recommended management for each nodule based on the tirads score and the nodule size and you should summarize [and comment upon in the discussion section] discrepancies on patient management based on the FARUS and the Doctor's results.

5. In Figure 6b you mention Figure 6b illustrates the nodule's composition as partial, leading to Please replace partial with mixed (meaning mixed cystic and solid)

6. You mention" In this research, we introduce the FARUS method, which aims to scan, detect, and classify nodules gathered from 19 individuals. Furthermore, we compare the data of each patient with the evaluations provided by doctor. " You present data about nodules and patients in a confusing way. How many nodules were identified by each method? in how many patients? You do not provide data about final diagnoses (either histologic or stability on follow-up). Making statements about accuracy is not advisable, unless you state doctors to be the gold standard, and then you need to provide numbers of sensitivity and specificity (for presence of nodules and possibility for malignancy requiring FNA vs observation). You should mention data for nodules and not for patients and emphasize the possibility of FARUS missing a node that should be biopsied- does it exist? How often?

7. In page 22you mention "Moreover, It is quite noticeable to found that our proposed FARUS can detect nodules that are not detected by doctor. This observation may be attributed to the isoechoic properties of nodules, as the lack of significant contrast between isoechoic nodules and the surrounding thyroid tissue makes them more prone to being missed during examination." Please make a comment on the size of these, was it significantly smaller?

8. As a conclusion, it could be mentioned that although FARUS is feasible and may provide detection of nodules and data regarding their characteristics for ACR TI-RADS classification, further clinical studies are required to investigate whether it is safe to implement this system as a screening tool for probably or definitely malignant nodules.

Response to reviewers' comments:

Reviewer #1

General Comments: *The authors changed some of the contributions and focused on nodule diagnosis (including TIRADS score). This has improved the paper. The authors answered sufficiently to most of reviewer's comments. They extended the study, now also including patients. This is definitely valuable and more interesting.*

Response: Thank you for your comments and it is much appreciated.

Comment 1. *They explained their reasoning for having volunteers sit and detect patient movement. However, it is not clear which images they use or discard while they detect patient movement. Please explain this more clearly within the paper. This needs to be done during the minor revision.*

Response: Many thanks for the question! In this work, the movement of participants is detected in two ways. Large-scale movements can be detected through the Kinect skeleton camera, and subtle movements can be detected by segmenting the thyroid mask from the ultrasound image. From a safety perspective, the scan is terminated when Kinect detects significant movement. The mask centroid of thyroid segmentation mask can be used to compensate for subtle movements. We have added more details about the patient movement detection. According to your valuable suggestion, we have revised this content in method (highlighted in yellow on page 25)

Manuscript Revision

Before revision (page 27):

Before scanning, the participant sat on a chair and adjusted the sitting position, then the Kinect camera captured the image of the participant, and we estimated the position of the thyroid from the skeleton joint point in the captured image.

After revision (page 25):

Prior to scanning, participants assumed a seated position on a chair and adjusted their posture. Subsequently, the Kinect camera captured a depth map, which was then employed to generate a 3D point cloud of the environment. This point cloud underwent analysis for real-time detection and tracking of the human body joints by the Kinect body-tracking algorithm [50]. Then the position of the thyroid was estimated based on spatial information derived from tracked human skeleton joints..... During the scanning process, the participant was allowed to move slightly and the robot was able to adapt to the participant's movement through the force control and image servo. However, the scanning will be terminated if FARUS detects that the

participant moves more than 2 cm left or right within one second, or 5 cm forward or backward.

Comment 2. They now mention literature correctly in the related work. However, they still use some approaches for example in plane scanning and do not refer from which works in literature these are inspired by for the robot scanning methodology part. Please make sure that such inspirations are clearly mentioned and cited in the method section. These need to be clear and will not reduce the impact of the paper, which is on proving its hypothesis, but providing the citations at the right place are needed from an academic perspective. This needs to be done during the minor revision.

Response: Thank you for your valuable observations. Yes, we have added all the relevant references and cited them in the paper in revision. Specifically, the in-plane-scanning (IPS) procedure employed in this work draws inspiration from the methodology introduced in [34]. Real-time detection and tracking of the human body rely on the Kinect body-tracking algorithm [49]. The image control loop is governed by a visual servo control law [50]. Additionally, the risk stratification system, named ACR TI-RADS, is established based on the consensus of radiologists [47].

Manuscript Revision

Before revision:

The probe initiates a scan in the upward direction until the upper end of the thyroid lobe becomes invisible in the segmentation image. (page 25)

then the Kinect camera captured the image of the participant, and we estimated the position of the thyroid from the skeleton joint point in the captured image. (page 27)

According to the general form of a control law $\dot{s} = L_s v_c$, the control law for the force control task is given by: (page 28)

where λ_f is the force control gain, k is the soft tissues stiffness and s_f^* is the desired force value. (page 28)

For determining the robot's desired pose during the scan, use the following formula: (page 28)

After revision:

Inspired by the IPS procedure introduced in [34], the probe then scanned upward and downward until both ends of the thyroid lobe were no longer visible in the segmentation image. (page 24)

This point cloud underwent analysis for real-time detection and tracking of the human body joints by the Kinect body-tracking algorithm [50]. (page 25)

According to the visual servo control law $\dot{s} = L_s v_c$ [51], the control law for the force control task is given by: (page 26)

where λ_f is the force control gain, s_f^* is the desired force value and k is the human tissues stiffness that is usually set in [125, 500] N/m according to [52]. (page 26)

Similar to [34], the image-based sideways correction is carried out in an iterative manner: (page 27)

The ACR TI-RADS is a risk stratification system designed to help radiologists categorize thyroid nodules based on their ultrasound characteristics [47]. (page 27)

Comment 3. *The writing and explanations have improved but the authors should do another grammar and spelling check. This needs to be done during the minor revision.*

Response: Thank you again for your time in reviewing our paper. We have gone through the paper a few more times, trying to minimize the number of possible grammatical errors and typos.

Reviewer #5

General Comments:

The authors have addressed comments of both reviewers properly.

The softened the language regarding the innovation. The paper still has several novelty aspects. Out-of-plane automatic scanning, automatic segmentation of gland and nodule, using real-time image segmentation information for closing the control loop and visual servoing are some of these innovations.

The authors enrolled more volunteers to address some of the comments. New references have been added to paper. 29 more volunteers were enrolled to collect age, height, weight, gender information. They also recruited 19 patients under approved IRB protocol to evaluate the system performance and showed the promise of their proposed technique.

The paper is well written. I do not have additional comments. I believe that the paper is ready for publication.

Response: Thank you very much again for your time in reviewing out paper.

Reviewer #6

General Comments: *Your manuscript offers useful data on a novel fully autonomous robotic ultrasound system (FARUS) for thyroid scanning on humans.*

Response: Thank you for these comments!

Comment 1: *Abstract: You mention that “Experimental results on human participants demonstrate that FARUS can perform high-quality ultrasound scans in comparison with manual scans obtained by clinicians. Moreover, FARUS proved to be a reliable and effective tool for nodule diagnosis.” One could state that your results do not fully support these statements. I agree that FARUS can perform highquality ultrasound scans and may identify some but not all nodules. You state (page 21 ...FARUS’s inability to scan nodules effectively....) and have shown that FARUS missed 8 nodules diagnosed by doctors and found 4 nodules not identified by doctors, as implied by tables 3 and 4. Moreover, you have shown discrepancies in TI-RADS classification between FARUS and Doctors (table 1). Consequently, I would recommend that you rephrase your conclusion and make statements based on your results, something like the following: “Experimental results on human participants demonstrated that FARUS can perform high-quality ultrasound scans, close to manual scans obtained by clinicians, may detect thyroid nodules and can provide data on nodule characteristics for ACR TI-RADS calculation.”*

Response: Thank you for your suggestions and the relevant part of the abstract has been revised accordingly.

Manuscript Revision

Before revision (the abstract):

.... Experimental results on human participants demonstrate that FARUS can perform high-quality ultrasound scans in comparison with manual scans obtained by clinicians. Moreover, FARUS proved to be a reliable and effective tool for nodule diagnosis.

After revision (the abstract):

.... Experimental results on human participants demonstrated that FARUS can perform high-quality ultrasound scans, close to manual scans obtained by clinicians, may detect thyroid nodules and can provide data on nodule characteristics for ACR TI-RADS calculation.

Comment 2: *Introduction: You mention “We substantiated the validity of our approach by conducting a comparative evaluation of FARUS-driven diagnostic results for thyroid nodules against the established hospital benchmark. We have conducted extensive*

evaluations and provided evidence of the system's performance and reliability. Our work addresses the gap between existing research and clinical application by demonstrating the successful deployment of this system in a real-world clinical setting."

Comment: You might want to reconsider some of these statements because you have actually showed based on your results that FARUS can be useful as an initial diagnostic approach, especially in remote areas, with many discrepancies between FARUS and doctor-based results, maybe something like: "We investigated the validity of our approach by conducting a comparative evaluation of FARUS-driven diagnostic results for thyroid nodules against the established hospital benchmark. We have conducted extensive evaluations and studied the system's performance and reliability. Our work addresses the gap between existing research and clinical application by demonstrating the deployment of this system in a real-world clinical setting."

Response: Thank you again for your suggestions and it has been incorporated accordingly (highlighted in yellow on page 3).

Manuscript Revision

Before revision:

We substantiated the validity of our approach by conducting a comparative evaluation of FARUS-driven diagnostic results for thyroid nodules against the established hospital benchmark. We have conducted extensive evaluations and provided evidence of the system's performance and reliability. Our work addresses the gap between existing research and clinical application by demonstrating the successful deployment of this system in a real-world clinical setting.

After revision:

We investigated the validity of our approach by conducting a comparative evaluation of FARUS-driven diagnostic results for thyroid nodules against the established hospital benchmark. We have conducted extensive evaluations and studied the system's performance and reliability. Our work addresses the gap between existing research and clinical application by demonstrating the deployment of this system in a real-world clinical setting.

Comment 3: Page 13, you mention "2.5 Thyroid nodule imaging reporting with FARUS The results of the scoring and classification process according to the ACR TIRADS are presented in Fig. 6. "Comment: This is not accurate. It is a display of examples of nodules, as seen by the two techniques and examples of scoring and classification according to the ACR TIRADS. In Fig 6a you display 6 indicative nodules, as identified with FARUS and the respective US images provided by doctors. In 6b you display examples of two separate nodules, and explain their TIRADS scores.

Response: Thank you for your detailed review. Upon careful consideration, we agree with you on these descriptions.

In the new version, we have revised the subtitle in accordance with your suggestion to better reflect the content of our paper. The new subtitle is "ACR TI-RADS Risk Stratification using FARUS." Secondly, we have revised Fig 6 captions and corresponding descriptions in the manuscript.

Manuscript Revision

Before revision:

Subtitle: 2.5 Thyroid nodule imaging reporting with FARUS

Figure 6 captions:

- a. Comparison of the thyroid nodules found by FARUS and doctor.
- b. Result of scoring and classification diagnosed by FARUS based on ACR-TIRADS.

Descriptions in the manuscript:

Figure 6 a illustrates scoring and classification of nodule diagnosed by doctor and FARUS. This can be attributed to the ultrasound image's appropriate brightness and contrast, resulting in well-distributed pixels.

Figure 6 b shows the nodule's composition is categorized as solid, leading to a score of 2, while its echogenicity is classified as hypo, also resulting in a score of 2.

After revision(page 13 and 17):

Subtitle: 2.5 ACR TI-RADS Risk Stratification using FARUS

Figure 6 captions:

- a. Indicative nodules identified with FARUS and the respective US images provided by doctors
- b. Examples of two separate nodules, with accompanying explanations of their TIRADS scores.

Descriptions in the manuscript:

Additionally, we conduct a comparative analysis between FARUS-generated classifications and evaluations provided by professional sonographer. Figure 6a displays five nodules that demonstrate complete agreement with the sonographer's diagnosis.

Results from the scoring and classification process by FARUS, following ACR TI-RADS criteria, are depicted in Fig 6b.

Comment 4. Your results of ACR TIRADS classification are summarized in table 1. You should add a column stating the recommended management for each nodule based on the tirads score and the nodule size and you should summarize [and comment upon in the discussion section] discrepancies on patient management based on the FARUS and the Doctor's results.

Response: Thank you for your insightful comments and suggestions. In revision, we have added another column in Table 1 to include the recommended management for each nodule. This is based on the TIRADS score and the corresponding nodule size, providing a more comprehensive view of the results. Furthermore, we have included a dedicated section in the discussion to thoroughly summarize and comment upon these variations. **Please refer to updated Table1 and highlighted parts of the manuscript for more details (on page 15-17).**

Manuscript Revision

After revision:

The doctor's report indicates that out of the 19 patients, one patient has two nodules requiring fine-needle aspiration (FNA) and follow-up, respectively. For this patient, FARUS results were consistent with the doctor's assessment. In the cases of the nodule #5, diagnostic inconsistency arose from variations in echo characteristics, leading to divergent recommended management.

Comment 5. In Figure 6b you mention Figure 6b illustrates the nodule's composition as partial, leading to Please replace partial with mixed (meaning mixed cystic and solid).

Response: Thank you for your comments. The relevant part has been revised accordingly.

Manuscript Revision

Before revision (Page 13):

Figure 6b illustrates the nodule's composition as partial, leading to a score of 1, and its echogenicity as hypo, resulting in a score of 2.

After revision (Page 13):

Figure 6b illustrates the nodule's composition as mixed, leading to a score of 1, and its echogenicity as hypo, resulting in a score of 2.

Comment 6: *You mention " In this research, we introduce the FARUS method, which aims to scan, detect, and classify nodules gathered from 19 individuals. Furthermore, we compare the data of each patient with the evaluations provided by doctor. " You present data about nodules and patients in a confusing way. How many nodules were identified by each method? in how many patients? You do not provide data about final diagnoses (either histologic or stability on follow-up). Making statements about accuracy is not advisable, unless you state doctors to be the gold standard, and then you need to provide numbers of sensitivity and specificity (for presence of nodules and possibility for malignancy requiring FNA vs observation). You should mention data for nodules and not for patients and emphasize the possibility of FARUS missing a node that should be biopsied-does it exist? How often?*

Response: Thank you for your comments and we apologize for the unclear description in the previous submission.

In previous submission, we focused more on patient data rather than nodule data, in order compare directly between the FARUS system and sonographer ACR-TIRADS ratings.

However, we agree with you, and have added data about nodules and removed statements on accuracy in revision. In the revised version, Table 1, 2 and 3 presents a comparison of nodules detected by both sonographers and FARUS for the 19 patients in the study. Among the 19 patients, the doctor reported a total of 32 nodules that aligned quite well with the FARUS system's findings, detecting a total of 28 suspected nodules. If we considered the doctor's results as being correct and used them as a reference, the FARUS accurately identified 24 nodules.

Detecting small-sized nodules poses a challenge for FARUS, since it was originally designed for remote communities without professional doctors, so it prioritized preliminary screening over full-fledged functionalities. The probe of professional equipment used by doctor consistently produces a specific color of the thyroid gland for each patient, while the brightness and contrast of ultrasound images produced by FARUS's probe may vary across different patients with varying ages and weights.

Additionally, the FARUS system missed two larger-sized nodules (#31 and #32). Challenges arise with isoechoic nodules (#32) or mixed types of nodules (#31) due to low contrast with the thyroid, making identification on ultrasound images inherently difficult. Artifacts in ultrasound images were not considered in our current algorithm.

In this work, the doctor's report indicates that out of the 19 patients, with one patient having two nodules requiring fine-needle aspiration (FNA) and follow-up, respectively. For this high-risk patient, FARUS results were consistent with the doctor's assessment. These limitations and the direction of improvement for the current FARUS are discussed in greater detail in the subsequent sections in revision.

Manuscript Revision

After revision (Page 13, 21 and 22):

The scoring and recommended management of 24 nodules among the 13 individuals detected by both FARUS and the doctor were compared. Each nodule detected by FARUS was matched to the doctor's report based on its location and shape. Table 1 shows the comparison between FARUS and doctor scoring and the classification of thyroid nodules (please refer to Supplementary Table S1 for more details). Among the 24 nodules assessed, 10 were diagnosed with the same score distribution by both the doctor and the developed FARUS. This can be attributed to the ultrasound image's appropriate brightness and contrast, resulting in well-distributed pixels. Additionally, 8 nodules exhibited a score difference of 1, primarily attributed to discrepancies in echogenicity or composition classifications. There were four nodules with a score difference of 2 and another with a score difference of 4, possibly caused by different classifications in both echogenicity and composition. Nodule size variations in the scan may be attributed to differences in patient positioning: the doctor scans in a supine position, while FARUS scans in an upright sitting position due to security concerns.

The results of this study indicate that smaller nodules, such as #25, #26, #27, #28, #29 and #30 present greater difficulty for FARUS to detect. Similarly, nodules with isogenic properties, such as #31 and #32, also pose challenges for FARUS's detection capabilities.

The FARUS needs improvement in the future due to its limitations, especially for small-scale and low-contrast nodules. The existing nodule dataset lacks sufficient diversity in terms of nodule size. Artifacts in ultrasound images were not considered in our current algorithm. Additionally, work needs to be done to incorporate video streams.

Comment 7. In page 22 you mention “Moreover, It is quite noticeable to found that our proposed FARUS can detect nodules that are not detected by doctor. This observation may be attributed to the isoechoic properties of nodules, as the lack of significant contrast between isoechoic nodules and the surrounding thyroid tissue makes them more prone to being missed during examination.” Please make a comment on the size of these, was it significantly smaller?

Response: We have consulted multiple doctors and they have shown quite different opinions on the identified anomalies, and unanimity could not be reached among the doctors regarding the characterization of these anomalies as nodules. Consequently, we have classified these instances as possibly false positives.

Manuscript Revision

After revision (Page 21):

Moreover, the FARUS identified some nodules for which the doctor did not, as shown in Table 3. We sought opinions from multiple doctors and a firm conclusion could not be reached by doctors. In an exercise of prudence, we considered these occurrences as possibly false positives.

Comment 8. *As a conclusion, it could be mentioned that although FARUS is feasible and may provide detection of nodules and data regarding their characteristics for ACR TI-RADS classification, further clinical studies are required to investigate whether it is safe to implement this system as a screening tool for probably or definitely malignant nodules.*

Response: Thank you and your comments have been incorporated in revision as follows.

Manuscript Revision

Before revision:

The effectiveness of the FARUS system in nodule detection requires improvement. The current ultrasound equipment is insufficient for accurate screening of small-scale, low-contrast nodules. Additionally, the size and diversity of the existing nodule dataset are inadequate. Moreover, the algorithm does not consider the timing of data, which is essential in ultrasound doctors' practice of combining video data to analyze suspected nodules. The contact force between the US probe and the skin needs to be customized for different groups of people, and there is still available room for improvement.

After revision (Page 18):

However, the volunteering participants currently recruited typically have low-risk nodules, and clinical FNA is not advised. Although FARUS has demonstrated feasibility and potentiality in nodule detection and data collection for ACR TI-RADS classification, further clinical studies are essential to assess its safety as a screening tool for probably or definitely malignant nodules.

REVIEWERS' COMMENTS

Reviewer #1 (Remarks to the Author):

The authors have addressed the comments properly.

The authors adjusted the contribution and added correct references where needed. The overall contribution shows novelty and is of great interest for the community.

The paper improved in both writing and in figure and table design.

As a minor comment: the authors did not specify gender/sex and exact age range of the participants and patients. I suggest that the authors add this information.

Otherwise, in my opinion the paper is ready for publication.

Reviewer #6 (Remarks to the Author):

Dear Authors

The revised manuscript offers important information and you have made all necessary changes towards clarity, accuracy and increased importance of your work. Thank you for your hard work and congratulations.

Reviewer #6 (Remarks on code availability):

No opinion.

Response to reviewers' comments:

Reviewer #1

General Comments: *The authors have addressed the comments properly. The authors adjusted the contribution and added correct references where needed. The overall contribution shows novelty and is of great interest for the community. The paper improved in both writing and in figure and table design.*

Response: Thank you very much again for your time in reviewing our paper.

Comment 1. *As a minor comment: the authors did not specify gender/sex and exact age range of the participants and patients. I suggest that the authors add this information. Otherwise, in my opinion the paper is ready for publication.*

Response: Many thanks for the comment! According to your suggestion, we have added gender/sex and age range of the participants and patients in revised paper (page 12-13).

Manuscript Revision

Before revision (page 9):

...The first group predominantly comprised college students, and we manually collected thyroid ultrasound data from 66 volunteers within this group using handheld US equipment. Simultaneously, we employed FARUS system to autonomously scan 70 volunteers. To address the limitations of handheld US equipment in accurately diagnosing nodules, we opted to employ portable US equipment to gather two additional sets of data. The second set of data was obtained from thyroid US scans of 29 middle-aged and elderly individuals within the community, chosen specifically to facilitate the training of the nodule segmentation network. Finally, the third group was composed of 19 patients who underwent robotic thyroid scanning.

After revision (page 12-13):

... The first group predominantly comprised college students, and we manually collected thyroid US data from 66 volunteers within this group using handheld US equipment. Simultaneously, we employed FARUS system to autonomously scan 70 volunteers (20–30 years of age, 19 females, 41 males), 13 of whom were scanned manually by 5 doctors. To address the limitations of handheld US equipment in accurately diagnosing nodules, we opted to employ portable US equipment to gather two additional sets of data. The second set of data was obtained from thyroid US scans of 29 middle-aged and elderly individuals within the community, chosen specifically to facilitate the training of the nodule segmentation network. Finally, the third group was composed of 19 patients (age 53.05 ± 5.90 years old, 12 females, 7 males) who underwent robotic thyroid scanning.

Reviewer #6

General Comments: *The revised manuscript offers important information and you have made all necessary changes towards clarity, accuracy and increased importance of your work. Thank you for your hard work and congratulations.*

Response: Thank you for your comments!